# A force-sensitive mutation reveals a non-canonical role for dynein in anaphase progression

David Salvador-Garcia[1], Li Jin[1], Andrew Hensley[2], Mert Gölcük[3], Emmanuel Gallaud[4], Sami Chaaban[5], Fillip Port[1], Alessio Vagnoni[1], Vicente José Planelles-Herrero[1], Mark A. McClintock[1], Emmanuel Derivery[1], Andrew P. Carter[5], Régis Giet[4], Mert Gür[3,6], Ahmet Yildiz[2,7], and Simon L. Bullock[1]

**The diverse roles of the dynein motor in shaping microtubule networks and cargo transport complicate in vivo analysis of its functions significantly. To address this issue, we have generated a series of missense mutations in *Drosophila* Dynein heavy chain. We show that mutations associated with human neurological disease cause a range of defects, including impaired cargo trafficking in neurons. We also describe a novel microtubule-binding domain mutation that specifically blocks the metaphase–anaphase transition during mitosis in the embryo. This effect is independent from dynein's canonical role in silencing the spindle assembly checkpoint. Optical trapping of purified dynein complexes reveals that this mutation only compromises motor performance under load, a finding rationalized by the results of all-atom molecular dynamics simulations. We propose that dynein has a novel function in anaphase progression that depends on it operating in a specific load regime. More broadly, our work illustrates how in vivo functions of motors can be dissected by manipulating their mechanical properties.**

## Introduction

Motors of the dynein and kinesin families drive movement of a wide variety of cellular constituents along microtubules and provide pushing and pulling forces that shape microtubule networks. The cytoplasmic dynein-1 (dynein) motor is responsible for almost all microtubule minus-end-directed motor activity in the cytoplasm (Reck-Peterson et al., 2018). Dynein has six subunits—a heavy chain, an intermediate chain, a light intermediate chain, and three light chains—that are each present in two copies per complex (Fig. 1 A). The heavy chain has over 4,600 amino acids and comprises an N-terminal tail domain, which mediates homodimerization and association with other subunits, and a C-terminal motor domain.

The key elements of the motor domain are a ring of six AAA+ ATPase domains, a linker, and an anti-parallel coiled-coil stalk that leads to a globular microtubule-binding domain (MTBD) (Fig. 1 A; Schmidt and Carter, 2016). ATP binding to the AAA1 domain triggers conformational changes in the ring that are transmitted via a shift in the stalk's registry to the MTBD. This event lowers the affinity of the MTBD for the microtubule, leading to detachment. ATP hydrolysis and product release allow the MTBD to rebind the microtubule, initiating a force-producing linker swing. Repeated cycles of these events translocate dynein along the microtubule. The motor's processivity and force output are increased markedly by simultaneous binding of two cofactors to the tail domain: the multisubunit dynactin complex and one of a number of coiled-coil–containing cargo adaptors (termed "activating adaptors") (McKenney et al., 2014; Schlager et al., 2014; Belyy et al., 2016).

Dynein's diverse cellular roles make it very challenging to assess its functions in discrete processes. Mutations that strongly disrupt dynein function are not compatible with survival (Gepner et al., 1996; Harada et al., 1998; Mische et al., 2008), and acute inhibition of dynein with antibodies (Gaglio et al., 1997; Sharp et al., 2000; Yang et al., 2007; Yi et al., 2011) or small molecules (Firestone et al., 2012; Steinman et al., 2017; Hoing et al., 2018) impairs multiple processes that can be difficult to disentangle.

[1]Cell Biology Division, Medical Research Council Laboratory of Molecular Biology, Cambridge, UK; [2]Department of Physics, University of California, Berkeley, Berkeley, CA, USA; [3]School of Mechanical Engineering, Istanbul Technical University, Istanbul, Turkey; [4]Institut de Génétique et Développement de Rennes, Université de Rennes, Rennes, France; [5]Structural Studies Division, Medical Research Council Laboratory of Molecular Biology, Cambridge, UK; [6]Department of Computational and Systems Biology, University of Pittsburgh, Pittsburgh, PA, USA; [7]Department of Molecular and Cellular Biology, University of California, Berkeley, Berkeley, CA, USA.

Correspondence to Simon L. Bullock: sbullock@mrc-lmb.cam.ac.uk

D. Salvador-Garcia's current affiliation is the Randall Centre of Cell and Molecular Biophysics, King's College London, London, UK.   F. Port's current affiliation is the Division of Signaling and Functional Genomics, German Cancer Research Center, Heidelberg, Germany.   A. Vagnoni's current affiliation is the Department of Basic and Clinical Neuroscience, Institute of Psychiatry, Psychology and Neuroscience, King's College London, London, UK.

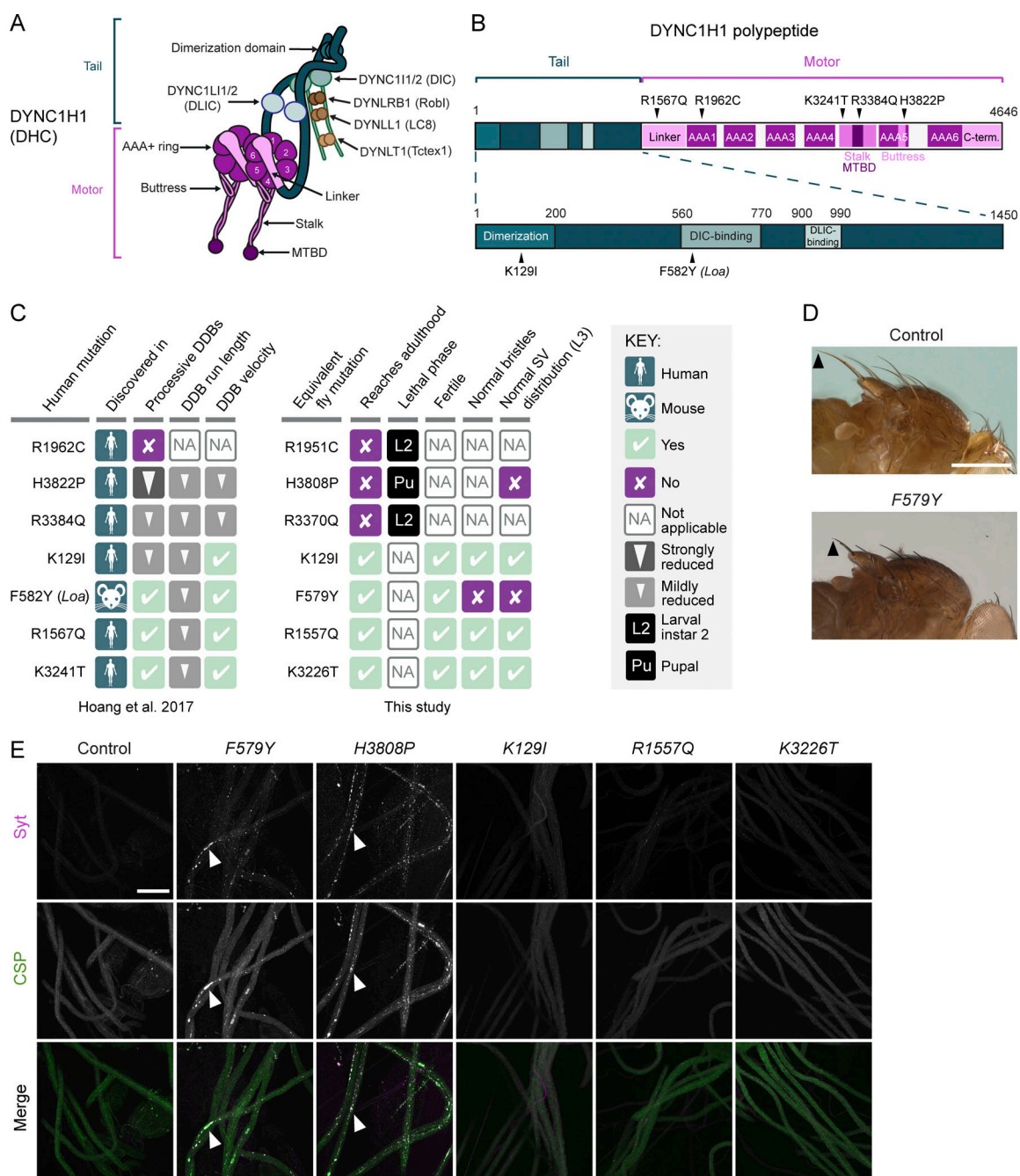

Figure 1. **Dynein organization and phenotypic analysis of Dhc disease-associated mutations in *Drosophila*. (A)** Cartoon of human dynein complex with alternative nomenclature for subunits shown. The C-terminal domain of DYNC1H1 is not visible in this view because it lies on the other face of the AAA+ rings. **(B)** Positions in human DYNC1H1 of the disease-associated mutations characterized in this study. Domains of DYNC1H1 are color-coded as in panel A. Mouse *Loa* mutation is numbered according to the equivalent residue in human DYNC1H1. Adapted from Hoang et al. (2017). **(C)** Summary of in vitro and in vivo effects of disease-associated mutations. SV, synaptic vesicle. In vitro effects refer to when both copies of DYNC1H1 in the dynein complex contain the mutation; in vivo effects refer to the homozygous condition. **(D)** Images showing short bristles on the notum of homozygous *F579Y* adult flies compared to controls (*yw*). Arrowheads point to posterior scutellar macrochaetae as an example. The bristle phenotype of *F579Y* flies was completely penetrant (>160 flies examined). **(E)** Confocal images of segmental nerves (taken proximal to the ventral ganglion; anterior to top; Z-projections) from fixed L3 larvae stained for the synaptic vesicle proteins Synaptotagmin (Syt) and Cysteine-string protein (CSP). Arrowheads show examples of synaptic vesicle accumulations in a subset of mutant genotypes. Images are representative of three to six larvae analyzed per genotype. Scale bars: D, 500 µm; E, 50 µm.

A striking illustration of the complexity of studying dynein function is in mitosis. Here, the motor has been implicated in centrosome separation, nuclear envelope breakdown, spindle pole focusing, attachment of kinetochores to microtubules, chromosome congression, creating tension between sister kinetochores, and silencing the spindle assembly checkpoint (SAC) by stripping regulatory factors from kinetochores (Hinchcliffe and Vaughan, 2017). Dynein was also proposed to

be directly responsible for segregating chromosomes in anaphase (Savoian et al., 2000; Sharp et al., 2000; Yang et al., 2007). However, it has since been argued that dynein inhibition impairs chromosome movement indirectly by interfering with earlier, kinetochore-related functions of the motor (Maddox et al., 2002; Bader and Vaughan, 2010; Vukusic et al., 2019).

In this study, we have taken a combined genetic and biochemical approach to dissect dynein's in vivo functions. We document the organismal and cellular effects of a large number of Dynein heavy chain missense mutations in *Drosophila*. As well as characterizing several mutations that are associated with human neurological disease, we describe a novel mutation in the MTBD that, remarkably, causes metaphase arrest of mitotic spindles in embryos but does not impair other dynein-dependent processes. The results of in vivo, in vitro, and in silico analysis of this mutation lead us to propose the existence of a novel function for dynein in promoting anaphase that is independent from its canonical role in SAC silencing and depends on the motor operating in a specific force regime.

## Results

### Characterization of disease mutations in *Drosophila*

We first set out to study the in vivo effects of six disease-linked missense mutations in the human Dynein heavy chain (DYNC1H1): K129I, R1567Q, R1962C, K3241T, R3384Q, and H3822P (Fig. 1 B). Heterozygosity for each of these mutations is associated with malformations in cortical development and intellectual disability (Schiavo et al., 2013). We previously showed using in vitro reconstitution that these mutations perturb, to varying degrees, the processive movement of human dynein complexes bound to dynactin and the N-terminal region of the prototypical activating adaptor, BICD2 (so-called "DDB" complexes) (Hoang et al., 2017; summarized in Fig. 1 C).

We used CRISPR/Cas9-based homology-directed repair (HDR) to generate *Drosophila melanogaster* strains with analogous mutations in the gene encoding Dynein heavy chain (*Dhc64C*, hereafter *Dhc*): K129I, R1557Q, R1951C, K3226T, R3370Q, and H3808P (Fig. 1 C). We also made a strain with the equivalent of the human F582Y mutation (F579Y), which corresponds to the *Loa* allele that causes neurodegeneration in heterozygous mice (Hafezparast et al., 2003) and also impairs movement of DDB in vitro (Hoang et al., 2017; Fig. 1 C).

None of the mutations caused overt phenotypes when heterozygous in flies. However, animals homozygous for the human R1962C, R3384Q, and H3822P equivalents (R1951C, R3370Q, and H3808P) failed to reach adulthood (Fig. 1 C). Complementation tests with a *Dhc* null allele confirmed that lethality was not due to off-target activity of Cas9 (Fig. S1 A). Like homozygous *Dhc* null mutants, *R1951C* and *R3370Q* mutants died during the second larval instar stage (L2) (Fig. 1 C and Fig. S1 B). Thus, these disease-associated mutations strongly disrupt dynein function. In contrast, *H3808P* mutants typically died as pupae (Fig. 1 C and Fig. S1 B), demonstrating residual dynein activity in these animals. Of the four homozygous viable mutations, only F579Y caused a morphological defect in adults; these mutants had

abnormally short bristles on the notum (Fig. 1 D), which is a feature of several classical hypomorphic *Dhc* mutations (Gepner et al., 1996; Melkov et al., 2016).

We previously proposed that the pathomechanism of the disease-associated mutations involves impaired cargo trafficking in neurons (Hoang et al., 2017). To investigate this notion, we used immunostaining to compare the distribution of synaptic vesicles in motor neuron axons of wild-type and mutant third instar larvae (L3). These vesicles undergo dynein- and kinesin-1–dependent bidirectional transport (Miller et al., 2005) and accumulate in large foci when either motor is disrupted (Martin et al., 1999). K129I, R1557Q, and K3226T did not cause focal accumulation of synaptic vesicles in axons (Fig. 1 E), consistent with our failure to detect any other phenotypes associated with these mutations. In contrast, homozygosity for the H3808P or F579Y mutations resulted in large axonal foci of synaptic vesicles (Fig. 1 E). This finding shows that at least some disease-associated mutations interfere with cargo translocation in neurons.

Fig. 1 C summarizes the effects of the disease-associated mutations in *Drosophila* and on DDB motility in vitro (Hoang et al., 2017). The three mutations with the strongest effects on DDB movement—R1962C, R3384Q, and H3822P—were those whose counterparts caused lethality in flies. However, the relative strengths of the in vitro and in vivo effects of the disease mutations were not equivalent. Even though H3822P impaired DDB motility more strongly than R3384Q, the analogous mutation in flies (H3808P) resulted in later lethality than the R3384Q equivalent. The milder phenotype of the H-to-P mutation in *Drosophila* was particularly surprising as it also strongly disrupts dynein function in budding yeast (Marzo et al., 2019). Moreover, whereas the effect of the human F582Y mutation on DDB motility was very similar to that of R1567Q and K3241T, and weaker than that of K129I, the equivalent *Drosophila* mutation (F579Y) was unique among this group in altering bristle morphology and synaptic vesicle distribution. These observations may reflect some mutations having species-specific effects or altering dynein's interplay with proteins in addition to dynactin and the BICD2 ortholog, BicD. Nonetheless, the range of mutant phenotypes observed in *Drosophila* and the overall correlation with the strength of inhibitory effects in vitro indicate that the fly is a valuable model for structure-function studies of dynein.

### The novel mutation S3372C specifically impairs an embryonic function of dynein

While generating the disease-associated mutations, we recovered four novel in-frame Dhc mutations that were caused by imprecise repair of Cas9-induced DNA breaks. L1952K and the 2-amino-acid deletion ΔL3810Y3811, which came from the experiments that produced R1951C and H3808P, respectively, caused lethality in homozygotes (Fig. S1 C) or when combined with the null allele (Fig. S1 A). Homozygosity for these mutations also resulted in L3 arrest and focal accumulation of synaptic vesicles in L3 motor neuron axons (Fig. S1, C and D). Y3811F was also recovered while making H3808P, although the homozygous mutants lacked detectable abnormalities (Fig. S1, A and C).

The fourth novel mutation—S3372C (a by-product of our efforts to make R3370Q)—had a particularly striking phenotype.

While *S3372C* homozygous adults were recovered at the normal Mendelian ratio and males were fertile, homozygous females had very strongly impaired fertility (<0.03% of eggs hatching into larvae; Fig. 2 A). No fertility issues were observed in *S3372C* heterozygous females (Fig. 2 A), showing that this is a recessive effect. Like *S3372C* homozygotes, trans-heterozygotes for the *S3372C* allele and the *Dhc* protein null allele (*S3372C/–*) also had normal survival to adulthood (Fig. S1 A) and female-specific infertility (Fig. 2 A). This observation reveals that the fertility defect is caused by the S3372C mutation, a conclusion corroborated by the ability of a wild-type *Dhc* transgene to restore fertility to *S3372C* females (Fig. S1 E).

In contrast to S3372C, the non-lethal disease-associated *Dhc* mutations did not strongly affect fertility (Fig. 1 C). For example, only 30% of embryos laid by *F579Y* homozygotes failed to hatch (compared to 10% of control embryos) (Fig. 2 A). Moreover, despite having much stronger fertility defects, *S3372C* homozygous and *S3372C/–* animals did not have the bristle defects (Fig. 2 B and Fig. S1 F) or axonal foci of synaptic vesicles (Fig. 2 C and Fig. S1 G) observed with F579Y. These observations reveal that S3372C only affects a subset of dynein functions.

To further explore which dynein-dependent processes are affected by S3372C, we tracked movement of GFP-labeled mitochondria in axons of the adult wing nerve. In this system, dynein and kinesin-1 are responsible for retrograde and anterograde mitochondrial movements, respectively (Vagnoni et al., 2016). F579Y was used as a positive control for the experiments, as the equivalent mutation impairs dynein-based cargo transport in other organisms (Hafezparast et al., 2003; Ori-McKenney et al., 2010; Sivagurunathan et al., 2012). Homozygosity for the *F579Y* allele reduced the frequency of retrograde and bidirectional transport of mitochondria without impairing retrograde velocity or travel distance, or any aspect of anterograde motion (Fig. 2, D and E; and Fig. S1, H and I). In contrast, S3372C did not impair any aspect of retrograde or anterograde motion of these organelles (Fig. 2, D and E; and Fig. S1, H and I). We also assessed the effect of S3372C on division of neuroblasts in live L3 brains expressing fluorescent α-tubulin. Inhibiting dynein–dynactin function in these neuroblasts perturbs spindle morphology and delays completion of mitosis (Wojcik et al., 2001; Siller et al., 2005). In contrast, no spindle abnormalities or mitotic delays were observed in *S3372C/–* neuroblasts (Fig. 2 F).

Taken together, our phenotypic analyses (summarized in Fig. 2 G) reveal a selective maternal effect of S3372C on dynein function during embryogenesis. A trivial reason for this observation could be that the mutation destabilizes Dhc protein specifically at embryonic stages. However, immunoblotting of embryo extracts with a Dhc antibody showed this is not the case (Fig. 2 H). Thus, S3372C affects the functionality, rather than the level, of Dhc in embryos.

### The conserved location of S3372 indicates an important role in dynein function
S3372 is located in Dhc's MTBD. To gain insight into how mutating this residue might affect dynein activity, we examined its position within the tertiary structure of this domain. Although an experimentally determined MTBD structure is not available

for *Drosophila* Dhc, we could confidently infer the position of S3372 from a 4.1-Å-resolution cryo-EM structure of the closely related mouse MTBD bound to an α/β-tubulin dimer (Fig. 3 A; Lacey et al., 2019). The equivalent residue in mouse (S3384) is located in a short loop that connects helix 6 (H6) of the MTBD—which directly contacts α-tubulin—to the base of coiled-coil 2 (CC2) of the stalk (Fig. 3, A and B).

We also used Alphafold2 (Jumper et al., 2021) to produce high-confidence structural predictions of the MTBD and stalk of the *Drosophila* and human dynein-1 heavy chains (Fig. 3 B and Fig. S2 A). In both these structures, serine occupied the same position as observed in the mouse MTBD structure. This serine was also conserved in the primary sequence of all dynein-1, dynein-2, and axonemal dynein heavy chain sequences examined, including those from single-cell eukaryotes (Fig. 3 C). High confidence predictions of human dynein-2 heavy chain and axonemal DYH7 structures from Alphafold2 indicated that the location of the serine in a loop between H6 and CC2 is also conserved (Fig. S2, A and B). These analyses suggest that the serine mutated in the *S3372C* fly strain plays an important role across the dynein family.

### Ectopic disulfide bond formation does not account for the S3372C phenotype
The structural analysis of the mutated serine additionally revealed that, while there is no nearby cysteine on the α/β-tubulin dimer, there is one at the base of CC2 in dynein-1 family members (Fig. 3, B and C). This observation led us to hypothesize that S3372C results in ectopic disulfide bonding with the equivalent CC2 cysteine in *Drosophila* (C3375). There is a global increase in oxidation at the egg-to-embryo transition (Petrova et al., 2018), which could conceivably trigger disulfide bonding at this site. Such an event could impair dynein's dynamics in the embryo and thereby explain arrest at this stage. To test this notion, we used HDR to change S3372 to cysteine and C3375 to serine in the same Dhc polypeptide, thus eliminating the potential for disulfide bonding at these positions (Fig. 4 A). Like *S3372C* homozygous females, females homozygous for this allele (*S3372C + C3375C*) were almost completely infertile (Fig. 4 B). This effect was not associated with the C3375S change, as the fertility of females homozygous for this mutation was only modestly reduced (Fig. 4 B). Thus, ectopic disulfide bonding within Dhc does not account for the S3372C-induced embryonic arrest.

### S3372C causes metaphase arrest of early nuclear divisions
Our finding that S3372C does not destabilize Dhc in the embryo or cause ectopic disulfide bonding pointed to a more nuanced effect on dynein activity, possibly related to a specific function during development. To elucidate which motor function is affected by the mutation, we investigated the embryonic phenotype in more detail. DNA staining of fixed embryos showed that the vast majority of embryos from *S3372C* homozygous mothers did not reach syncytial blastoderm stages, with ~80% arresting during early nuclear cleavage cycles (Fig. 5, A and B). Development of *S3372C* embryos to this stage was dependent on fertilization, as only a single nucleus was present in unfertilized

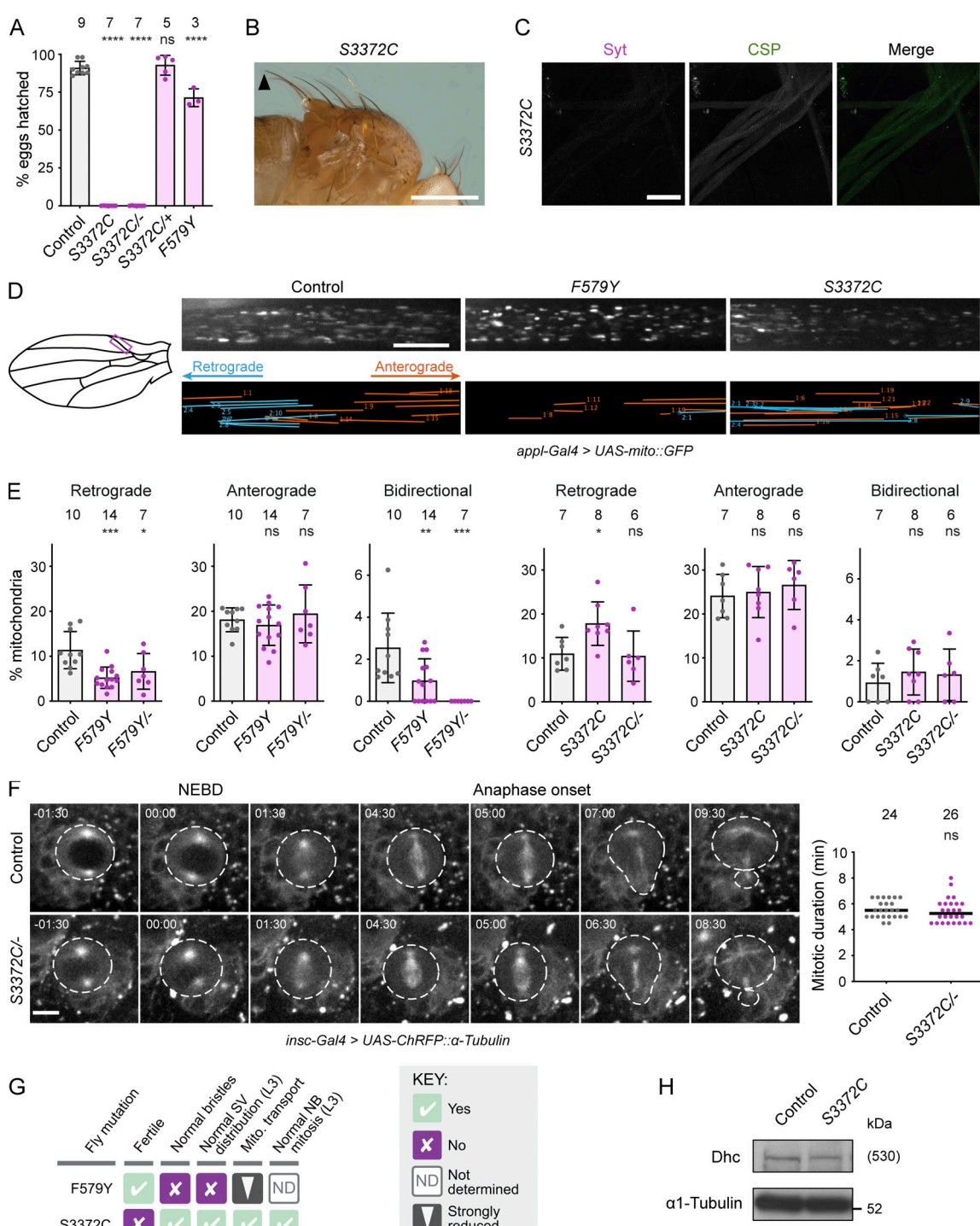

Figure 2. **The novel mutation S3372C selectively affects embryonic development. (A)** Hatching frequency of eggs laid by mated females of the indicated genotypes. Columns show mean values per egg collection; error bars represent SD; circles are values for individual egg collections. Number of collections per genotype (from three independent crosses) is shown above columns (114–575 eggs per collection). *S3372C/−* are trans-heterozygous for *S3372C* and a *Dhc* null allele. Control genotype: *yw*. **(B and C)** Images showing (B) normal bristle length in adults and (C) lack of synaptic vesicle accumulations in L3 segmental nerves (proximal to the ventral ganglion; anterior to the top; Z-projection) in *S3372C* homozygotes (for comparisons with controls, see Fig. 1, D and E). The arrowhead in B points to posterior scutellar macrochaetae. Images in B and C are representative of >160 flies and three larvae analyzed, respectively. **(D)** Analysis of mitochondrial motility in wing nerve axons of 2-day-old control, *F579Y*, and *S3372C* adult flies. Left, cartoon of wing region imaged (magenta box). Top images, example stills from 3-min time series (single focal plane) of fluorescent mitochondria (expression of mito::GFP with a pan-neuronal driver). Bottom images, traces of motile mitochondria in corresponding time series. Blue, retrograde tracks; orange, anterograde tracks. **(E)** Percentages of mitochondria transported in the retrograde or anterograde directions, or moving bidirectionally, during the 3 min of data acquisition. Columns show mean values per movie; error bars represent SD; circles are values for individual movies (each from a different wing). Number of wings analyzed shown above bars. Note that we observed an increased frequency of retrograde transport in *S3372C* homozygotes, but this was not recapitulated in *S3372C/−* animals. **(F)** Analysis of mitotic duration in

neuroblasts (NBs) in control and *S3372C/−* L3 larval brains. Left: Stills from image series of NBs with fluorescently labeled spindles (expression of RFP-tagged α-tubulin with an NB-specific driver). NBs and daughter cells highlighted with dashed lines. Timestamps are min:s after nuclear envelope breakdown (NEBD). Right: Quantification of mitotic duration (NEBD to anaphase onset). Lines show medians; circles are values for individual NBs. Number of NBs analyzed (from four wild-type or five *S3372C/−* larvae) shown above plot. **(G)** Summary of in vivo effects of homozygosity for *F579Y* and *S3372C*. SV, synaptic vesicle; Mito., mitochondrial. **(H)** Immunoblot of extracts of embryos from control and *S3372C* mothers (0–160-min collections), probed with antibodies to Dhc and α1-tubulin (loading control). The position of the molecular weight (Mw) marker is shown for α1-tubulin blot region; as there is no marker where Dhc migrates, the predicted Mw of Dhc is shown in parentheses. Evaluation of statistical significance (compared to control) was performed with a one-way ANOVA with Dunnett's multiple comparisons test (A and E) or a Mann-Whitney test (F): ****, $P < 0.0001$; ***, $P < 0.001$; **, $P < 0.01$; *, $P < 0.05$; ns, not significant. Scale bars: B, 500 µm; C, 50 µm; D, 10 µm; F, 5 µm. Source data are available for this figure: SourceData F2.

mutant eggs (Fig. S2 C). Arrest during early cleavage stages was also typical for embryos from *S3372C/−* mothers (Fig. S2 C), demonstrating the causal nature of the S-to-C mutation. In contrast, embryos from *F579Y* mothers developed normally to at least the blastoderm stage (Fig. S2 C). The partially penetrant *F579Y* hatching defect observed previously therefore appears to be due to problems arising later in development.

To visualize the mitotic apparatus of *S3372C* embryos, we additionally stained them for markers of centrosomes and microtubules. This revealed that the vast majority of spindles in the mutant embryos arrested in metaphase (Fig. 5 C). We also found that while 70% of spindles in the mutants had centrosomes, only 10% had the canonical arrangement of one centrosome at each pole (Fig. S2 D). The other mutant spindles had a range of abnormalities in centrosome arrangement, including missing or supernumerary centrosomes at one pole, or detachment of centrosomes from the spindle (Fig. 5 C and Fig. S2 D). The metaphase arrest and defects in centrosome arrangement were confirmed by time-lapse analysis of mutant embryos that had components of the spindle apparatus labeled fluorescently

(Fig. 5 D, Fig. S2 E, and Videos 1 and 2). The observation that some metaphase-arrested spindles had normal positioning and numbers of centrosomes (Fig. S2 D) suggested that altered centrosome arrangement is not primarily responsible for the metaphase arrest of *S3372C* spindles. Instead, a failure to progress to anaphase may uncouple mitotic cycles from centrosome duplication cycles, a scenario previously observed upon inhibition of mitotic regulators (McCleland and O'Farrell, 2008; Archambault and Pinson, 2010; Défachelles et al., 2015). We also observed that, while spindle poles were still focused in *S3372C* embryos, they tended to be broader than in controls (e.g., Fig. 5, C and D; and Videos 1 and 2), leading to a rounder spindle shape (Fig. S2 F). These features of spindle morphology have been observed previously when centrosome number and arrangement are impaired (Wakefield et al., 2000).

The above analysis shows that S3372C blocks anaphase onset of mitotic spindles in the early embryo. To investigate if all dynein functions are affected at this stage of development, we examined maintenance of the mitotic phase of polar bodies. Manipulating the function of cargo adaptors for dynein has

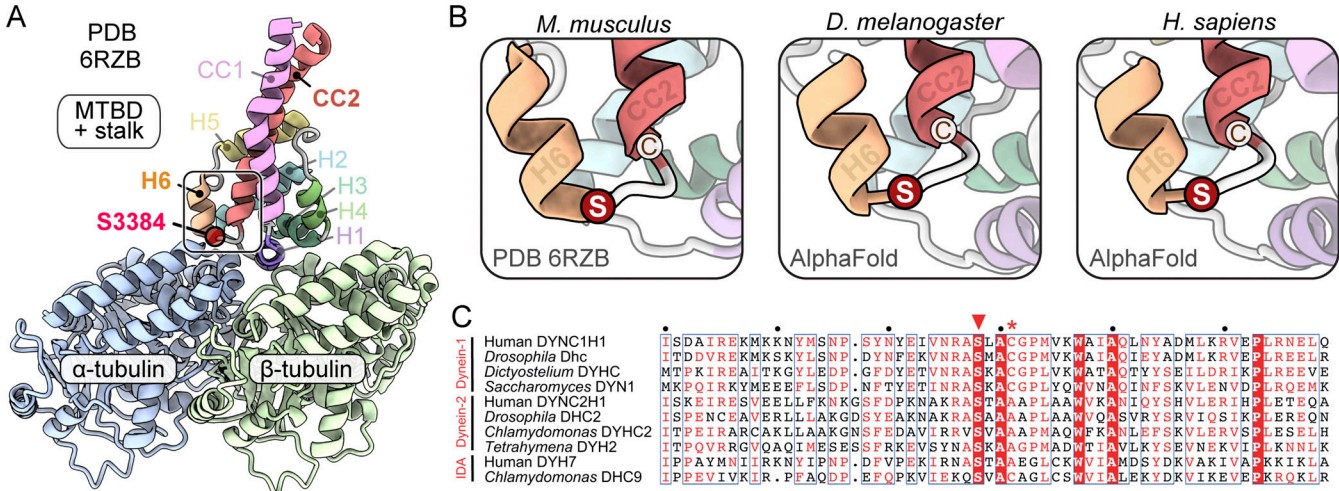

Figure 3. **S3372 has a conserved position in a loop between H6 of the MTBD and CC2 of the stalk. (A)** Overview of position of the equivalent residue to S3372 (S3384) in the cryo-EM structure of the mouse MTBD and portion of the stalk in complex with the α/β-tubulin dimer (PDB 6RZB; Lacey et al., 2019). Position of S3384 is highlighted by a red circle. **(B)** Zoom-ins of regions containing S3384 (S) in 6RZB and equivalent residues in Alphafold2-generated structures of the MTBD and stalk of *Drosophila melanogaster* Dhc and human DYNC1H1. Neighboring cysteine (C) residues (C3387 mouse; C3375 *Drosophila*; C3389 human) are also shown. **(C)** Alignment of sequences from the MTBD and CC2 of the stalk of the indicated dynein family members. White letters on red background, residues present in all sequences; red letters, residues present in ≥50% of sequences; blue boxes, regions with ≥50% conservation; red arrowhead, residues equivalent to S3372 of *Drosophila* Dhc; red asterisk, residues equivalent to C3375 of *Drosophila* Dhc. Uniprot accession numbers: human (*H. sapiens*) DYNC1H1, Q14204; *D. melanogaster* Dhc, P37276; *Dictyostelium discoideum* DYHC, P34036; *Saccharomyces cerevisiae* DYN1, P36022; human DYNC2H1, Q8NCM8; *D. melanogaster* DHC2, Q0E8P6; *Chlamydomonas reinhartii* DYHC2, Q9SMH5; *Tetrahymena thermophila* DYH2, Q5U9X1; human inner dynein arm (IDA) DYH7, Q8WXX0; *Chlamydomonas reinhartii* IDA DHC9, Q4AC22.

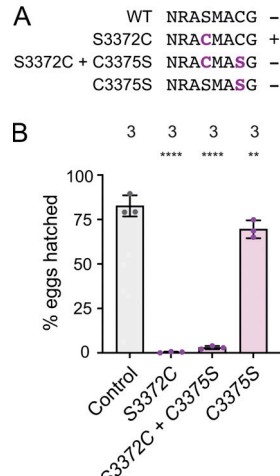

| | | |
|---|---|---|
| WT | NRASMACG | – |
| S3372C | NRA**C**MACG | + |
| S3372C + C3375S | NRA**C**MA**S**G | – |
| C3375S | NRASMA**S**G | – |

Figure 4. **The S3372C embryonic arrest is not caused by ectopic disulfide bonding in Dhc. (A)** Sequences of relevant regions of wild-type (WT), S3372C, S3372 + C3375S, and C3375S *Drosophila* Dhc. Mutated residues are shown in bold magenta. The potential for intramolecular disulfide bond formation is indicated with +. **(B)** Quantification of hatching frequency of eggs laid by mated females of the indicated genotypes. Columns show mean values per egg collection; error bars represent SD; circles are values for individual egg collections. Numbers of collections per genotype (each from an independent cross; 245–736 eggs per collection) shown above bars. The control genotype was homozygous for a wild-type *Dhc* allele recovered from the same CRISPR-Cas9 mutagenesis experiment that generated the *Dhc* mutant alleles. Evaluation of statistical significance (compared to control) was performed with a one-way ANOVA with Dunnett's multiple comparisons test: ****, P < 0.0001; **, P < 0.01.

strongly implicated the motor in this process (Défachelles et al., 2015; Vazquez-Pianzola et al., 2022). There was no defect in mitotic phase maintenance in *S3372C* embryos, as judged by normal condensation of polar body DNA and its association with phosphorylated histone H3 and the SAC component BubR1 (Fig. S3, A and B). Thus, the S3372C mutation appears to affect a subset of dynein functions in the early embryo that include—and are perhaps limited to—the transition of mitotic spindles from metaphase to anaphase.

## S3372C increases dynein accumulation in the vicinity of kinetochores

To attempt to narrow down how S3372C causes metaphase arrest, we investigated its effect on dynein's localization on the mitotic apparatus. This was done by comparing the distribution of GFP-tagged Dynein light intermediate chain (GFP-Dlic) in embryos laid by wild-type and *S3372C* homozygous females. In wild-type embryos, GFP-Dlic associated with the spindle for the duration of the mitotic cycle with a transient, weak enrichment at kinetochores during prometaphase and metaphase (Fig. 6 A and Video 3). This pattern is in keeping with previous observations of dynactin localization during embryonic mitosis (Wojcik et al., 2001). In contrast, all *S3372C* embryos had strong enrichment of GFP-Dlic in the vicinity of the kinetochores, which often extended along neighboring regions on microtubules (Fig. 6, A and B; and Video 3). The mutant embryos did not, however, have enrichment of GFP-Dlic on other parts of the

spindle apparatus, including the spindle pole or centrosome (Fig. 6 A and Video 3). To test if the abnormal accumulation of dynein at the kinetochore of mutant embryos is an indirect effect of the metaphase arrest, we examined spindle localization of GFP-Dlic in embryos of *S3372C/+* mothers, which have no apparent mitotic defects. These embryos had a modest, but statistically significant, increase in GFP-Dlic signal in the equator region relative to controls (Fig. 6 B), which was associated with the enrichment of the protein near a subset of kinetochores (Fig. 6 A). Collectively, these observations suggest that S3372C directly influences dynein's behavior at the kinetochore. Consistent with altered kinetochore-related processes in the mutant embryos, the intensity of the kinetochore marker we used in the above experiments (the Knl1-Mis12-Ndc80 [KMN] network component Spc25) was also elevated in the *S3372C* homozygous embryos (Fig. 6 A and Fig. S3 C).

## S3372C alters kinetochore-associated dynein functions

The above findings prompted us to examine kinetochore-associated roles of dynein in the mutant embryos. Functions of dynein at this site are closely linked with silencing the SAC (Gassmann, 2023), which delays anaphase until chromosomes become biorientated by inhibiting co-activators of the anaphase-promoting complex/cyclosome (APC/C) ubiquitin ligase. Dynein contributes to SAC silencing by generating end-on attachments of kinetochores with microtubule plus ends (Bader and Vaughan, 2010; Barisic and Maiato, 2015), stabilizing microtubule-kinetochore attachments by creating tension between sister kinetochores (Siller et al., 2005; Yang et al., 2007; Varma et al., 2008; Maresca and Salmon, 2010), and transport of checkpoint proteins away from kinetochores via the Rough Deal (Rod)-Zwilch-Zeste White 10 (RZZ)-Spindly adaptor complex (so called "streaming"; Howell et al., 2001; Wojcik et al., 2001; Basto et al., 2004; Griffis et al., 2007; Gassmann et al., 2010).

We first assessed microtubule–kinetochore attachment in the mutant embryos. In principle, a defect in this process could explain the increased intensity of Spc25 at the kinetochores of mutant spindles (Fig. S3 C) as, in at least some systems, the KMN network expands when not connected to microtubules (Sacristan et al., 2018). In *S3372C* embryos, the ends of microtubule bundles were closely apposed with Spc25-RFP (Fig. 7 A) and another kinetochore-associated protein, Spindly (Fig. S3 D). This observation reveals that the metaphase arrest is not due to a failure to attach microtubules to kinetochores, although we cannot rule out a subtle defect in the number or orientation of these attachments.

To assess interkinetochore tension, we measured the distance between sister kinetochores using the centroids of Spc25-RFP signals. While there was no difference in the mean inter-kinetochore distance between wild-type and *S3372C* metaphase spindles (Fig. 7 B), there was substantially more variability in this metric in the mutant (Levene test, P = 0.001). These data raise the possibility that the S3372C mutation results in variable levels of tension between kinetochore pairs.

To assess the transport of SAC components away from kinetochores in *S3372C* embryos, we monitored the behavior of GFP-tagged Rod using time-lapse imaging. GFP-Rod showed an

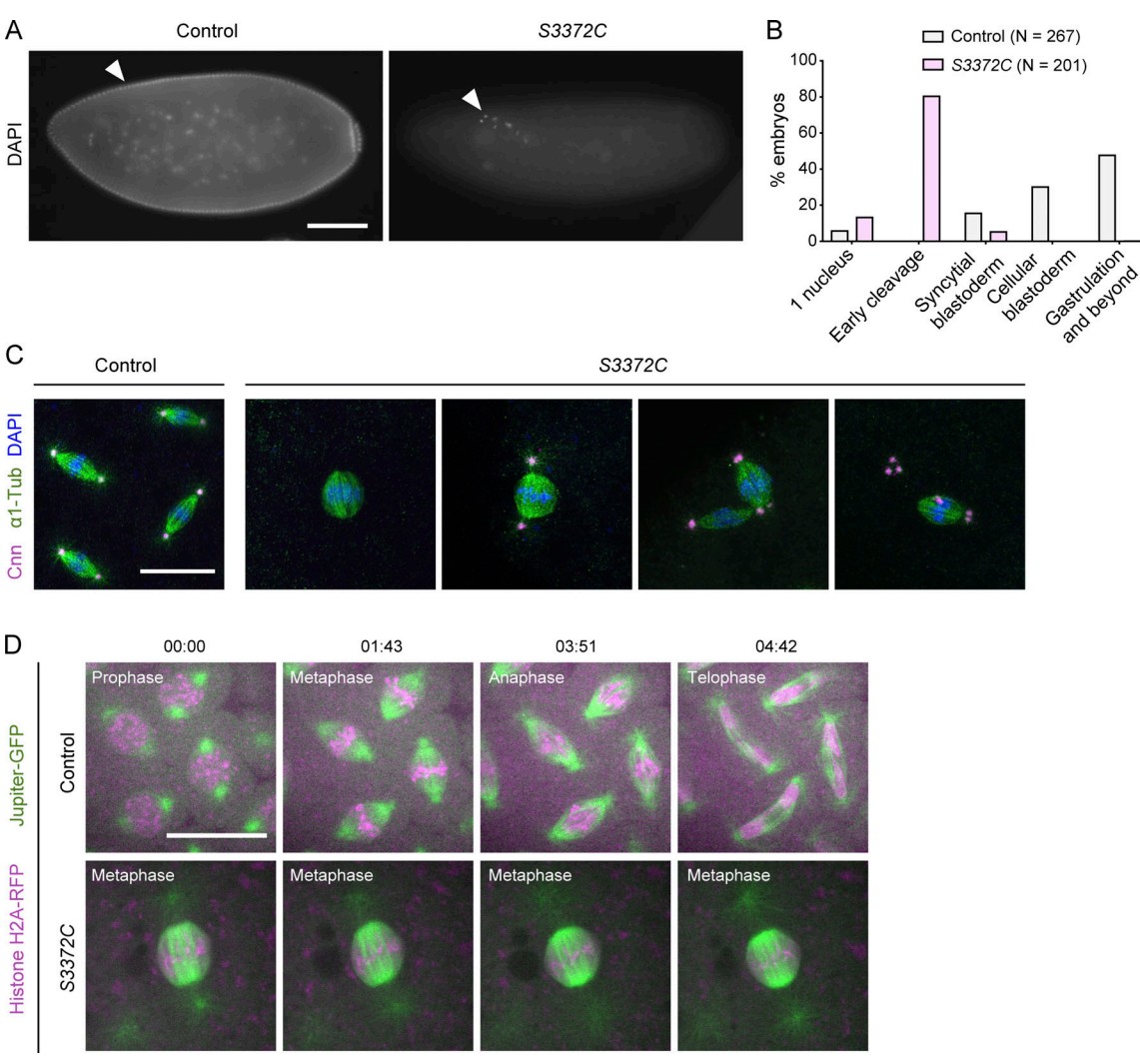

Figure 5. **S3372C causes metaphase arrest in the embryo. (A)** Example widefield images of fixed embryos from a 2- to 4-h egg collection from mated control (*yw*) and *S3372C* mothers stained with DAPI (arrowheads show examples of DNA staining). **(B)** Quantification of stages of control and *S3372C* embryos from 2- to 4-h egg collections. *N*, number of embryos scored. **(C)** Example confocal images of mitotic spindles in fixed control and *S3372C* embryos (Z-projection) stained with antibodies to Centrosomin (Cnn; centrosome marker) and α1-tubulin (α1-Tub; microtubules), as well as DAPI. **(D)** Example stills from time series (single focal plane) of control and *S3372C* embryos acquired during preblastoderm cycles. Jupiter-GFP and His2Av-mRFP label microtubules and chromatin, respectively. Timestamps are min:s. Scale bars: A, 100 µm; C and D, 20 µm.

abnormal build-up at the kinetochore in *S3372C* mutants (Fig. 7, C and D; and Fig. S3 C), consistent with the accumulation of dynein at this site (Fig. 6). While streaming of GFP-Rod away from kinetochores could be observed in the mutant embryos, it occurred at a lower velocity than in the wild-type (Fig. 7, C–E; and Video 4). These observations lend further support to the notion that S3372C interferes with functions of dynein at the kinetochore.

## S3372C can block anaphase progression in a Mad2-independent manner

It is conceivable that alterations in the number or orientation of kinetochore–microtubule attachments, the tension between sister kinetochores, or the dynamics of Rod streaming (or a combination of these scenarios) are sufficient to prevent SAC silencing in *S3372C* embryos and thereby cause metaphase arrest. Consistent with the SAC remaining functional in the mutant

embryos, Cyclin B—which is degraded once APC/C becomes active (reviewed by Murray, 1995; Huang and Raff, 1999)—remained associated with *S3372C* spindles (Fig. S3 E). Persistence of Cyclin B on the mutant spindles was not due to a failure to recruit the APC co-activator Fzy/Cdc20 (Raff et al., 2002) to the kinetochore (Fig. S3 F; Fzy/Cdc20 actually showed a greater accumulation at this site in the *S3372C* mutants than in controls [Fig. S3 C]). These data raise the possibility that S3372C blocks the metaphase–anaphase transition by preventing SAC inactivation.

To test if this is the case, we introduced into the *S3372C* background a mutation in the gene encoding the key SAC protein Mad2, which is stripped from aligned kinetochores by RZZ-Spindly-dynein to relieve inhibition of APC/C. It was previously shown that this mutation (*mad2^P*) results in no detectable Mad2 protein in larval stages (Buffin et al., 2007; Gallaud et al., 2022), and we confirmed that this is also the case in early embryos (Fig. 8 A). Consistent with this finding, it has been reported that

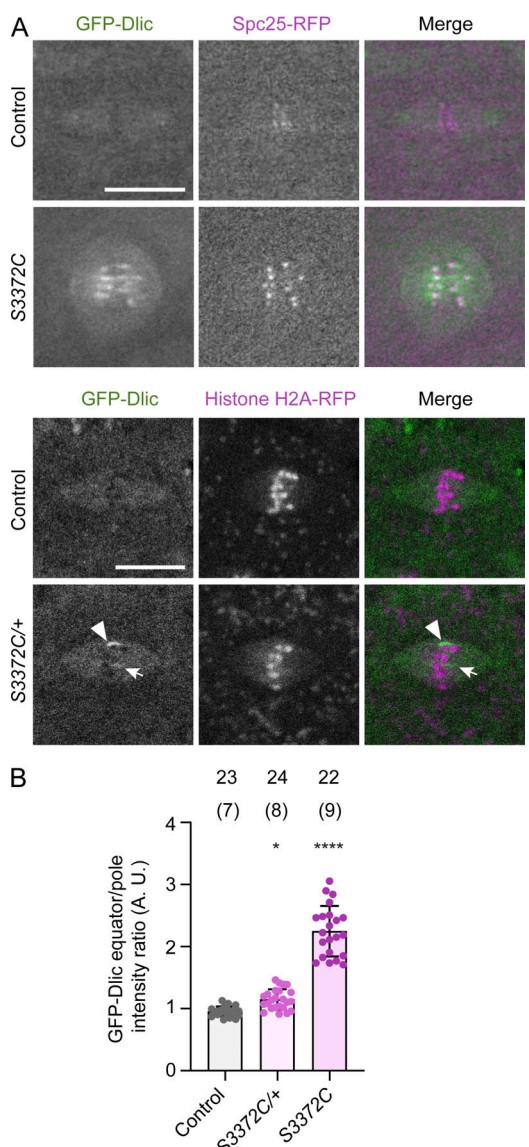

**B**

23    24    22
(7)   (8)   (9)

Figure 6. **S3372C causes ectopic accumulation of dynein in the vicinity of kinetochores. (A)** Example confocal images (Z-projections) of mitotic spindles in live control and *S3372C* (top panels) and *S3372C/+* (bottom panels) embryos expressing fluorescently labeled dynein (GFP-Dlic). Embryos additionally contain markers of kinetochores (Spc25-RFP; top) or chromatin (Histone H2A-RFP; bottom). In the bottom panels, the arrow and arrowhead show examples of accumulation of dynein in the vicinity of a subset of chromosomes at the metaphase plate in *S3372C/+* embryos. Scale bars: 10 μm. **(B)** Quantification of GFP-Dlic signal at the equator versus pole regions for the indicated genotypes. Columns show the mean of "per spindle" equator enrichment values; error bars represent SD; circles are values for individual spindles. Number of spindles analyzed shown above bars (number of embryos analyzed shown in parentheses). Statistical significance was evaluated with a one-way ANOVA with Dunnett's multiple comparisons test, comparing the per spindle values. *, P < 0.05; ****, P < 0.0001.

the SAC is inactive in *mad2^P* embryos (Défachelles et al., 2015; Yuan and O'Farrell, 2015). This was demonstrated by the ability of the mutant embryos to overcome the metaphase arrest induced by the microtubule-targeting drug colchicine and proceed to interphase. Corroborating this result, in *mad2^P* mutant embryos, we observed a highly penetrant rescue of the metaphase

arrest induced by colchicine injection (Fig. 8, B and C; 96% of nuclei in metaphase in wild-type embryos versus 4.3% in *mad2^P* embryos [P < 0.0001, Fisher's Exact test]).

As the SAC is defective in *mad2^P* mutants, the *S3372C* mitotic arrest should be rescued by the *mad2* mutation if it is solely due to a failure to alleviate SAC inhibition. However, this was not the case. The vast majority of *mad2^P S3372C* double mutant embryos still arrested at metaphase (Fig. 8 D; 81% of nuclei compared to 94.6% in *S3372C* single mutants). Like *S3372C* embryos, the double mutant embryos also exhibited a highly penetrant metaphase arrest following colchicine injection (Fig. 8 E). Thus, even in a situation in which the SAC is strongly required to prevent mitotic progression, Mad2 disruption is not sufficient to relieve the S3372C-induced metaphase arrest. The *S3372C* mitotic phenotype in untreated embryos could also not be rescued by a maternal effect mutation in the *rod* gene that impairs the association of the RZZ complex with the kinetochore and inactivates the SAC (Défachelles et al., 2015) (Fig. S3, G and H). Although these results do not rule out the MTBD mutant impairing SAC silencing, they strongly indicate that dynein has another, non-canonical role that is important for anaphase progression.

### The S-to-C mutation specifically impairs dynein behavior under load

We next attempted to elucidate the molecular basis of S3372C's selective effect on anaphase progression. We first considered the possibility that the mutation specifically impairs dynein's interaction with the α4-tubulin isotype, which is only expressed during the first 2 h of embryogenesis (Matthews et al., 1989) and co-operates with the ubiquitously expressed α1-tubulin to facilitate early nuclear divisions (Matthews et al., 1993; Mathe et al., 1998; Venkei et al., 2006). Immunostaining revealed that α4-tubulin is enriched at the spindle pole rather than the equator (Fig. S4 A). Thus, this isotype does not seem well placed to mediate the effects of S3372C dynein at the kinetochore. Moreover, although α4-tubulin and α1-tubulin have only 67% amino acid identity, the dynein-binding region of the two proteins is identical (Fig. S4 C). Thus, it seems unlikely that S3372C specifically affects binding of dynein to α4-tubulin.

We next hypothesized that S3372C alters a general aspect of dynein behavior that is particularly important for anaphase progression. We therefore set out to determine the effect of the mutation on the activity of purified dynein in vitro. To this end, the equivalent mutation (S3386C) was introduced into the full human dynein complex, for which a recombinant expression system is available (Schlager et al., 2014). As expected from its position in the MTBD, the mutation did not impair formation of the dynein complex (Fig. S5, A and B).

We first investigated the effect of S3386C on the movement of fluorescently labeled dynein along microtubules in the presence of dynactin and BICD2N using total internal reflection fluorescence (TIRF) microscopy (Fig. 9 A). Unlike the human disease-associated mutations (Hoang et al., 2017; Fig. 1 C), S3386C had no effect in this assay on the frequency of processive movements of dynein, or the distance or velocity of these events (Fig. 9, B–D; and Fig. S5, C and D). Thus, the MTBD mutation

Figure 7. **S3372C alters kinetochore-associated processes. (A)** Example confocal images of mitotic spindles (single focal plane) in live control and *S3372C* embryos that have fluorescently labeled microtubules (Jupiter-GFP) and kinetochores (Spc25-RFP). Arrowheads, examples of association of bundled microtubules with a kinetochore. **(B)** Quantification of interkinetochore distance (using centroids of Spc25-RFP signals) in live control and *S3372C* metaphase spindles. Columns show mean values; error bars represent SD; circles are values for individual kinetochore pairs. Number of kinetochore pairs analyzed (from seven control and eight *S3372C* embryos) shown above bars. **(C)** Example stills from a time series (single focal plane) showing transport of GFP-Rod particles away from the kinetochore in control and *S3372C* embryos (arrows). Timestamps are min:s. **(D)** Example kymographs showing GFP-Rod streaming (e.g., arrows). d, distance; t, time. Diagonal lines in the control label poleward movements of whole kinetochores in anaphase. **(E)** Quantification of GFP-Rod streaming velocity. Violin plots show values for individual motile Rod particles and circles show mean values per embryo. Horizontal lines show mean ± SD of values for individual motile particles. Numbers without parentheses above bars are the number of embryos, with the total number of particles given in parentheses. In B and E, statistical significance was evaluated with an unpaired, two-tailed *t* test (in E, comparisons were between mean values per embryo). *, P < 0.05; ns, not significant. Scale bars: A and C, 10 µm; D distance, 2 µm; D time, 30 s.

does not perturb translocation of isolated dynein-dynactin-activating adaptor complexes along microtubules.

The absence of cargo in the above experiments meant that dynein was operating under minimal load. To determine if the S-to-C mutation influences dynein performance under resistive loads, we quantified force production by the motor using high-resolution optical trapping (Fig. 10 A). Beads were sparsely decorated with BICD2N-GFP via a GFP antibody, incubated with dynactin and either wild-type or S3386C dynein, and brought near an immobilized microtubule using an optical trap (Belyy et al., 2016). Whereas wild-type DDB moved rapidly along the microtubule at low resistive forces (~1 pN), the S3386C version had a tendency to pause (Fig. 10 A). Following these pauses, the mutant DDB either detached from the microtubule or continued to move forward. Moreover, whereas the stall force of beads driven by wild-type DDB was 4.3 ± 0.1 pN (mean ± SEM),

consistent with previous observations (Belyy et al., 2016), the mutant stall force was 2.9 ± 0.1 pN (Fig. 10 B). Thus, the S-to-C mutation reduces dynein's peak force production by one-third. Mutant complexes in the optical trap also had increased stall times compared with the wild type (Fig. 10 C), which may be related to the motor stalling at lower resistive forces, as well as reduced velocity (Fig. 10 D). Collectively, these results show that mutant dynein has defective motility in the presence of external load.

**The S-to-C mutation modulates dynein's microtubule interaction and stalk position**

Remarkably, the effects of the S-to-C change on dynein's function in embryonic mitosis and its behavior under load stem from the substitution of a single oxygen atom for a sulfur atom within the ~90,000-atom Dhc polypeptide. To evaluate the effects of

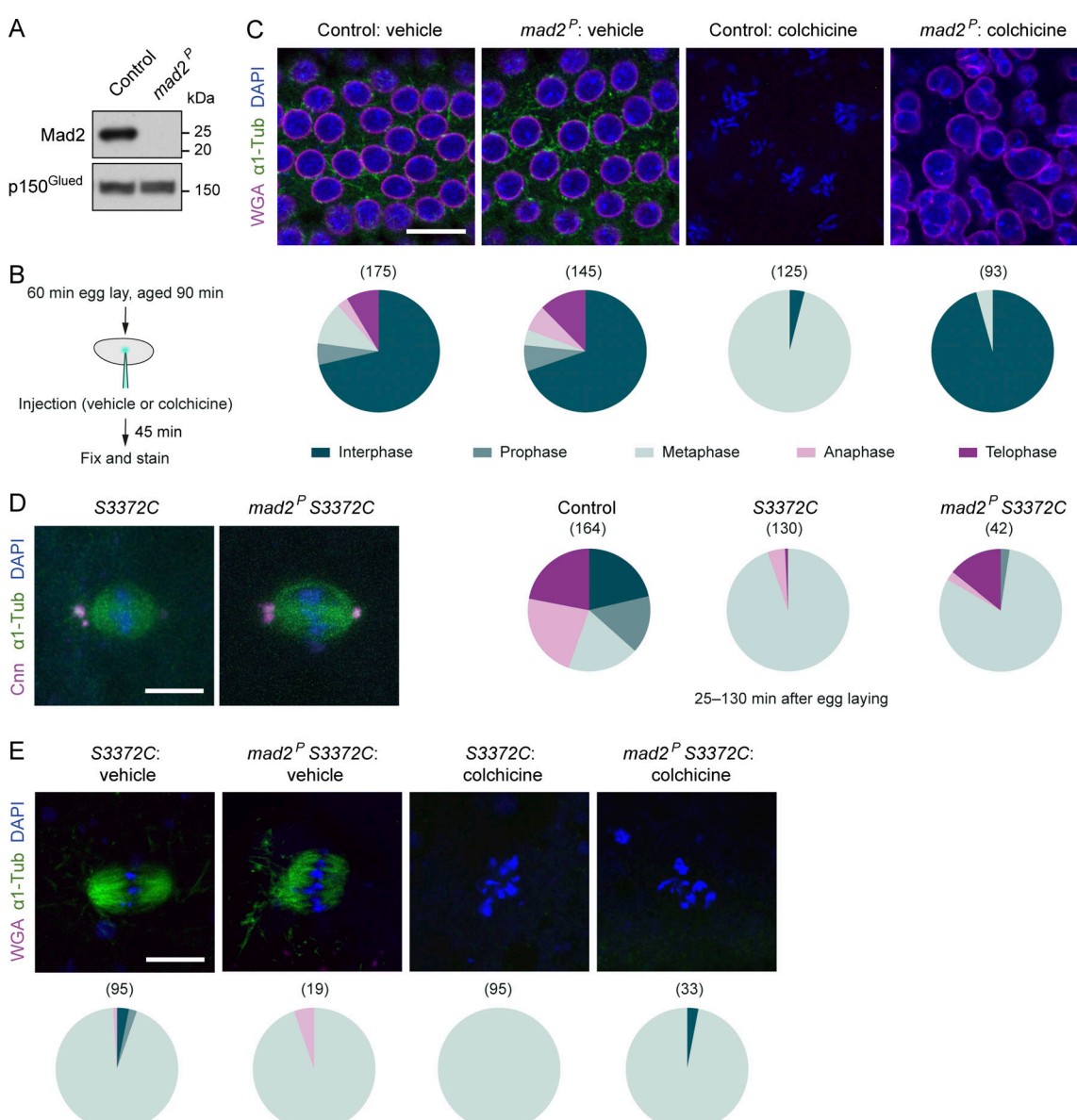

Figure 8. **Mad2-independent metaphase arrest in *S3372C* mutant embryos. (A)** Immunoblot of extracts of embryos from control (*w¹¹¹⁸*) and *mad2P* mothers (0–120-min collections), probed with antibodies to Mad2 and p150Glued (loading control). **(B)** Summary of experimental protocol for colchicine injections; vehicle is 1% DMSO. **(C)** Top, example confocal images (single focal plane) of control (*w¹¹¹⁸*) and *mad2P* embryos injected with vehicle or colchicine and stained with antibodies to α1-tubulin (α1-Tub; microtubules), as well as WGA (nucleoporins, which are absent from the nuclear envelope in metaphase and anaphase [Katsani et al., 2008]) and DAPI (DNA). Colchicine-injected wild-type embryos arrest in metaphase with condensed chromosomes whereas *mad2* embryos decondense their chromosomes and reach interphase (Yuan and O'Farrell, 2015). Bottom, incidence of mitotic stages for each condition. **(D)** Left, example confocal images (Z-projection) of metaphase spindles from fixed embryos of *S3372C* and *mad2P, S3372C* mothers stained as indicated, and, right, incidence of mitotic stages in control, *S3372C* and *mad2P, S3372C* embryos (25–130-min collection). Compared with the vehicle control in panel C, a smaller proportion of embryos are in interphase in this control as the relatively young age of embryos in the cohort enriches for rapid early division cycles. **(E)** Top: Example confocal images (single focal plane) of embryos of the indicated genotypes injected with vehicle or colchicine and stained with antibodies to α1-Tub, as well as WGA and DAPI. Bottom: Incidence of mitotic stages for each condition. In pie charts in D and E, mitotic stages are color-coded as in C. In C–E, numbers in parentheses indicate number of nuclei counted per genotype; data are from 35 control-vehicle, 29 *mad2P*-vehicle, 25 control-colchicine and 20 *mad2P*-colchicine (C), 34 control, 49 *S3372C*, and 33 *mad2P, S3372C* (D), and 34 *S3372C*-vehicle, 13 *mad2P S3372C*-vehicle, 24 *S3372C*-colchicine, and 26 *mad2P S3372C*-colchicine (E) embryos, with no more than five randomly selected nuclei analyzed per embryo. Scale bars: C, 15 μm; D and E, 10 μm. Source data are available for this figure: SourceData F8.

the mutation on the structure and dynamics of the MTBD, as well as its interaction with the microtubule, we performed all-atom molecular dynamics (MD) simulations of the human wild-type and S3386C MTBD together with a portion of the stalk-bound to the α/β-tubulin dimer (modeled on the cryo-EM structure of Lacey et al., 2019). Four sets of 900-ns-long simulations were performed for each variant in the presence of an explicit solvent.

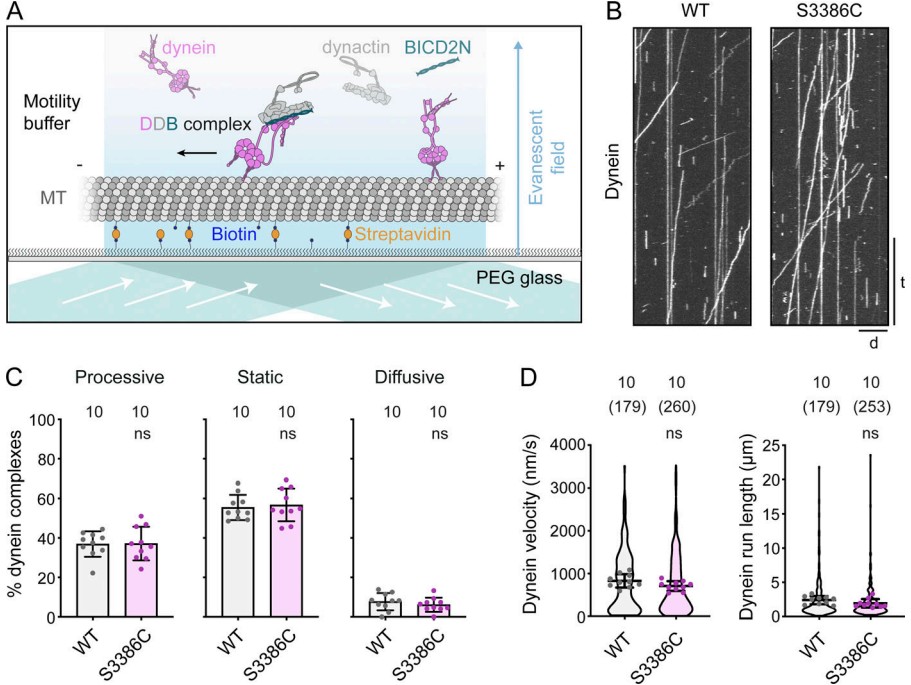

**Figure 9.** **The S-to-C mutation does not alter motility of isolated dynein-dynactin-activating adaptor complexes. (A)** Diagram of TIRF microscopy–based in vitro motility assay using an assembly of TMR-labeled dynein, dynactin, and BICD2N. In the absence of dynactin and BICD2N, dynein is autoinhibited and not capable of long-range transport. MT, microtubule; PEG, polyethylene glycol passivation surface. **(B)** Example kymographs of wild-type (WT) or S3386C tetramethylrhodamine (TMR)-dynein motility in the presence of dynactin and BICD2N. The microtubule minus end is to the left. d, distance; t, time. Scale bar: distance, 5 μm; time, 20 s. **(C and D)** Quantification of the percentage of microtubule-associated dynein complexes that exhibit processive, static, or diffusive behavior (C), and velocity and run length of the processive dynein fraction (D). In C, columns display mean values for individual microtubules; error bars are SD; circles are values for individual microtubules. Numbers above each column indicate the total number of microtubules analyzed. In D, violin plots show values for individual motile DDB complexes and circles show mean values per microtubule. Horizontal lines show mean ± SD of values for individual DDB complexes. Numbers without parentheses above bars are numbers of microtubules, with total numbers of motile DDB complexes in parentheses. In C and D, statistical significance was evaluated with an unpaired, two-tailed *t* test (in D, comparisons were between mean values per microtubule). ns, not significant. Data were summed from two experiments per condition. See Fig. S5, C and D for histograms of velocity and run length.

Analysis of the ensemble of conformations in the simulation trajectories of the wild type revealed that S3386 did not make contact with the α/β-tubulin dimer. The residue did, however, frequently form hydrogen bonds with the main chain of residues in H6 of the MTBD (V3382) and the base of CC2 in the stalk (C3389) (Fig. S5 E; and Table S1). Thus, the serine appears to stabilize the H6–stalk interaction. In simulations with the mutant dynein, the overall fold of the MTBD was not overtly different from the wild type. C3386 also did not make contact with tubulin and had a very similar frequency of interactions with V3382 and C3389 to that observed with S3386 (Table S2). However, the mutation induced new hydrophobic interactions with several neighboring residues in the MTBD (Fig. S5 E and Table S3). This was accompanied by frequent formation of a salt bridge between E3306 within H1 of the MTBD and R402 of α-tubulin, which was seldom seen in the wild-type simulations (Fig. 10, E and F; 62.4% of mutant and 8.4% of wild-type simulations). The mutation also increased the size of fluctuations of several elements in the MTBD relative to the microtubule (mean increase of 16.9% versus wild type), with those located between the end of H3 and the start of CC2 of the stalk showing the greatest effect (mean increase of 25.7% versus wild type) (Fig. 10 G). Thus, the S-to-C mutation alters the MTBD's interaction with the microtubule.

While the stalk's coiled-coil registry was not sensitive to the S-to-C change, the mutation induced either clockwise or anticlockwise tilting of the MTBD and base of the stalk around the longitudinal axis of α/β-tubulin (Fig. 10 H; peak values of 11.5° clockwise and 7.9° counterclockwise compared with the wild type). In addition, the angle of the stalk relative to the longitudinal

axis of the tubulin dimer was decreased by the mutation (Fig. 10 I; peak values of 59.9° and 46.3° for wild-type and mutant dynein, respectively). As discussed below, the changes in stalk positioning relative to the microtubule could account for the force-sensitive nature of the mutant motor.

## Discussion
### Mutational analysis of Dhc in vivo
Dynein's diverse cellular roles complicate in vivo analysis of its functions significantly. We set out to address this problem by combining mutagenesis in *Drosophila* with in vitro analysis of motor behavior. We first assessed the in vivo phenotypes of disease-associated mutations in Dhc for which in vitro effects were already defined. Abnormalities were only evident in the homozygous condition, whereas the equivalent mutations in human or mouse cause neurological disease when heterozygous (Schiavo et al., 2013). This may reflect *Drosophila*'s relatively short neurons, or its short lifespan, reducing sensitivity to partially impaired cargo transport. Nonetheless, the homozygous *Drosophila* phenotypes provide valuable information about the consequences of disease-associated *Dhc* mutations in an animal model and thereby extend previous work on these lesions in vitro (Hoang et al., 2017) and in yeast (Marzo et al., 2019). For example, our observations that disease-associated mutations can cause focal accumulation of synaptic vesicles and impaired mitochondrial transport in axons support our hypothesis that they interfere with the motility of cargo–motor complexes in neurons (Hoang et al., 2017).

The scope of our study was increased by the serendipitous recovery of novel *Dhc* alleles while generating disease-associated

Figure 10. **The S-to-C mutation alters dynein behavior under load and MTBD/stalk positioning relative to the microtubule. (A)** Left: Representative traces of beads driven by a single wild-type (WT) or S3386 DDB complex in a fixed optical trapping assay. Traces were downsampled to 250 Hz. Red arrowheads show detachment of the motor from a microtubule after the stall; green arrowheads show pausing of the mutant dynein at low forces. The inset cartoon shows DDB attachment to beads through GFP-BICD2N/GFP-antibody linkage. Right: Normalized histograms showing probability of a dynein-driven bead sampling different forces. Dwelling of the beads at 0 pN when they are not tethered to the microtubule was subtracted from the histogram. Solid curves represent a multiple Gaussian fit to calculate the mean and percent population of the two peaks. **(B)** Stall forces of DDB complexes with WT and mutant dynein (errors are SEM; N is number of stalls). **(C)** Inverse cumulative distribution (1-CDF) of motor stall times. Time constants (τ ± SEM) were calculated from a fit to a

double exponential decay (solid curves). **(D)** Force velocity plots for DDB complexes with WT and mutant dynein (errors are SEM; *N* is number of events). In A–D, data are from 11 beads in five independent experiments for WT and from 14 beads in seven independent experiments for mutant. In B and D, statistical significance was evaluated with an unpaired, two-tailed *t* test. ****, P < 0.0001; ns, not significant. In D, comparisons were to WT values at the same force. **(E)** Example snapshots of E3306(dynein) and R402(α-tubulin) positioning in MD simulations of WT and S3386C MTBD plus portion of the stalk in the presence of the α/β-tubulin dimer. **(F)** Distance distributions of E3306(dynein)-R402(α-tubulin) in WT and S3386C simulations. **(G)** Fluctuations of WT and S3386C MTBD and portion of the stalk with respect to the microtubule in MD simulations. **(H)** Left, superimposition of examples from simulations with the WT and S3386C MTBD plus a portion of stalk when viewed down the longitudinal axis of the microtubule. Gray and magenta vectors show most frequent MTBD orientation for WT and S3386C, respectively. Right, angular orientation histogram (normalized frequency) for the MTBD around the longitudinal axis of the microtubule. The WT distribution exhibited a single peak (assigned a value of 0), whereas the S3386C mutant exhibited a double-peaked distribution (peaks of fitted curves ± SD are shown). **(I)** Left: Superimposition of examples from simulations with WT and S3386C MTBD plus a portion of the stalk when viewed facing the longitudinal axis of the microtubule. The remainder of the dynein motor domain structure (PDB 7Z8F; Chaaban and Carter, 2022) was superimposed on the simulations to show the predicted position of the linker (darker shading and yellow asterisk) and AAA+ ring. Right, angular orientation histogram (normalized frequency) for the stalk relative to the longitudinal axis of the microtubule (peaks of distributions ± SD are shown). In F–I, *N* is number of simulation trajectories for each type of complex.

mutations. Our characterization of both types of mutation establishes an allelic series that should facilitate studies of dynein's involvement in multiple processes during development and in adulthood. Some of the mutations we investigated cause phenotypes—synaptic vesicle accumulations, impaired mitochondrial transport, and short bristles—that were previously seen with classical hypomorphic alleles of *Dhc* or RNAi-based knockdowns (Gepner et al., 1996; Martin et al., 1999; Melkov et al., 2016; Vagnoni et al., 2016). Thus, these mutations may impair core dynein functions to varying degrees, rather than affect discrete processes. In sharp contrast, the novel MTBD mutation, S3372C, only affects a subset of dynein functions.

S3372C does not cause visible abnormalities in adults, or defects in synaptic vesicle localization or mitochondrial transport in axons, yet homozygous mothers are infertile because mitotic spindles in their embryos arrest at metaphase. While some classical hypomorphic *Dhc* alleles, as well as the F597Y mutation, have maternal effects on embryogenesis, these only become evident at later embryonic stages and are accompanied by morphological defects in the parents (Gepner et al., 1996; Robinson et al., 1999; Wilkie and Davis, 2001; Melkov et al., 2016; this study). Thus, S3372C specifically disrupts a maternal activity of dynein that is critical for early embryogenesis. The effect of S3372C contrasts sharply with that of the neighboring R3370Q mutation, which causes zygotic mutants to arrest as larvae. These findings validate the use of mutagenesis in *Drosophila* to dissect in vivo functions of the motor complex.

### The mechanistic basis of S3372C's selective phenotype and force sensitivity

Why does S3372C have such a selective effect on Dhc activity? The finding that embryos of heterozygous *S3372C* mothers have no developmental defects suggests the mutation causes a loss, rather than a gain, of dynein function. We have shown that neither stage-specific instability of Dhc protein nor ectopic disulfide bonding is responsible for the embryo-specific phenotype. We have also provided evidence that an altered interaction of dynein with the maternal α4-tubulin isotype is not a contributing factor. While we cannot rule out the mutation blocking a phosphorylation event on S3372 that is important for anaphase onset, this seems unlikely because frequent hydrogen bonding of its sidechain with residues in H6 and CC2 is expected to restrict access to a kinase. Indeed, there is no evidence that S3372

is posttranslationally modified in previous proteome-wide studies of embryos from multiple *Drosophila* species (Hu et al., 2019) or our ongoing analysis of dynein complexes immunoprecipitated from *D. melanogaster* embryos.

We did, however, observe a striking in vitro effect of the S-to-C mutation on the behavior of dynein-dynactin-activating adaptor complexes operating with a resistive load. Whereas motility was normal in the absence of load, mutant complexes exhibited excessive pausing when exposed to a resistive force, as well as a substantially reduced peak force output and velocity. Thus, while we cannot rule out the mutation affecting another aspect of dynein function that is important for anaphase progression, the most parsimonious explanation for the restricted phenotype is that the metaphase to anaphase transition in the embryo needs dynein to work in a specific load regime that is problematic for the mutant motor complex. Consistent with this notion, we found that the slower movement of purified dynein under load was mirrored by slower transport of GFP-Rod away from kinetochores in vivo.

Our MD simulations of the human Dynein heavy chain's interaction with the microtubule allowed us to speculate how the mutation causes force sensitivity. Replacement of serine's hydroxyl group with the more hydrophobic sulfhydryl group of cysteine (Catalano et al., 2021) leads to several new hydrophobic interactions with neighboring residues, which are accompanied by increased flexibility of the MTBD, as well as repositioning of the stalk, relative to the microtubule. The MTBD and stalk in the mutant dynein tilt abnormally in both the anticlockwise and clockwise direction in the plane perpendicular to the microtubule's longitudinal axis. Increased mobility of dynein in this plane may reduce dynein's ability to overcome an opposing force. The mutation also lowers the angle between the stalk and the microtubule's longitudinal axis, a change that is expected to increase the distance the linker moves along this axis during its powerstroke. This would require more work to be done against the opposing horizontal force during the linker swing, contributing to reduced force output. An intriguing implication of these ideas for wild-type dynein mechanism is that the MTBD's ability to dictate stalk orientation plays an important role in tuning motor performance.

### Insights into dynein function in mitosis

As described in the Introduction, a prime example of the difficulties of disentangling dynein's in vivo functions is in mitosis,

where the motor has been implicated in a wide range of processes. Our discovery of a missense mutation that causes a highly penetrant metaphase arrest in the embryo without disrupting other dynein functions offers a unique tool to study the role of the motor in anaphase progression.

Although we cannot exclude an effect of the S3372C mutation on dynein's functions in other parts of the spindle apparatus, such as the poles and centrosomes, the build-up of dynein in the vicinity of the kinetochore, as well as slower transport of Rod away from this site, suggest that impaired kinetochore functions of dynein make an important contribution to the mitotic arrest. In light of our discovery of force sensitivity imparted by the S-to-C mutation, we speculate that tight bundling of microtubules at kinetochores in embryonic spindles, or other physical constraints of this environment, provide a strong opposition to motility that cannot be overcome by the mutant motor. In such a scenario, build-up of S3372C dynein on the kinetochore and associated microtubules may be the manifestation of the increased pausing of the motor complex observed in vitro under resistive loads. According to this view, the failure of S3372C to block mitosis in other cell types, including L3 neuroblasts, could reflect differences in the forces encountered by dynein near the kinetochore. Alternatively, there may be redundant mechanisms for initiating anaphase in these systems.

Dynein's known roles in licensing anaphase onset culminate in relieving inhibition of APC/C by the SAC protein Mad2. This is achieved by direct stripping of Mad2 away from kinetochores via the RZZ-Spindly adaptor complex. If the metaphase arrest in S3372C embryos is caused solely by a failure to inactivate the SAC, it should be reversed by a null mutation in the mad2 gene. Surprisingly, we found that this is not the case. While we cannot rule out the possibility that the S3372C mutation renders Mad2 dispensable for the SAC, it is difficult to conceive of how this could be the case. Moreover, we show that inactivation of the SAC through another means—a maternal effect mutation in Rod—is also not sufficient to suppress the S3372C metaphase arrest. Based on these findings, we propose that dynein has a novel, non-canonical role in licensing anaphase onset in addition to its well-known function in RZZ stripping and SAC silencing.

This novel dynein function cannot be associated with targeting the APC/C co-activator Fzy/Cdc20 to the spindle apparatus as this is not inhibited by the MTBD mutation. However, dynein could conceivably directly promote the coupling of Fzy/Cdc20 to APC/C. Alternatively, APC/C activation may be triggered indirectly by another kinetochore-associated process that depends on the motor. For example, the increased variability in interkinetochore distance in S3372C embryos may reflect abnormal tension between these structures as a result of suboptimal force generation by dynein, which might prevent APC/C activation through the complex series of signaling events that respond to chromosome biorientation (Liu et al., 2012; Krenn and Musacchio, 2015; Yuan and O'Farrell, 2015; Fujimitsu and Yamano, 2021; McVey et al., 2021). It is also possible that force production by dynein plays a physical role in separating sister chromosomes downstream of APC/C activation. Investigating these, as well as other, potential explanations for the metaphase arrest will be the focus of future studies.

## Outlook

As well as pointing to novel mechanisms controlling anaphase progression, the S-to-C mutation in the MTBD may be valuable in other contexts. As the mutated serine is widely conserved, including in dynein-2 and axonemal dyneins, its substitution with cysteine may allow load-dependent functions of dynein family members to be dissected in other contexts. Moreover, if we are correct and the mutation causes force-responsive dwelling of the motor on microtubules in vivo, the mutant complex's location may act as a reporter of subcellular regions and events where high load is experienced. This information could be useful for producing quantitative models of motor behavior in cells. We anticipate our results will also stimulate further efforts to dissect the function of cytoskeletal motors by genetic manipulation of their mechanical properties.

## Materials and methods

### Drosophila culture and existing strains

Drosophila strains were cultured on Iberian fly food (5.5% [wt/vol] glucose, 5% [wt/vol] baker's yeast, 3.5% [wt/vol] organic wheat flour, 0.75% [wt/vol] agar, 0.004% [vol/vol] propionic acid, 16.4 mM methyl-4-hydroxybenzoate [Nipagin; Sigma-Aldrich]) at 25°C and 50% ± 5% relative humidity with a 12 h-light/12 h-dark cycle. The following previously established strains and alleles were used in this study: w[1118] (Research Resource Identification Portal Number [RRID]:BDSC_5905); yw (RRID:BDSC_1495); Dhc[null] (Fumagalli et al., 2021); appl-Gal4 (Torroja et al., 1999); UAS-mito::GFP (Pilling et al., 2006); insc-Gal4>UAS-ChRFP::α-tubulin (Hartenstein et al., 2015; Gallaud et al., 2022); His2Av::mRFP (Heeger et al., 2005); Jupiter::GFP (Morin et al., 2001); Asl::mCherry (Conduit et al., 2015); Spc25::mRFP (Schittenhelm et al., 2007); GFP::Dlic (Pandey et al., 2007); Dhc::3HA (Dhc genomic rescue construct; Iyadurai et al., 1999); GFP::rod (Basto et al., 2004); GFP::fzy (Raff et al., 2002); mad2[P] (Buffin et al., 2007); and rod[Z3] (Défachelles et al., 2015). The mad2[P] S3372C and rod[Z3] S3372C strains were generated by recombination. The presence of the desired alleles in the recombinant stocks was confirmed by PCR and Sanger sequencing.

### CRISPR/Cas9-mediated knock-in of Dhc mutations

Mutations in the Drosophila Dhc gene (Dhc64C) were generated using previously established procedures (Port et al., 2014; Port and Bullock, 2016). Briefly, versions of the pCFD3 plasmid (RRID: Addgene_49410) were generated that express, under the control of the U6:3 promoter, single gRNAs that target Dhc close to the codon to be mutated (see Table S4 for sequences of oligonucleotides used for gRNA cloning). For the experiments designed to produce F579Y, R1951C, R3370Q, S3372C + C3375S, C3375S, and H3808P, transgenic strains expressing the gRNAs were established, with males of these strains crossed to nos-cas9 females (CFD2 strain [RRID:BDSC_54591]; Port et al., 2014) to produce nos-cas9/+; gRNA/+ embryos. These embryos were injected with a 500 ng/μl solution of a donor oligonucleotide (Ultramer, IDT) that codes for the desired missense mutation (see Table S4 for donor sequences). In cases where the mutation would not disrupt targeting by the gRNA, synonymous changes that prevent

recutting of the modified allele by the Cas9/gRNA complex were also introduced (Table S4). For the generation of the other mutations, the above procedures were replicated, except the gRNAs were introduced by co-injection of the pCFD3-gRNA plasmid (100 ng/μl solution) with the donor oligonucleotide into *nos-cas9* embryos.

Flies containing the desired mutation were identified by sequencing PCR products containing the target region, as described (Port and Bullock, 2016), followed by establishment of stocks using balancer chromosomes. Other in-frame mutations in *Dhc*, which resulted from imprecise repair of Cas9-mediated DNA cleavage, were also retained for phenotypic analysis. With the exception of *S3372C* and *R3370Q*, all mutations were isogenized by backcrossing to the $w^{1118}$ strain for 6–10 generations. In addition, we balanced a chromosome in which *Dhc* had not been mutated during the CRISPR process. This "CRISPR WT" chromosome was used as the control genotype for a subset of experiments.

### Assaying lethality and fertility

To assess lethality, flies heterozygous for the *Dhc* missense mutations (or the CRISPR wild-type chromosome) and balanced with TM6B were crossed together or with *Dhc^null^*/TM6B flies. Absence of non-TM6B adult offspring (based on the *Hu* marker) indicated developmental arrest of homozygotes or trans-heterozygotes. To assess the stage of development arrest, crosses were performed with stocks balanced with the *TM3 [actin5C>GFP]* fluorescent balancer. In these experiments, cohorts of homozygous (GFP-negative) embryos were transferred to plates containing apple juice agar (1.66% [wt/vol] sucrose, 3.33% [wt/vol] agar, 33.33% [vol/vol] apple juice, and 10.8 mM methyl-4-hydroxybenzoate), and the number of animals that survived until early L2, late L2, L3, and pupal stages scored through regular inspections. Genotypes that did not arrest before pupal stages but did not reach adulthood were classed as pupal lethal and this was confirmed in independent crosses with the TM6B balancer, which has the *Tb* marker that is visible at pupal stages.

Fertility of *Dhc* mutant and control females was assessed in crosses to wild-type ($w^{1118}$) males 5 days after eclosion of the females. After 24 h, crosses were transferred to egg-laying cages capped with apple juice agar plates, and the proportion of total embryos that had hatched 28–48 h after egg laying was recorded in multiple technical replicates over the next 3 days.

### Immunostaining

L3 wandering larvae were dissected and fixed in 4% formaldehyde as described (Hurd and Saxton, 1996). With the exception of those that were injected with colchicine or vehicle, embryos were collected, dechorionated, fixed at the interface of 4% formaldehyde and n-heptane, and devitellinized by replacement of the formaldehyde layer with methanol (Port et al., 2014). Larval preparations were washed in phosphate-buffered saline (PBS)/0.1% Tween (PBT) and blocked in 20% Western Blocking Buffer (Sigma-Aldrich) in PBT, whereas embryos were washed in PBS/0.1% Triton X-100 (PBST) and blocked in 20% Western Blocking Buffer in PBST. Details of primary and secondary antibodies, including working dilutions and available RRID codes, are provided in Tables S5 and S6. Where indicated, 5 μg/ml of Alexa555-labeled wheat germ agglutinin (WGA) (Thermo Fisher Scientific) was included with the secondary antibody to facilitate identification of mitotic stages (Katsani et al., 2008). Samples were mounted in Vectashield containing DAPI (Vector Laboratories). Segmental nerves were imaged with a Zeiss 780 laser-scanning confocal microscope using a 40×/1.3 NA oil-immersion objective. Embryos were typically imaged with a Zeiss 710 or 780 laser-scanning confocal microscope using 40×/1.3 NA or 63×/1.4 NA oil-immersion objectives, the exception being the use of a Nikon Ti2 widefield microscope equipped with a 20×/0.75 NA air objective and an Orca Flash 4.0 (Hamamatsu) camera for initial documentation of the stage of embryonic arrest of *S3372C* mutants.

### Injection of embryos with colchicine

Colchicine injection was performed essentially as described by Yuan and O'Farrell (2015). Colchicine (Stratech Scientific) was dissolved in DMSO to make a stock solution of 100 mg/ml, which was diluted on the day of the experiment to 1 mg/ml with $dH_2O$. Embryos from a 60-min egg lay were aged for a further 60 min before being dechorionated and lined up in rows on an apple juice plate with a fine paintbrush, transferred to a 9 × 35-mm coverslip coated with heptane glue, and desiccated for 4–5 min with silica gel (Sigma-Aldrich). The coverslip was then adhered to a microscope slide via a drop of water and embryos were covered in a thin layer of Voltalef 10S halocarbon oil (VWR). 1.5–2.5 h after egg laying, embryos were injected with ~100 pl of either 1 mg/ml colchicine or 1% DMSO (vehicle control) using a laser-pulled borosilicate glass needle mounted on a Narishige micromanipulator that was attached to a Nikon Eclipse TS100 microscope. Embryos were aged for 45 min after injection of the last embryo on the coverslip (~53 min after injection of the first embryo) before fixing in heptane saturated with 37% formaldehyde for 20 min and transferring to double-sided tape, followed by the addition of PBT. The vitelline membrane was then manually removed with a hypodermic syringe needle. Embryos were subsequently immunostained, mounted, and imaged as described above.

### Assessing mitochondrial transport in the wing nerve

Fly wing mounting, imaging, and analysis were performed as described previously (Vagnoni and Bullock, 2016; Vagnoni et al., 2016). Briefly, $CO_2$-anesthetized male or female flies that had eclosed 2 days earlier were mounted on double-sided sticky tape and wings coated in Voltalef 10S halocarbon oil. Movements of mitochondria (labeled with mito-GFP under the control of the neuronal driver *appl-GAL4*) in the arch region of the wing nerve were visualized with an UltraVIEW ERS spinning disk system (PerkinElmer) equipped with an Orca ER Charge-Coupled Device (CCD) camera (Hamamatsu) using a 60×/1.2 NA oil-immersion objective on an IX71 microscope (Olympus). Images were acquired at 22–23°C. A single focal plane was imaged with an acquisition rate of 0.5 frames/s for 3 min. Images were processed and analyzed with Fiji (Schneider et al., 2012; RRID: SCR_002285). The genotypes of the image series were hidden

from the experimenter using the BlindAnalysis macro (Stephen Royle, University of Warwick, Coventry, UK), followed by image straightening, stabilization, and manual tracking of motile mitochondria in MTrackJ, as described previously (Vagnoni and Bullock, 2016; Vagnoni et al., 2016). We additionally quantified the total number of mitochondria per 50 µm of the wing nerve and the proportion of these that underwent transport (i.e., had at least one continuous bout of net motion of ≥ 2 µm [a "run"]).

### Quantification of spindle circularity and protein localization on the spindle

Spindle circularity was calculated in Fiji after manually drawing an eclipse surrounding the spindle-associated microtubule signal. Quantification of localization of spindle-associated proteins was performed on randomly selected metaphase spindles using Fiji. GFP-Dlic quantification was performed using embryos that additionally expressed Histone H2A-RFP. These data were captured as single Z-planes corresponding to the maximum width of the spindle. On one half of each spindle, three to four squares of 10 × 10 pixels each were placed adjacent to the His2Av signal, corresponding to the region expected to contain kinetochores and microtubule plus ends. Mean gray values from each square were averaged between the set of squares, and this value was divided by the mean gray value of a single 10 × 10-pixel square drawn on the pole region of the same half spindle to produce an "equator/pole" value for each spindle. A similar procedure was used to quantify intensity values for Spc25-RFP, Rod, and Fzy/Cdc20 puncta, except the squares were drawn around regions of focal accumulation of each protein in the equator region in the absence of other markers of kinetochores or DNA. Gray values were then subjected to background subtraction using a single 10 × 10-pixel square in the cytoplasmic region (which had an equivalent intensity to the pole region in these stainings). For quantification of localization of α1-tubulin and α4-tubulin, a Z-plane that was parallel to the long axis of the spindle was selected based on the presence of both centrosomes (marked with Cnn antibodies). Two 10 × 10-pixel squares, corresponding to the pole region and a microtubule plus end-containing region at the center of the equator (adjacent to the DAPI signal), were then drawn on one half spindle. A third square of the same area was drawn outside the spindle region for background subtraction. Background subtracted mean gray values for α1-tubulin and α4-tubulin were obtained for each square and used to calculate a "pole/equator" intensity value for each protein per spindle.

### Analysis of *Drosophila* neuroblast divisions

Brains of L3 larvae (~120 h after egg laying) were dissected in Schneider's media (Sigma-Aldrich) containing 10% fetal calf serum (Thermo Fisher Scientific) and transferred to 50-µl wells of Angiogenesis µ-Slides (Ibidi) for live imaging. Mutant and control brains were imaged in parallel at 25°C. Z-series with a height of 20 slices and 1-µm spacing were acquired every 30 s using a spinning disk system consisting of a Leica DMi8 microscope equipped with a 63×/1.4 NA oil-immersion objective, a CSU-X1 spinning disk unit (Yokogawa) and an Evolve

Electron-Multiplying CCD (EMCCD) camera (Photometrics). The microscope was controlled by Inscoper Imaging Suite software (Inscoper). Images were processed with Fiji.

### SDS-PAGE and immunoblotting

*Drosophila* embryo extracts were generated for immunoblotting as described (McClintock et al., 2018). SDS-PAGE and protein transfer to Immobilon-P polyvinylidene difluoride membrane (Merck Millipore) were performed with the NuPAGE Novex and XCell II Blot Module systems (Thermo Fisher Scientific) according to the manufacturer's instructions. Membranes were blocked either with 5% dried skimmed milk powder (Marvel)/PBS and washed in PBS/0.05% Tween-20 (for probing Dhc or α-tubulin) or 10% dried skimmed milk powder in Tris-buffered saline (TBS)/0.1% Tween (TBST) and washed in TBST (for probing Mad2 and p150$^{Glued}$). Details of primary and secondary antibodies, including working dilutions and available RRID codes, are provided in Tables S5 and S6. Secondary antibodies were detected with the ECL Prime system (Cytiva Amersham) using Super RX-N x-ray film (Fujifilm). Films were digitized with an Epson V850 Pro scanner.

### Sequence and structure analysis

Alignments of Dynein heavy chain and α-tubulin sequences were produced with ESPript (https://espript.ibcp.fr/ESPript/ESPript/ [RRID:SCR_006587]; Robert and Gouet, 2014). Visualization and analysis of experimentally determined MTBD structures were performed with PyMOL (version 2.5.1; Schrödinger [RRID:SCR_000305]) and UCSF ChimeraX 1.2.5 (https://www.cgl.ucsf.edu/chimerax/ [RRID:SCR_015872]). The cryo-EM structure of the mouse dynein-1 MTBD and the portion of the stalk bound to the microtubule (PDB 6RZB; Lacey et al., 2019) were generated from a "cysteine-light" dynein in which C3389 was mutated to alanine. The alanine residue was therefore substituted for the native cysteine in Fig. 3 B.

Structure predictions were performed using a local installation of ColabFold 1.2.0 (Mirdita et al., 2022), running MMseqs2 (RRID:SCR_022962; Mirdita et al., 2019) for homology searches, and Alphafold2 (Jumper et al., 2021) for predictions with three recycles. For the MTBD and stalk regions of different dyneins, predictions were performed with the following amino acid sequences: *D. melanogaster* Dhc (Uniprot ID P37276) 3212-3451, *Homo sapiens* DYNC1H1 (Uniprot ID Q14204) 3227-3465, *H. sapiens* DYNC1H2 (Uniprot ID Q8NCM8) 2922-3162, and *H. sapiens* DYH7 (Uniprot ID Q8WXX0) 2614-2866. The top-ranking models were visualized in ChimeraX 1.2.5., with secondary structures colored based on their structural homology to PDB 6RZB.

### Live imaging of mitosis in *Drosophila* embryos

Dechorionated transgenic embryos with fluorescently marked centrosomes, histones, microtubules, kinetochores, Dlic, Rod, or Fzy/Cdc20 were filmed under Voltalef 10S halocarbon oil using the Ultraview ERS spinning disk system described above with a 60×/1.2 NA water-immersion objective or using a Zeiss 710 laser scanning confocal with a 63×/1.4 NA oil-immersion objective. Images were acquired at 22–23°C. A single focal plane was imaged with an acquisition rate of 0.2 frames/s for up to 15 min

to capture complete nuclear division cycles. To capture Rod streaming, an acquisition rate of 0.5 frames/s was used. Exposure times were typically maintained at 300 or 500 ms for microtubules, histones, Dlic, and Rod and at 1,000 or 1,500 ms for kinetochores and centrosomes. Images were processed with Fiji, including analysis of streaming of Rod signals with kymographs.

### Introduction of a C3386 codon into human DYNC1H1

Phusion High Fidelity Master Mix with GC buffer (New England Biolabs) was used for site-directed mutagenesis of pDyn1, which contains human DYNC1H1 sequences (NCBI accession number NM_001376.4) that are codon-optimized for Sf9 insect cell expression. The presence of the mutation encoding the S3386C substitution, as well as the absence of other non-synonymous mutations, was confirmed using Sanger sequencing (Genewiz) and whole-plasmid next-generation sequencing (Massachusetts General Hospital Center for Computational and Integrative Biology DNA Core).

### Production of dynein, dynactin, and BICD2N

Wild-type and S3386C-containing human dynein complexes were produced as described (Schlager et al., 2014; Hoang et al., 2017). Briefly, they were expressed recombinantly in Sf9 cells by transposition into the baculovirus genome of sequences encoding wild-type or S3386 DYNC1H1 (tagged with SNAP for fluorescent labeling, and ZZ [a synthetic Fc region-binding domain of protein A] for protein purification), as well as the other dynein subunits (DYNC1I2 [DIC2; AF134477], DYNC1LI2 [DLIC2; NM_006141.2], DYNLT1 [Tctex1; NM_006519.2], DYNLL1 [LC8; NM_003746.2], and DYNLRB1 [Robl1; NM_014183.3]). Dynein complexes were captured from Sf9 cell lysates using IgG Sepharose 6 FastFlow beads (GE Healthcare), labeled with SNAP-Cell-TMR-Star, and eluted with Tobacco Etch Virus protease by virtue of a cleavage site between the ZZ and SNAP tags. Dynein complexes were further purified by fast protein liquid chromatography (TSKgel G4000SWxl column [TOSOH Bioscience]) and concentrated by centrifugation through an Amicon Ultra-4 Centrifugal Filter Device (Merck Millipore). Native dynactin was purified from fresh pig brains as previously described (Schlager et al., 2014; Urnavicius et al., 2015) using a series of chromatography steps (XK 50/30 cationic exchange column [GE Healthcare] packed with SP-Sepharose Fast Flow [GE Healthcare], MonoQ HR 16/10 anionic exchange column [GE Healthcare], and TSKgel G4000SWxl column). SNAP-BICD2N and BICD2N-GFP (both containing residues 1-400 of mouse BICD2) were expressed and purified using the Sf9 baculovirus system, as previously described (Schlager et al., 2014; Belyy et al., 2016).

### Assessing motility of isolated DDB complexes

Motility assays were performed using established protocols (Schlager et al., 2014; Hoang et al., 2017; McClintock et al., 2018). Briefly, biotin- and HiLyte-488–labeled pig brain microtubules were polymerized in vitro using commercial tubulin sources (Cytoskeleton Inc.) and stabilized with taxol (Sigma-Aldrich) and guanosine-5'-[(α,β)-methyleno]triphosphate (Jena Bioscience) before being adhered to a streptavidin-coated coverslip within the flow chamber. Approximately 30 min before

imaging, a DDB assembly mix was prepared on ice by combining 100 nM TMR-labeled human dynein (wild type or S3386C), 200 nM pig dynactin, and 1 µM SNAP-BICD2N in motility buffer (30 mM HEPES pH 7.3, 5 mM MgSO$_4$, 1 mM EGTA pH 7.3, 1 mM DTT). A "motility mix" comprising 2.5% (vol/vol) DDB assembly mix, 2.5 mM MgATP (Sigma-Aldrich), and 10% (vol/vol) oxygen scavenging system (1.25 µM glucose oxidase [Sigma-Aldrich], 140 nM catalase [Sigma-Aldrich], 71 mM β-mercaptoethanol, and 25 mM D-glucose) in motility buffer supplemented with 50 mM KCl, 1 mg/ml α-casein, and 20 µM taxol was introduced into the microtubule-containing flow chambers.

TIRF imaging was performed at room temperature (23 ± 1°C) using a Nikon TIRF system equipped with a back-illuminated EMCCD camera iXon$^{EM+}$ DU-897E (Andor) and a 100×/1.49 NA oil-immersion APO TIRF objective, and controlled with µManager software (Edelstein et al., 2010). Three movies were acquired per chamber, with acquisition for 500 frames at the maximum achievable frame rate (~2 frames/s) and 100-ms exposure per frame. Pixel size for each image series was 105 × 105 nm.

Analysis of dynein behavior was performed manually using kymographs in Fiji, as described (Hoang et al., 2017). Kymographs were generated from five microtubules per movie, followed by tracking of dynein movements with the Multipoint tool. Microtubule binding events were only counted if they lasted for ≥1.5 s (3 pixels in the y-axis) and processive events were only counted when travel distance was ≥500 nm (5 pixels in the x-axis). Because DDB complexes could change speed during runs, mean velocity was calculated from individual velocity segments, as described previously (Schlager et al., 2014). The identities of kymographs were blinded from the user before analysis using the BlindAnalysis macro.

### Optical trap-based force measurements

Optical trapping of DDB complexes was performed as described previously (Belyy et al., 2016). Briefly, DDB complexes were assembled with 1 µl of 0.38 mg/ml (mutant) or 1 µl 1.06 mg/ml (wild-type) dynein, 1 µl of 2.25 mg/ml dynactin, and 1 µl of 1.04 mg/ml BICD2N-GFP (Belyy et al., 2016) for 10 min at 4°C. The protein mixture was then added to 800-nm diameter carboxylated polystyrene beads coated with a rabbit polyclonal anti-GFP antibody (MMS-118P; BioLegend) and incubated for 10 min. Flow chambers were decorated with sea urchin axonemes in MB buffer (30 mM HEPES pH 7.0, 5 mM MgSO$_4$, 1 mM EGTA, 10% [vol/vol] glycerol). The motor-bead mixture was then introduced to the chamber in imaging buffer (MB supplemented with oxygen scavenging system and 2 mM MgATP). The bead concentration was held constant for all measurements. To ensure that more than ~95% of beads were driven by single dynein motors, the protein mixture was diluted before incubating with beads such that a maximum of 30% of beads exhibited activity when brought into contact with an axoneme (Belyy et al., 2016).

Optical trapping experiments were performed on a custom-built optical trap microscope set-up. DDB-bound beads were trapped with a 2 W 1,064-nm laser beam (Coherent) that was focused on the image plane using a 100×/1.49 NA oil-immersion

objective (Nikon). Axonemes were located by brightfield imaging and moved to the center of the field-of-view with a locking XY stage (M-687; Physik Instrumente). Trapped beads were lowered to the axoneme surface with a piezo flexure objective scanner (P-721 PIFOC; Physik Instrumente). The position of the bead relative to the trap center was monitored by imaging the back-focal plane of a 1.4 NA oil-immersion condenser (Nikon) on a position-sensitive detector (First Sensor). A pair of perpendicular acousto-optical deflectors (AA Opto-Electronic) was used to control beam steering. To calibrate the detector response, a trapped bead was rapidly raster-scanned by the acousto-optical deflector, and trap stiffness was derived from the Lorentzian fit to the power spectrum of the trapped bead. The laser power was adjusted with a half-wave plate on a motorized rotary mount and maintained at a constant value of 80 mW, corresponding to a spring constant of 0.05–0.06 pN/nm. Before data collection for each sample, the spring constant of a trapped bead was recorded.

In the case of DDB complexes formed with wild-type dynein, custom MATLAB software was used to extract stall forces and stall times from raw traces (following down-sampling from 5,000–250 Hz). Stall events were defined as a stationary period of a bead at forces >2.5 pN and durations >100 ms, which were followed by snapping the back of the bead to the trap center. Stall force was defined as the mean force during the last 20% of the stall event. The stall time was defined as the interval that the bead spent at a force of at least 80% of the stall force. All stall events were plotted and manually reviewed to confirm the accuracy of the computed values. Due to the distinct stall behavior of S3386C dynein, stall forces and stall times of this complex were calculated manually by reviewing each trace for stationary periods of a bead lasting >150 ms at forces >1 pN. The probability of dynein-driven beads sampling different forces under the trap was calculated by combining all the trajectories of wild-type or mutant dynein, plotting the normalized histogram of dynein at binned forces, and fitting the histogram to three Gaussians in MATLAB. The peak at near 0 pN represents the time the beads spend unbound from the microtubule. In Fig. 10 A, this peak was subtracted from the histograms to show the behavior of the beads when they are driven by dynein along the microtubule. To produce force–velocity plots, all traces were recorded at 5 kHz with those containing stalls concatenated. Custom MATLAB software was used to apply a median filter with a window size of 200 points. Subsequently, all events contained within 1 pN wide bins centered on 2, 3, 4, and 5 pN were identified. By dividing the change in force in each event by the typical spring constant used to measure the stall force (0.06 pN/nm), we were able to estimate the average velocity. To prevent the incorporation of detachment events, where dynein is not engaged with the microtubule, only non-negative average velocities were recorded. Force–velocity plots were then generated using the mean velocity of all trajectories within each bin.

## MD simulations

The mouse cytoplasmic dynein-1 MTBD and partial stalk structure in complex with α/β-tubulin (PDB 6RZB; Lacey et al., 2019)

were used to generate a model of the equivalent wild-type human cytoplasmic dynein-1 sequences bound to the tubulin heterodimer. Sequence alignment of the mouse and human MTBDs revealed no sequence divergence. However, two C-to-A mutations (at residues 3,323 and 3,387) had been introduced in the mouse MTBD used for cryo-EM and these were corrected in the human dynein model using the Mutator plugin of VMD (RRID: SCR_001820; Humphrey et al., 1996). We used the same plugin to subsequently introduce the S3386C mutation into the model.

α-Tubulin residues P37 to D47 are missing in PDB 6RZB, presumably because they are unstructured. A peptide was therefore constructed via the Molefacture plugin of VMD that has P37-D47 as an unstructured region and the flanking regions E23-M36 and S48-E55 as an α-helix and β-sheet, respectively. The constructed peptide was solvated in a water box and 150 mM KCl for MD simulations. The peptide system was minimized for 10,000 steps, followed by 2 ns of equilibration. After equilibration, the peptide was fitted into the remaining tubulin structure via targeted MD simulations (Schlitter et al., 1994) in which the flanking structured regions were slowly (10 ns) pushed into their crystal coordinates. Subsequently, the P37–D47 stretch was incorporated into the remaining tubulin structure using VMD's merge extension.

MTBD–tubulin systems were aligned with the longitudinal axis of the α/β-tubulin dimer in the z direction. Each system was then solvated in a TIP3P water box with 35 Å cushions in each x-direction (70 Å water cushion in total) to provide enough space for stalk movements; 15 Å cushions were applied in all other directions. Systems were neutralized and ion concentrations were set to 150 mM KCl. The size of the solvated system was ~230,000 atoms. Simulations were performed at 310 K and 1 atm pressure with a time step of 2 fs. Langevin dynamics with a damping coefficient of 1/ps were used to keep the temperature constant. The pressure was kept at 1 atm using the Langevin Nosé-Hoover method with a 100-fs oscillation period and a 50-fs damping time scale. Long-range electrostatic interactions were calculated with the particle-mesh Ewald method with a cut-off of 12 Å for van der Waals interactions. Prior to running MD simulations, 10,000 steps of minimization followed by 2 ns of equilibration were performed by keeping the proteins fixed. Subsequently, 10,000 steps of minimization were applied without any constraints on dynein and on tubulin, followed by 4 ns of equilibration with 1 kcal/mol/Å² harmonic constraints on the $C_\alpha$. Starting from the equilibrated systems, 900-ns-long production runs were initiated with constraints on R2-H28, F53-E55, H61-D69, P72-R79, L92-S94, F103-N128, G131-N139, G144-Y161, S165-I171, V182-L195, V202-L217, Y224-S241, I252-L259, L269-A273, V288-F296, Y312-G321, P325-T337, F352-V353, R373-S381, R373-S381, and I384-H393 residues of α-tubulin, and R2-H28, D41-T55, R58-D67, G71-A78, G82-F92, W101-G126, L130-S138, S145-L151, I163-V169, V180-S188, E198-M200, N204-T214, Y222-R241, G244-L246, L250-N256, F265-A271, V286-M299, Y310-G319, T323-N337, I349-C354, M363-S371, A373-Q391, R384-A393, and L395-G400 residues of β-tubulin to prevent structural deformation of the α/β-tubulin dimer due to the rest of the microtubule being absent. All MD simulations were performed in NAMD v2.14 (RRID:SCR_014894; Phillips et al., 2020) using the

CHARMM36 all-atom additive protein force field (Best et al., 2012).

To calculate MTBD angles in the plane perpendicular to the longitudinal microtubule axis, principal axes (PA) of microtubules were obtained using the VMD Orient tool. All PAs started from the origin located at the center of the mass of tubulin and were referenced to the MTBD-tubulin structure. PA1 corresponds to the longitudinal axis of the protofilament. PA2 is the radial axis pointing toward the center of the A3295 and W3395 $C_\alpha$ atoms of dynein. PA3 is the tangential axis perpendicular to both PA1 and PA2. The MTBD vector, which indicates the combined MTBD and stalk base orientation with respect to the tubulin, was defined as the vector pointing from the center of dynein's A3295 and W3395 $C_\alpha$ atoms toward the center of its A3288 Cα and Y3402 Cα atoms. To calculate the MTBD angle, the MTBD vector was projected onto the plane created by PA1 and PA2, and the angle between the projected vector and PA2 was evaluated.

We calculated the stalk angles relative to the microtubule longitudinal axis using the same procedure employed for the MTBD angles but with a different selection of dynein atoms. Specifically, we superimposed the full-length monomeric dynein structure (PBD 7Z8F; Chaaban and Carter, 2022) with the MD conformations by using the Cα atoms of the stalk's base, including residues A3295 to W3395 and A3288 Cα to Y3402, for alignment. We defined the stalk vector as the vector that originates from the center of dynein's A3295 and W3395 Cα atoms as sampled in the MD simulation and extends toward the center of its R3191 Cα and S3501 Cα atoms within the superimposed structure.

The criteria for determining interactions were as follows: salt bridge formations were detected using a maximum cut-off distance of 4 Å between the basic nitrogen and acidic oxygens (Barlow and Thornton, 1983); for hydrophobic interactions, a cutoff distance of 8 Å between side chain carbon atoms was used (Manavalan and Ponnuswamy, 1977; Stavrakoudis et al., 2009; Stock et al., 2015); and to detect hydrogen bond formation, a maximum 3.5 Å distance between hydrogen bond donor and acceptor and a 30° angle between the hydrogen atom, the donor heavy atom, and the acceptor heavy atom was used (Durrant and McCammon, 2011).

### Statistics and data plotting

Statistical analysis and data plotting were performed with Prism 7.0b (GraphPad; RRID:SCR_002798). Appropriate statistical tests were selected based on confirmed or assumed data distributions (Gaussian or non-Gaussian), variance (equal or unequal), sample size, and number of comparisons (pairwise or multiple). Data normality was confirmed with Anderson-Darling, D'Agostino-Pearson, Shapiro-Wilk, and Kolmogorov-Smirnov normality tests. Details of statistical tests applied for each dataset are provided in the relevant figure legends. Violin plots were used instead of bar charts when each group within a dataset had at least 30 values.

### Online supplemental material

Fig. S1 shows supplementary phenotypic analysis of *Dhc* mutations. Fig. S2 shows supplementary data on structural and phenotypic analysis of the S-to-C mutation. Fig. S3 shows supplementary data from analysis of the *S3372C* mitotic phenotype. Fig. S4 shows investigations of potential role of α4-tubulin in the *S3372C* phenotype. Fig. S5 shows supplementary data from in vitro and in silico analysis of S-to-C mutant dynein. Table S1 and Table S2 show hydrogen bond pairs involving S3386 and C3386, respectively, in MD simulations. Table S3 shows hydrophobic interactions formed by C3386 in MD simulations. Table S4 gives details of oligonucleotides used for CRISPR experiments. Table S5 and Table S6 give details of primary and secondary antibodies, respectively. Video 1 shows exemplar movies of wild-type and *S3372C* mutant spindles with labeled histones and microtubules. Video 2 shows exemplar movies of wild-type and *S3372C* mutant spindles with labeled centrosomes and microtubules. Video 3 shows exemplar movies of wild-type and *S3372C* mutant spindles with labeled dynein and kinetochores. Video 4 shows exemplar movies of wild-type and *S3372C* mutant spindles with labeled Rod.

### Data availability

All data are included in the manuscript or supplement, or are available from S.L. Bullock on reasonable request.

## Acknowledgments

We thank members of the Bullock group and Medical Research Council (MRC) Laboratory of Molecular Biology fly community for support and input, Roger Karess (Institut Jacques Monod) for discussions, and many members of the international fly community for sharing reagents.

Work in the Bullock, Derivery, and Carter groups was supported by the MRC, as part of UK Research and Innovation (file reference numbers MC_U105178790, MC_UP_1201/13, and MC_UP_A025_1011, respectively). The work was also supported by a National Centre for the Replacement, Refinement and Reduction of Animals in Research David Sainsbury Fellowship (to A. Vagnoni), a Marie-Curie IntraEuropean Fellowship (to F. Port), and awards from National Institutes of Health (GM136414) and National Science Foundation (MCB-1055017 and MCB-1617028) (to A. Yildiz), the Fondation pour la Recherche Médicale (DEQ20170336742) (to R. Giet), a European Molecular Biology Organization Postdoctoral Fellowship (ALTF 334-2020) (to S. Chaaban), the Partnership for Advanced Computing in Europe (PRA2019215144 and PRA2021250119), and the Scientific and Technological Research Council of Türkiye (TUBITAK, 122N045 and 215Z398) (to M. Gur). M.A. McClintock was supported by a Biotechnology and Biological Sciences Research Council project grant (BB/T00696X/1) (to S.L. Bullock). Open Access funding provided by MRC Laboratory of Molecular Biology.

Author contributions: D. Salvador-Garcia: Conceptualization, Formal analysis, Investigation, Methodology, Resources, Validation, Visualization, Writing - original draft, Writing - review & editing, L. Jin: Formal analysis, Investigation, Methodology, Resources, Validation, A. Hensley: Formal analysis, Investigation, Writing - original draft, Writing - review & editing, M. Golcuk: Data curation, Formal analysis, Investigation, Methodology, Resources, Software, Validation, Visualization, Writing - original

draft, Writing - review & editing, E. Gallaud: Investigation, Writing - review & editing, S. Chaaban: Formal analysis, Visualization, Writing - review & editing, F. Port: Methodology, Writing - review & editing, A. Vagnoni: Funding acquisition, Methodology, Supervision, Validation, V.J. Planelles-Herrero: Investigation, Writing - review & editing, M.A. McClintock: Methodology, Supervision, E. Derivery: Conceptualization, A.P. Carter: Conceptualization, Funding acquisition, Supervision, R. Giet: Funding acquisition, Supervision, Writing - review & editing, M. Gur: Data curation, Formal analysis, Funding acquisition, Investigation, Methodology, Resources, Software, Supervision, Validation, Visualization, Writing - original draft, Writing - review & editing, A. Yildiz: Funding acquisition, Supervision, Writing - original draft, Writing - review & editing, S.L. Bullock: Conceptualization, Funding acquisition, Investigation, Project administration, Supervision, Validation, Visualization, Writing - original draft, Writing - review & editing.

Disclosures: The authors declare no competing interests exist.

Submitted: 6 October 2023

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

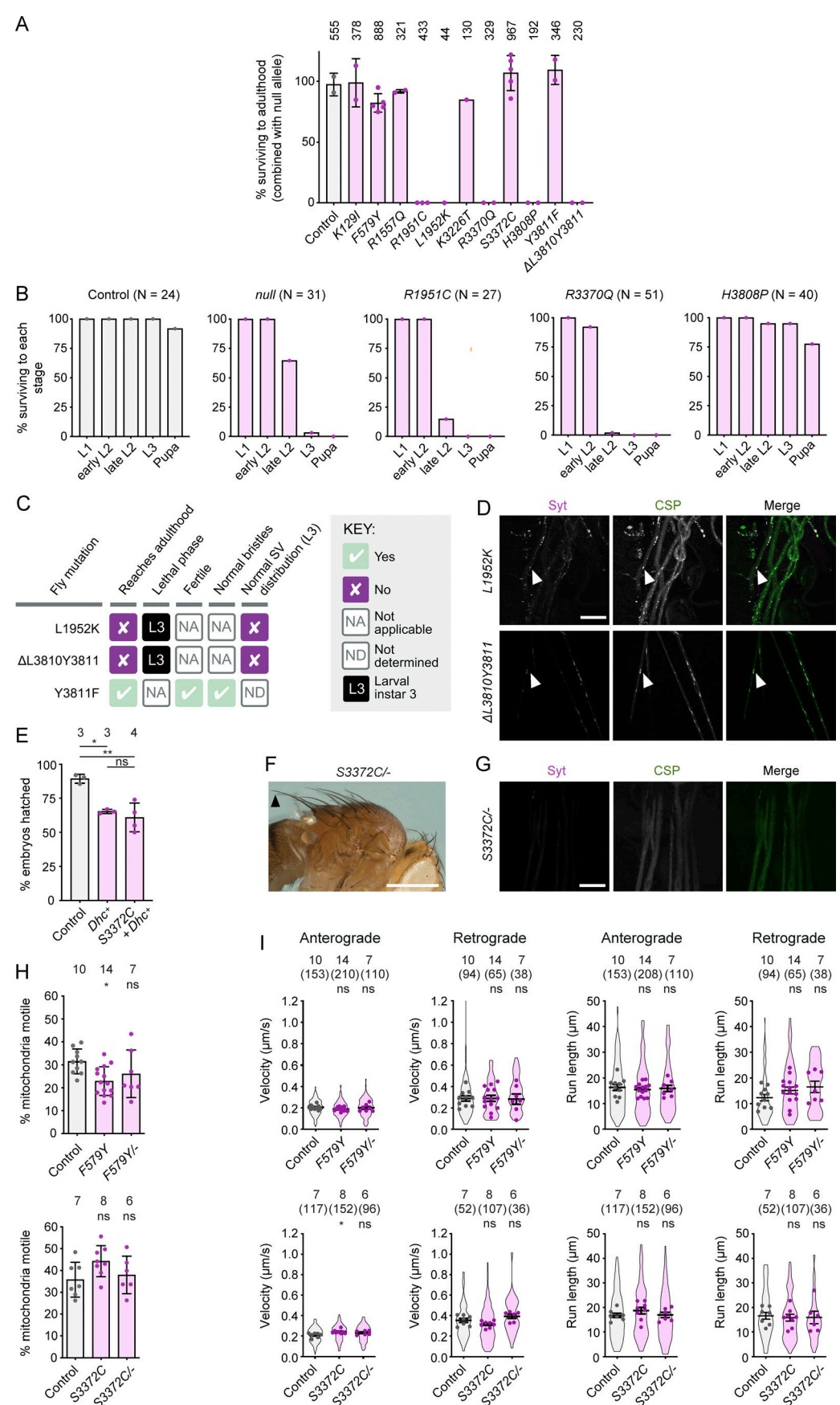

Figure S1. **Supplementary data from phenotypic analysis of Dhc mutations. (A)** Complementation tests showing percentage survival to adulthood of flies trans-heterozygous for the indicated *Dhc* mutation and a *Dhc^null* allele. Columns display mean values from individual crosses; error bars are SD; circles are values for individual crosses. Data were normalized based on the number of each genotype expected if there was no lethality given the total number of offspring. Numbers above each column show the total number of offspring assessed for each genotype. **(B)** Percentage of L1 larvae of indicated genotypes reaching the specified stage (there was no overt lethality at the embryonic stage). *N* is the number of animals analyzed for each genotype. In A and B, control animals were trans-heterozygous for a wild-type *Dhc* allele (recovered from the same CRISPR-Cas9 mutagenesis experiment that generated the *Dhc* mutant alleles) and the *Dhc^null* allele. **(C)** Summary of in vivo effects of L1952K, ΔL3810Y3811 and Y3811F. SV, synaptic vesicle. **(D)** Confocal images of segmental nerves (taken proximal to the ventral ganglion; anterior to the top; Z-projection) from L3 larvae stained for Synaptotagmin (Syt) and Cysteine-string protein (CSP). Arrowheads show examples of synaptic vesicle accumulations in mutants. Images are representative of three to six larvae analyzed per genotype. See Fig. 1 E for images from control. **(E)** Quantification of hatching frequency of eggs laid by mated females of indicated genotypes. Columns show mean values per egg collection; error bars represent SD; circles are values for individual egg collections. Numbers of collections per genotype (each from an independent cross; 414–1,107 eggs per collection) shown above bars. Control genotype was *yw*. *Dhc^+* is a genomic rescue construct; note this construct reduces hatching rate in the wild-type background and that fertility defects of *S3372C* mothers are only suppressed by the transgene to this point. **(F)** Image (representative of >160 flies examined) showing normal bristles in *S3372C/–* adult flies. Arrowhead points to posterior scutellar macrochaetae. See Fig. 1 D for control image. **(G)** Confocal images of segmental nerves (taken proximal to the ventral ganglion; anterior to the top; Z-projection) from fixed L3 larvae stained for Syt and CSP, showing lack of abnormal synaptic vesicle accumulations. Images are representative of three larvae analyzed. See Fig. 1 E for images from control. **(H)** Quantification of the percentage of mitochondria that exhibit transport in any direction in the adult wing nerve during the 3 min of data acquisition. Columns show mean values per movie; errors bars represent SD; circles are values for individual movies, each from a different wing. Number of wings analyzed shown above bars. **(I)** Quantification of velocity and run length of transported mitochondria in the adult wing nerve. Violin plots show values for individual mitochondria; circles show mean values per wing. Horizontal lines show mean ± SD of values for individual wings. Numbers without parentheses above bars are number of wings, with numbers of mitochondria in parentheses. Evaluation of statistical significance (compared to control) in E, H, and I was performed with a one-way ANOVA with Dunnett's multiple comparisons test (in I, the mean values per wing were compared): **, $P < 0.01$; *, $P < 0.05$; ns, not significant. Scale bars: D and G, 50 μm; F, 500 μm.

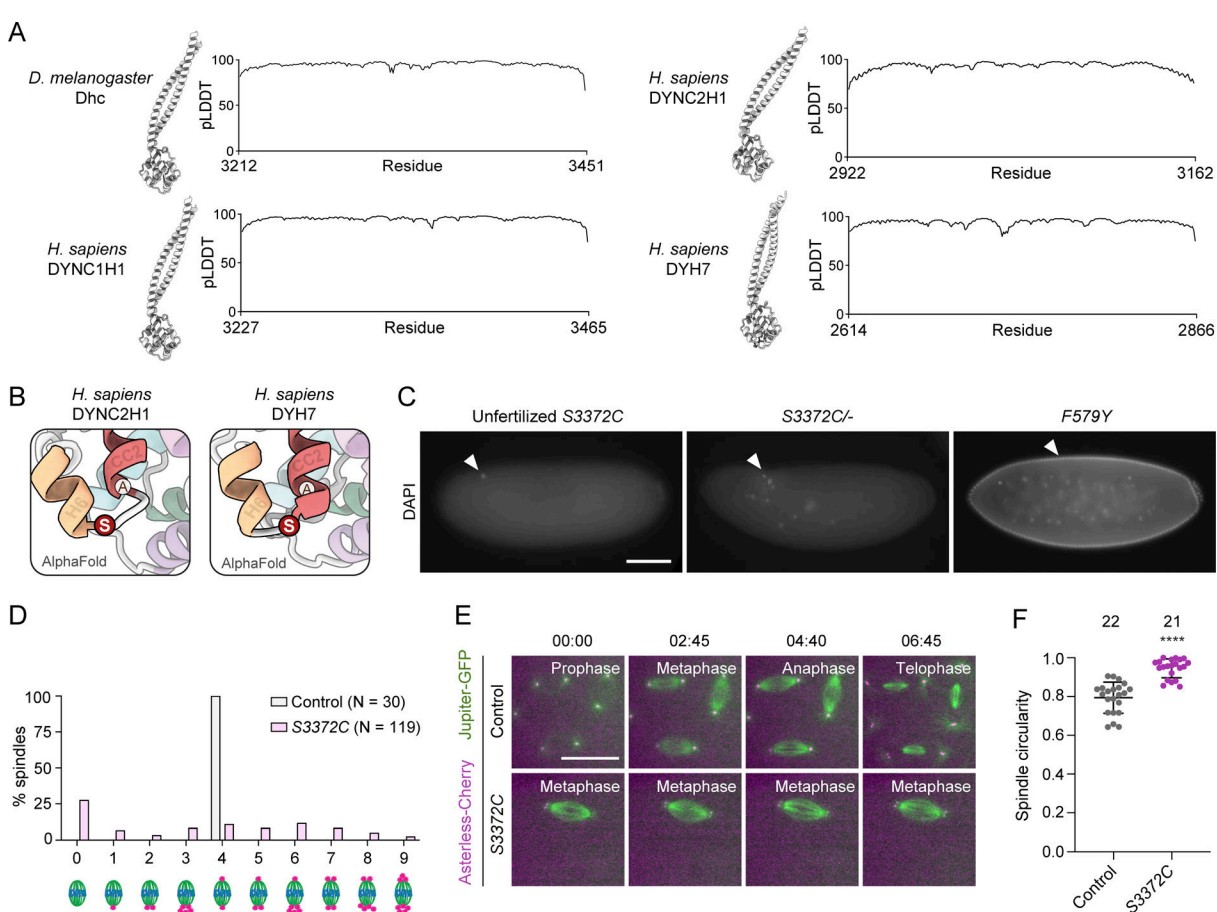

Figure S2.   **Supplementary data from structural and phenotypic analysis of S3372. (A)** Structural overview and pLDDT (predicted local distance difference test) plots (Jumper et al., 2021) of Alphafold2-generated structures of dynein MTBD and stalk regions. The high pLDDT values show that predictions are high confidence. **(B)** Zoom-ins of regions containing serine (S) residues equivalent to *Drosophila* S3372 in Alphafold2-generated structures of the MTBD and stalk regions of human DYNC2H1 (dynein-2 heavy chain) and DYH7 (inner arm axonemal dynein). Positions of residues equivalent to *Drosophila* S3372 are shown in red; alanines (A) at residues equivalent to the cysteines at the base of CC2 in several other dyneins are also shown. **(C)** Example widefield images from a 2- to 4-h egg collection of fixed, DAPI-stained unfertilized eggs from virgin *S3372C* females, or embryos from mated *S3372C/–* or *F579Y* females (arrowheads show DNA staining). Images are representative of at least 150 embryos examined. **(D)** Categorization of mitotic spindle phenotypes in *S3372C* embryos based on centrosome number and arrangement. A range of mitotic stages was present in control embryos (*yw*), whereas >90% of mutant spindles were at metaphase; only those control and mutant embryos in metaphase were scored for this analysis. *N* is the number of spindles scored (from six control and 49 *S3372C* embryos; no more than five randomly selected metaphase spindles analyzed per embryo). **(E)** Example stills from time series (single focal plane) of control and *S3372C* embryos acquired during preblastoderm cycles. Jupiter-GFP and Asterless-Cherry label microtubules and centrosomes, respectively. Note abnormal presence of two centrosomes at each pole of the mutant spindle. Images were binned 2 × 2. Timestamps are min:s. **(F)** Quantification of circularity of randomly selected spindles from early cleavage stage control and *S3372C* embryos (larger number equates to more circularity). Horizontal lines show means; error bars are SD; circles are values for individual spindles. Number of spindles analyzed per genotype shown above bars. Statistical significance was evaluated with an unpaired, two-tailed *t* test. ****, P < 0.0001. Scale bars: C, 100 µm; E, 20 µm.

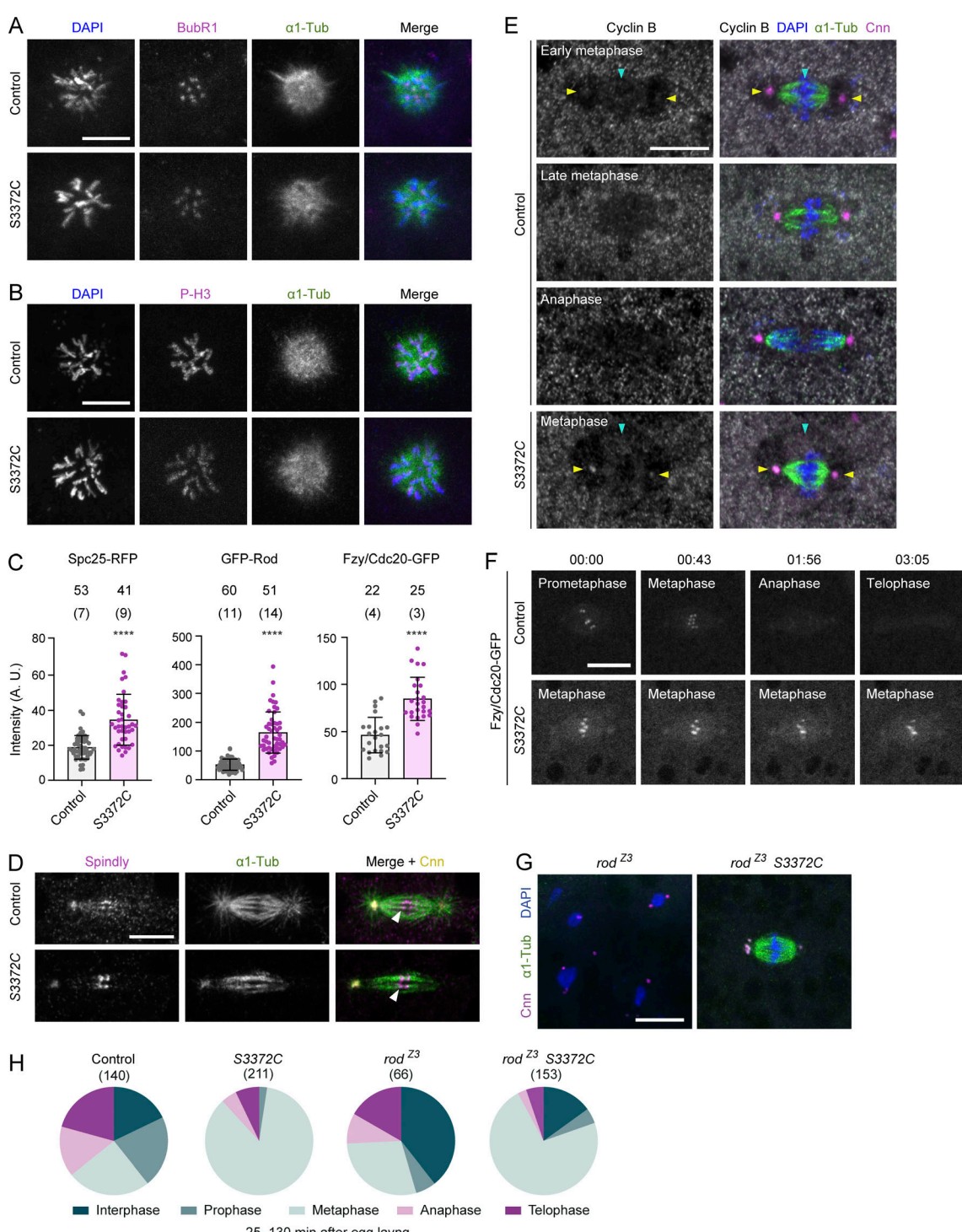

**Figure S3. Supplementary data from analysis of S3372C mitotic phenotype. (A and B)** Example confocal images of polar bodies in fixed control and *S3372C* embryos stained with DAPI, as well as antibodies to BubR1 and α1-tubulin (α1-Tub) (A) or phospho-histone H3 (P-H3) and α1-Tub (B) (Z-projections). **(C)** Quantification of intensity of Spc25-RFP, GFP-Rod, and Fzy/Cdc20-GFP puncta at the metaphase plate of control and *S3372C* embryos. Columns show means of values per punctum; error bars represent SD; circles are values for individual puncta. Number of puncta analyzed per genotype is shown above bars (number of embryos analyzed in parentheses). Statistical significance was evaluated with an unpaired, two-tailed *t* test (comparing per punctum means). ****, P < 0.0001. **(D)** Example confocal images of mitotic spindles in control and *S3372C* embryos stained with antibodies to Spindly, α1-Tub, and Centrosomin (Cnn), as well as DAPI. Arrowheads indicate an example of close apposition of plus ends of microtubule bundles and bright Spindly puncta at the kinetochore. Single focal planes were chosen to facilitate visualization of microtubule plus ends, which resulted in only one centrosome being visible in each image. **(E)** Example confocal images (Z-projections) of mitotic spindles in control and *S3372C* embryos stained with antibodies to Cyclin B, α1-Tub, and Cnn, as well as DAPI. Cyclin B shows weak accumulation in the vicinity of the spindle (blue arrowhead) and at centrosomes (yellow arrowheads) in early metaphase control embryos, which is lost by anaphase. In metaphase-arrested *S3372C* embryos, Cyclin B is detected in the vicinity of the spindle (blue arrowhead) and at centrosomes (yellow arrowheads). **(F)** Example stills of time series (single focal planes) of mitotic spindles acquired during preblastoderm cycles in live control and *S3372C* embryos

expressing Fzy/Cdc20-GFP. Fzy/Cdc20-GFP is localized to the metaphase plate in metaphase-arrested mutant spindles (see panel C for quantification). In A–B and D–F, at least 30 embryos were examined per condition. In F, timestamps are min:s. **(G and H)** Example confocal images of embryos of the indicated genotypes stained with antibodies to α1-Tub and Cnn, as well as DAPI (G) and incidence of mitotic stages for each condition (H). There was a significantly higher incidence of metaphase in $rod^{Z3}$ S3372C versus $rod^{Z3}$ embryos (72.6% versus 28.8% [P < 0.0001; Fisher's Exact test]). Data are from 28 control, 48 *S3372C*, 64 $rod^{Z3}$, and 119 $rod^{Z3}$ *S3372C* embryos with no more than five randomly selected nuclei or spindles analyzed per embryo. Note abnormal morphology of interphase nuclei in $rod^{Z3}$ single mutants. Scale bars: A, B, and D–F, 10 µm; G, 20 µm.

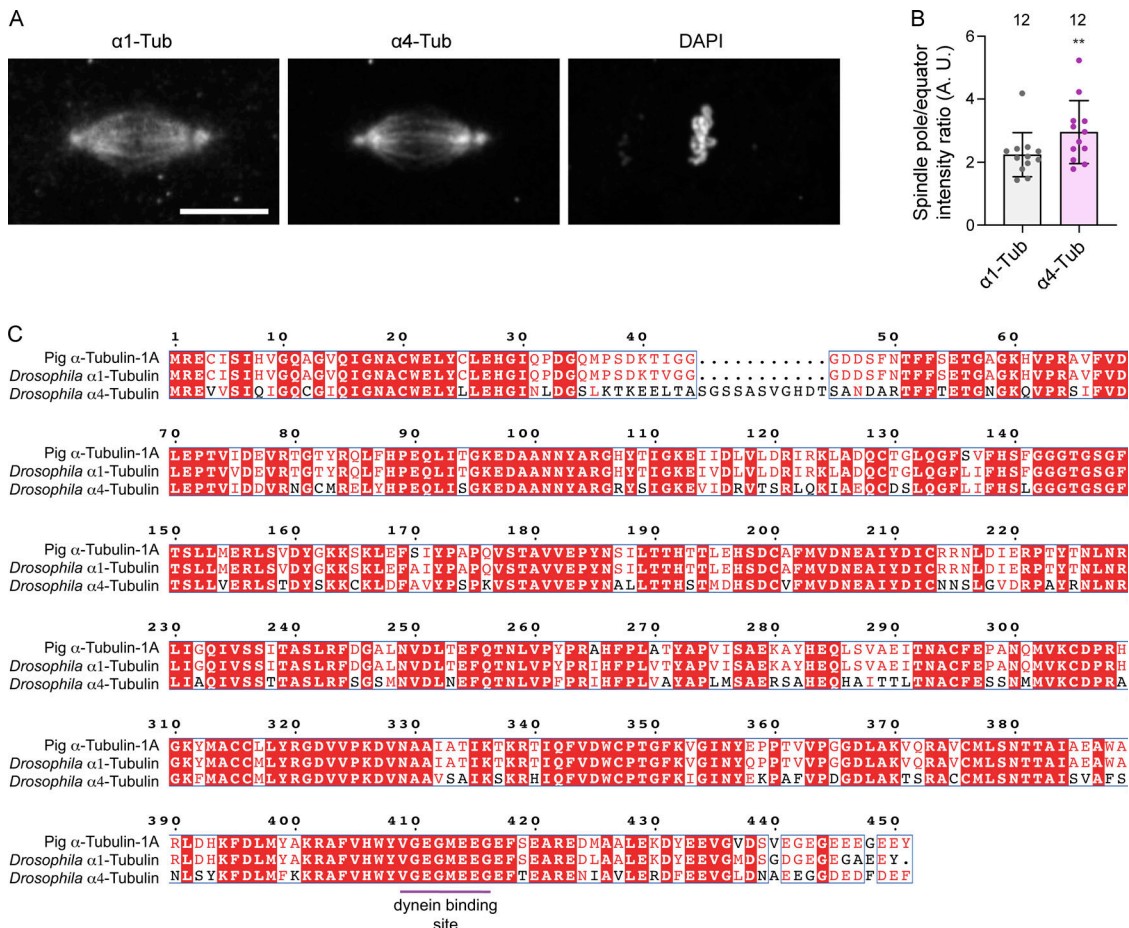

Figure S4. **Assessing a potential α4-tubulin–specific effect of S3372C. (A)** Example image of a metaphase spindle in a wild-type embryo stained with antibodies to α1-tubulin (α1-Tub) and α4-tubulin (α4-Tub), as well as DAPI. Scale bar: 10 µm. **(B)** Quantification of ratio of intensity of α1-Tub and α4-Tub at the pole versus the equator of metaphase spindles. Columns show mean values per spindle; error bars represent SD; circles are values for individual spindles. Number of spindles (randomly selected from five embryos of each genotype) shown above columns. Statistical significance between the ratio values was evaluated with a paired, two-tailed *t* test. **, P < 0.01. Higher α1-Tub signal at the spindle poles than the equator is presumably due to increased density of microtubules in this region. The observation that α4-Tub has a greater enrichment at the pole than the canonical tubulin isotype indicates preferential accumulation in microtubules at this site. **(C)** Alignment of protein sequences of mammalian (pig) α-Tub-1A, *Drosophila* α1-Tub, and *Drosophila* α4-Tub. White letters on a red background indicate residues present in all sequences; red letters indicate residues present in ≥50% of sequences; blue boxes show regions with ≥50% conservation; magenta horizontal line, region contributing to dynein binding (based on the mouse MTBD-microtubule structure [Lacey et al., 2019]), which is identical in all three proteins. Uniprot accession numbers are: pig (*Sus scrofa*) α-Tub-1A, P02550; *D. melanogaster* α1-Tub, P06603; *D. melanogaster* α4-Tub, P06606.

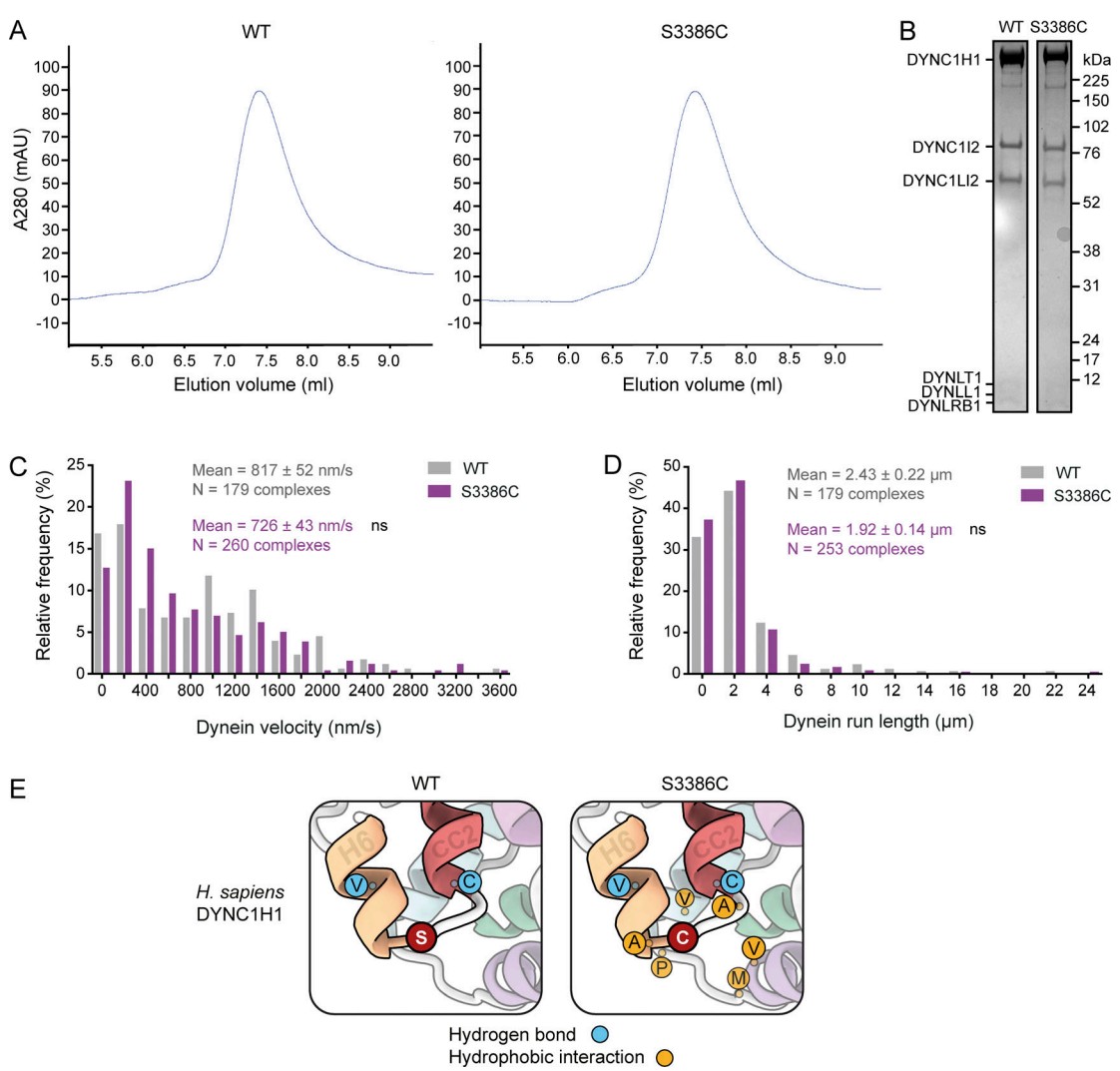

Figure S5. **Supplementary data from in vitro and in silico analysis of S3386C human dynein. (A)** Size-exclusion chromatography traces for wild-type (WT) and S3386C human dynein complexes, showing very similar profiles and lack of aggregation. **(B)** Cropped images from a Coomassie Blue–stained SDS-PAGE gel of pooled and concentrated fractions collected from the WT and S3386C mutant dynein peaks shown in panel A. **(C and D)** Velocity (C) and run length (D) frequency distributions for processive WT and S3386C mutant dynein complexes in the presence of dynactin and BICD2N. Errors are SEM. Evaluation of statistical significance was performed with a Mann-Whitney test. ns, not significant. **(E)** Zoom-ins of regions of representative examples of WT and S3386C dynein MTBD and stalk MD simulations. Single-letter amino acid codes are shown. Frequent hydrogen bonding interactions of S3386 and C3386 are shown as blue circles and new hydrophobic interactions of C3386 as gold circles; see Tables S1, S2, and S3 for details of interacting residues. Source data are available for this figure: SourceData FS5.

Video 1. **Live imaging of microtubules and chromatin in embryonic mitosis.** Composite of example movies (generated with spinning disk confocal microscopy) of a single focal plane of control and *S3372C* preblastoderm cycle embryos. Jupiter-GFP (green) and His2Av-mRFP (magenta) label microtubules and chromatin, respectively. In 57 out of 60 movies (of 15-min duration each) of S3372C embryos that expressed fluorescent markers of the mitotic apparatus, no progression of metaphase-arrested spindles was observed. In the remaining movies, progression of highly abnormal mitotic spindles that were formed by fusion of two spindles was seen. Images were collected at 0.2 frames/s with a playback rate of 10 frames/s. Timestamps are min:s. Scale bar, 10 µm. Related to Fig. 5.

Video 2. **Live imaging of microtubules and centrosomes in embryonic mitosis.** Composite of example movies (generated with spinning disk confocal microscopy) of a single focal plane of control and *S3372C* preblastoderm cycle embryos. Jupiter-GFP (green) and Asterless-Cherry (magenta) label microtubules and centrosomes, respectively. Images were collected at 0.2 frames/s with a playback rate of 15 frames/s. Images were binned 2 × 2. Timestamps are min:s. Scale bar, 10 µm. Related to Fig. S2.

Video 3.  **Live imaging of dynein and kinetochores in embryonic mitosis.** Composites of example movies (generated with spinning disk confocal microscopy) of a single focal plane of control and *S3372C* preblastoderm cycle embryos. GFP-Dlic (green in top merged movies) and Spc25-RFP (magenta in top merged movies) label dynein complexes and kinetochores, respectively. Two loops of the movie are shown; the second loop pauses to show transient accumulation of GFP-Dlic at kinetochores (arrows). Images were collected at 0.2 frames/s with a playback rate of 10 frames/s. Timestamps are min:s. Scale bar, 10 µm. Related to Fig. 6.

Video 4.  **Live imaging of GFP-Rod streaming in embryonic mitosis.** Composite of example movies (generated with spinning disk confocal microscopy) of a single focal plane of control and *S3372C* preblastoderm cycle embryos showing GFP-Rod dynamics. Streaming of faint GFP-Rod signals away from the kinetochores was analyzed in kymographs. Images were collected at 0.5 frames/s with a playback rate of 10 frames/s. Timestamps are min:s. Scale bar, 20 µm. Related to Fig. 7.

**Provided online are six tables. Table S1 shows occurrence in simulations of hydrogen bond pairs involving S3386. Table S2 shows occurrence in simulations of hydrogen bond pairs involving C3386. Table S3 shows occurrence in simulations of hydrophobic interaction partners of C3386. Table S4 shows oligonucleotides used to create Dhc mutations in *Drosophila*. Table S5 shows primary antibodies used in this study. Table S6 shows secondary antibodies used in this study.**

