## [Peer Review File · The Journal of Cell Biology]

A force-sensitive mutation reveals a non-canonical role for dynein in anaphase progression

David Salvador-Garcia, Li Jin, Andrew Hensley, Mert Golcuk, Emmanuel Gallaud, Sami Chaaban, Phillip Port, Alessio Vagnoni, Vicente Planelles-Herrero, Mark McClintock, Emmanuel Derivery, Andrew Carter, Regis Giet, Mert Gur, Ahmet Yildiz, and Simon Bullock

Corresponding Author(s): Simon Bullock, MRC Laboratory of Molecular Biology

Review Timeline:

Submission Date:	2023-10-06
Editorial Decision:	2023-10-24
Revision Received:	2024-04-29
Editorial Decision:	2024-06-05
Revision Received:	2024-06-12

Monitoring Editor: Hironori Funabiki

Scientific Editor: Tim Fessenden

Transaction Report:

DOI: <https://doi.org/10.1083/jcb.202310022>

Revision 0

Review #1

1. Evidence, reproducibility and clarity:

Evidence, reproducibility and clarity (Required)

The authors start out by examining the cellular and organismal effects of 6 human disease-linked mutations, as well as the mouse legs-at-odd-angles (Loa) mutation, after introducing the mutations into the *D. melanogaster* dynein heavy chain (Dhc) by genome editing. This reveals an overall correlation between the severity of the effect on dynein motor activity *in vitro* (determined in a previous study) and the penetrance of the corresponding mutant phenotype in the fly, with a couple of interesting exceptions that illustrate the value of performing structure-function analysis of dynein in animal models. The authors then focus on an additional missense mutation in the Dhc microtubule binding domain, fortuitously generated by imprecise editing, that results in a striking phenotype. The S3372C mutation supports normal development, including normal axonal transport of mitochondria and asymmetric mitosis of larval neuroblasts, but female flies are infertile. Through elegant genetics, ectopic disulfide bond formation with a nearby residue is ruled out as the cause of the maternal effect. S3372C results in metaphase arrest of early embryonic divisions, characterized by over-accumulation of dynein light intermediate chain (Dlic) and the dynein recruitment factor Rod at kinetochores, as well as by reduced poleward streaming of Rod along spindle microtubules. Surprisingly, the S3372C-induced metaphase arrest cannot be bypassed by inhibiting the spindle assembly checkpoint, implying that dynein promotes the metaphase-to-anaphase transition not solely through its known function in spindle assembly checkpoint silencing. *In vitro* motility and optical trapping experiments show that the mutant motor performs normally in a load-free regime but exhibits reduced peak force production and excessive pausing under load. Furthermore, molecular dynamics simulations reveal how the mutation affects dynein's interaction with microtubules, including a change in the positioning of the stalk. The authors conclude that the S3372C mutation specifically perturbs high-load functions of dynein, explaining the selective phenotype observed *in vivo*.

The experiments are technically on a very high level, the results are presented in a clear manner, and the conclusions are fully supported by the data.

****Minor suggestions (optional):****

- In the first part of the paper, where Dhc mutations associated with neurological disease are examined, the H3808P and F579/Loa mutations are shown to cause mis-accumulation of synaptic vesicles in axons. The authors may want to perform this assay for the K129I, R1557Q, and K3226T mutations, as this would strengthen the comparative analysis of *in vitro* versus *in vivo* effects, summarized in Figure 1C. For example, K129I has a more severe effect *in vitro* than the Loa mutation, but the Loa mutation has a more pronounced phenotype on the

organismal level. Would this also be the case in a cell-based assay?

- The observation that the metaphase arrest of S3372C mutant embryos cannot be alleviated by the checkpoint-defective Mad2 mutant is very intriguing, as is the observation that Dlic and the RZZ subunit Rod over-accumulate at/near kinetochores. As discussed by the authors, one possibility is that the arrest is a consequence of dynein's failure to disassemble the corona by stripping, but, surprisingly, in a manner unrelated to dynein's role in SAC silencing. In this regard, it is interesting to note that fly RZZ mutants do not undergo metaphase arrest in the early embryo (Williams and Goldberg, 1994; Défachelles et al., 2015), whereas knockdown of Spindly, which functions in dynein recruitment downstream of RZZ, does lead to arrest (see Figure 2 in Clemente et al., 2018; PMID 29615558). Taken together, this raises the possibility that it is the failure to remove RZZ (and other associated corona components) from kinetochores that inhibits anaphase onset in S3372C embryos. It would therefore be interesting to test whether the metaphase arrest in S3372C embryos is alleviated in RZZ mutants.

****Referees cross-commenting****

The Mad2 mutant the authors use is a P-element insertion that was described by Buffin et al 2007 as a null mutant with regards to SAC signaling (it also does not produce any detectable protein by Western blot; Figure 1b). Nevertheless, since the analysis in Buffin et al was restricted to larval brains, I agree with reviewer #2 that it remains to be formally demonstrated that this Mad2 mutant fully abolishes the SAC in the early embryo. Unfortunately, as far as I am aware, reversine does not work well in *Drosophila*. An alternative would be to combine Dhc(S3372C) with the other Mad2 mutant used by Buffin et al, which (besides not producing detectable protein) lacks the Mad1 binding domain and can therefore be expected to be a definitive checkpoint null in all tissues.

2. Significance:

Significance (Required)

The cytoplasmic dynein 1 motor complex participates in a multitude of cellular processes that require microtubule minus-end-directed motility. Whereas *in vitro* reconstitution efforts have led to a detailed understanding of the motor's motile properties, the essential requirement of dynein for development has made it challenging to dissect how the motor contributes to specific aspects of intracellular organization and cell division *in vivo*. The need for a mechanistic understanding of how dynein motility is used for diverse functions *in vivo* is underscored by missense mutations in the motor subunit that cause human neurological disease. In this interesting and insightful study, Salvador-Garcia and colleagues characterize several missense mutations in dynein heavy chain (Dhc) using biochemical assays and genetic approaches in the fly, which reveals the distinct effects of disease-causing mutations *in vivo* and uncovers an unanticipated novel function of dynein in regulating mitotic progression.

This beautifully executed study has important implications for dynein's role in mitosis, in particular its role at the kinetochore, and is of broad interest to cell biologists studying the cytoskeleton, as it demonstrates that examining motor mutants with altered mechanical properties *in vivo* can reveal specific motor functions.

3. How much time do you estimate the authors will need to complete the suggested revisions:

Estimated time to Complete Revisions (Required)

(Decision Recommendation)

Between 1 and 3 months

Yes

Review #2

1. Evidence, reproducibility and clarity:

Evidence, reproducibility and clarity (Required)

In the manuscript by Salvador-Garcia et al., the authors assess the physiological consequences of dynein mutations in flies and in vitro. In addition to characterizing the manner by which disease causing mutations affect fly development and some aspects of their cell physiology, the authors focus on sporadic mutations that arose during the course of generating their mutant fly lines. Of particular interest was a mutation in the dynein MTBD: S3722C. This mutation caused very interesting phenotypes in flies (e.g., infertility in females likely due to mitotic arrest) as well as in reconstituted motility assays (e.g., reduced stall force). The authors posit that the mitotic arrest phenotype is a consequence of a dynein's role in initiating anaphase onset, and that this role is distinct from its well established role in silencing the spindle assembly checkpoint.

The paper is very well written, and the data are of high quality. Most of the claims - with one major exception (described below) - are well supported by the data. I have a few comments that might help to strengthen the conclusions.

****Major comment:****

1. In brief, I'm not convinced the mitotic arrest phenotype is not a consequence of impaired SAC silencing by the mutant dynein. The main tool the authors use to support their claim is a Mad2 mutant. They use this to determine if preventing SAC function (with the mutant Mad2) can override the ability of S3722C cells to progress into anaphase. The Mad2 mutant does in fact increase the proportion of cells exiting mitosis (from 0.8 to 14% of cells); however, the low number (14%) suggests that an inability to silence the SAC is not the reason the cells are not entering anaphase (i.e., it is SAC silencing-independent). My question is how penetrant the Mad2 mutant is? For example, how many cells with this Mad2 mutant would exit mitosis if the authors perturbed mitosis some other way (e.g., treatment with high concentrations of nocodazole)? If the number is still low (~14%), then this might be why the mutant can't rescue the mitotic exit phenotype for S3722C cells, and would challenge the following statements: "...it suggests that this can make, at best, a minor contribution to the mitotic arrest phenotype"; and "Remarkably, the MTBD mutation does not appear to block anaphase progression in embryos by preventing the well-characterized role of kinetochore-associated dynein in silencing the SAC, as the defect persists when the checkpoint is inactivated by mutation of Mad2. Collectively, these observations indicate that kinetochore dynein has a novel role in licensing the transition from metaphase to anaphase." Although previous studies might have assessed the penetrance of this mutant in other cell types, given the cell specificity of the mitotic arrest phenotype for S3722C (in embryonic cells, but not L3 neuroblasts), it will be important to provide such additional evidence in embryonic cells (e.g., nocodazole treatment of embryonic cells) to support these statements, especially in light of the bold conclusions and hypotheses they are making (e.g., "For example, the apparent variability in tension between sister kinetochores in S3372C embryos, which could reflect abnormal force generation by the mutant motor complex, might prevent APC/C activation through the complex series of signaling events that respond to chromosome biorientation."). Although it would be fascinating if the authors are correct that dynein provides another role in licensing anaphase onset, the well-established role for dynein in checkpoint silencing currently seems like the most parsimonious explanation.

****Minor comments:****

1. The S3722C mutant appears to accumulate to higher-than-normal levels at KT's and to some extent along the spindle MTs. In addition to the representative images, it would be helpful to see a quantitation of this phenomenon for WT, S3722C, and S3722C/+ along with statistics.
2. Although the mean inter-KT distance was unchanged between WT and S3722C cells, the authors note that the deviation from the mean was higher. Could this simply be a consequence of more highly dynamic oscillations of KT pairs (similar to that seen with Hec1-S69D in DeLuca et al., JCB 2018)? More dynamic oscillations could potentially lead to more variable distances between KT pairs.
3. It is interesting that the S3722C/Mad2-mutant cells are enriched in telophase (Fig. 7G). Does this not suggest another arrest point for these cells?
4. The authors state: "Immunostaining revealed that whereas α 1-tubulin was present throughout the spindle apparatus, α 4-tubulin was enriched at the spindle poles (Figure S7A)." Although I agree the α 4-tubulin appears somewhat enriched at the poles with respect to α 1-

tubulin, a quantitation (with statistics) would be needed to support this claim. That being said, I agree the isotype is unlikely to account for the S3722C phenotype.

5. Trapping data show reduced stall force, yet increased stall time at low resistive forces for the mutant. This finding could potentially account for the reduced velocity of GFP-Rod noted in cells; however, I wonder if the authors noted altered velocity for dynein-driven bead movement under load in their trapping assays? This information would be useful to include in their manuscript.

6. Is there a defocused spindle pole phenotype in the mutant cells? The cells in Video 1 and Fig. S6c appear to show as much, although other cells do not.

7. The authors state: "This may reflect the relatively short length of *Drosophila* neurons making them less sensitive to partially impaired cargo transport." Could the extent of the phenotypes also be related to the lifespan of the flies? Do any of the diseases caused by these mutations have late-onset in patients? I wonder if a subtle defect in dynein behavior might not manifest for numerous years due to only minor changes in motility?

****Referees cross-commenting****

I wanted to reiterate my skepticism regarding the possibility that their data strongly support a SAC-independent role for dynein in the metaphase-to-anaphase transition (it seems Reviewer #3 might agree with me). I don't think it's impossible, but I'm not convinced they've made a very strong case for this model, which is noted in the title. The fact that proteins are accumulating at aligned kinetochores in the mutant cells (e.g., Rod and DLIC) in fact are consistent with a SAC silencing defect. Along these lines, I think reviewer #1's point regarding RZZ is a good one (that the mutant dynein is incapable of evicting RZZ specifically from aligned KT's), and should be tested prior to publication.

My primary concern is that their conclusion is based entirely on the fact that the Mad2 mutant does not fully restore mitotic exit to the dynein mutant cells. Given the cell-specificity of their dynein phenotype (in embryonic cells, but not L3 neuroblasts), I think testing the penetrance of their Mad2 mutant in the embryonic cells would need to be assessed. In my review I suggested nocadazole, when I realized I meant to say reversine (oops!).

2. Significance:

Significance (Required)

The manuscript by Salvador-Garcia et al. is a very interesting study dissecting the physiological consequences of dynein mutations in flies and in vitro. This study will be of high interest to those in the dynein/molecular motor field, as well as those that study mitosis and kinetochore function. One of the most interesting findings in the study is the identification of a mutation in dynein that specifically impacts its motility in conditions of high load. This mutant provides a novel tool to dissect load-dependent transport for dynein in other systems. Moreover, the study suggests a novel role for dynein in promoting anaphase onset; if the authors can provide additional support for this claim, the impact of this study would be greater.

Field of expertise: molecular motors, kinetochore function

3. How much time do you estimate the authors will need to complete the suggested revisions:

Estimated time to Complete Revisions (Required)

(Decision Recommendation)

Between 1 and 3 months

Yes

Review #3

1. Evidence, reproducibility and clarity:

Evidence, reproducibility and clarity (Required)

****Summary****

In this manuscript Salvador-Garcia et al. examine how several mutations in the Dynein heavy chain (DHC) influence Dynein function in an in vitro reconstituted system and in *Drosophila* embryos. Most importantly, they identify a novel substitution mutation (S3372C, generated as a by-product of a targeted CRISPR mutagenesis screen) that leads to a novel phenotype specifically in early *Drosophila* embryos (metaphase arrest) and that only impairs DHC function under load in vitro. Most surprisingly, the metaphase arrest in embryos does not appear to be due to a failure to inactivate the spindle-assembly-checkpoint (SAC), a known DHC function. This suggests that DHC has a hitherto unappreciated function in allowing spindles in the *Drosophila* embryo to progress from metaphase-to-anaphase.

The manuscript is generally well written and conveys the major conclusions clearly and

concisely. The data is generally of high quality, and largely supports the main conclusions, although there is one set of relatively straight-forward experiments that I think would be an important addition (see major comment #1, below).

****Major Comments****

1. The observation that the S3372C mutation causes a mitotic arrest that is not SAC dependent (i.e. it still largely occurs even in a Mad2 mutant background) is very surprising, and is the basis of the authors claim of a new, DHC-dependent, mechanism that allows embryo spindles to progress into anaphase. I think it would be important to assess whether the SAC components are still localising to the kinetochores in these S3372C, Mad2 double mutants (e.g. is Rod still recruited to high-levels in the double mutant?). If the SAC components are still being recruited to the spindle (suggesting that they are still detecting that the spindle is not ready to go into anaphase), is it worth considering that Mad2 may not be essential for SAC function in these embryos? I say this because I find it hard to imagine how any, presumably mechanical, failure at the kinetochore that leads to the improper metaphase/anaphase transition in the S3372C mutants, would signal to the rest of the spindle to not transition to anaphase if the SAC is truly inactivated. Do the authors think these embryos have a completely unrelated surveillance system that detects the S3372C-dependent error (whatever that is) and arrests the spindles specifically in embryos? Or is the error itself sufficient to cause a spindle-wide arrest, which seems improbable?

2. I was surprised the authors made no attempt to quantify the level of over-accumulation of Dlic (Figure 6) or Rod (Figure 7) (and the lack of over-accumulation in other regions of the spindle). The images are convincing, so I don't doubt that this is the case, but I think some sort of quantification would be useful and I don't think it would be hard to come up with a way to do this (even just drawing a ROI around the approximate areas of interest). It would also be interesting to know whether other proteins like Spc25 (Figure 6) and Cdc20 (Figure S6) are recruited to normal levels at kinetochores.

****Minor Comments****

1. In the Discussion the authors state: "Our discovery of a missense mutation that strongly affects nuclear divisions in the embryo without disrupting other dynein functions offers a unique tool to study the mitotic roles of the motor". This should be reworded, as it suggests that the mutation effects all mitotic functions of DHC, which is clearly not the case (and also applies only to the embryo).

2. I think it worth more explicitly stating that there is no evidence that the defects the S3372C mutation lead to in the in vitro reconstituted system are the cause of the in vivo defects observed in the embryo. The authors are careful not to directly claim this, and I agree with their assertion that this is the most "parsimonious explanation" for their data, but I'm sure they would agree that this is far from proven, and it might be worth emphasising this point a little more.

2. Significance:

Significance (Required)

This is a well conducted study that significantly extends the author's previous work on how mutations in DHC (initially indentified in human patients) effect DHC function (Hoang et al., PNAS, 2017). The paper reports the striking central finding that the S3372C mutation produces a very unusual mitotic arrest phenotype specifically in Drosophila embryos, and the authors link this to the also striking finding that this mutation only disrupts DHC function in vitro when DHC is working under load. As mentioned above, this link is not proven here, but this is a solid working hypothesis that is potentially of significant interest to those working on molecular motors and their role in fundamental cell biology and human disease.

I am a cell biologist with expertise in the cytoskeleton, particularly during early Drosophila embryogenesis.

3. How much time do you estimate the authors will need to complete the suggested revisions:

Estimated time to Complete Revisions (Required)

(Decision Recommendation)

Less than 1 month

No

Revision Plan

Manuscript number: #RC-2023-02116

Corresponding author(s): Simon Bullock

1. General Statements

This section is optional. Insert here any general statements you wish to make about the goal of the study or about the reviews.

We are very grateful to the reviewers for evaluating the manuscript in detail, as well as their constructive advice for revising the study. Please note that many of the reviewers' general comments, as well as their significance statements, are not included in the following point-by-point revision plan.

2. Description of the planned revisions

Insert here a point-by-point reply that explains what revisions, additional experimentations and analyses are planned to address the points raised by the referees.

Reviewer 1:

The experiments are technically on a very high level, the results are presented in a clear manner, and the conclusions are fully supported by the data.

Minor suggestions (optional):

- In the first part of the paper, where Dhc mutations associated with neurological disease are examined, the H3808P and F579/Loa mutations are shown to cause mis-accumulation of synaptic vesicles in axons. The authors may want to perform this assay for the K129I, R1557Q, and K3226T mutations, as this would strengthen the comparative analysis of in vitro versus in vivo effects, summarized in Figure 1C. For example, K129I has a more severe effect in vitro than the Loa mutation, but the Loa mutation has a more pronounced phenotype on the organismal level. Would this also be the case in a cell-based assay?

This is a very good suggestion. We will perform immunostainings for synaptic vesicle markers in K129I, R1557Q, and K3226T mutant 3rd instar larvae and incorporate the results in the manuscript.

- The observation that the metaphase arrest of S3372C mutant embryos cannot be alleviated by the checkpoint-defective Mad2 mutant is very intriguing, as is the observation that Dlic and the RZZ subunit Rod over-accumulate at/near kinetochores. As discussed by the authors, one possibility is that the arrest is a consequence of dynein's failure to disassemble the corona by stripping, but, surprisingly, in a manner unrelated to dynein's role in SAC silencing. In this regard,

Revision Plan

it is interesting to note that fly RZZ mutants do not undergo metaphase arrest in the early embryo (Williams and Goldberg, 1994; Défachelles et al., 2015), whereas knockdown of Spindly, which functions in dynein recruitment downstream of RZZ, does lead to arrest (see Figure 2 in Clemente et al., 2018; PMID 29615558). Taken together, this raises the possibility that it is the failure to remove RZZ (and other associated corona components) from kinetochores that inhibits anaphase onset in S3372C embryos. It would therefore be interesting to test whether the metaphase arrest in S3372C embryos is alleviated in RZZ mutants.

We agree that this is a potentially very informative experiment. During the review process, we had in fact generated a double mutant chromosome for *Dhc [S3372C]* and the maternal effect *Rod* allele (*Z3*), which has been shown to prevent RZZ association with the kinetochore and inactivate the SAC in embryos (Défachelles et al., 2015). Our aim here was to determine if interfering with RZZ function and the SAC in a way other than Mad2 depletion alleviates the metaphase arrest caused by the dynein mutation. Our preliminary evidence suggests that *Rod [Z3]* does not suppress the metaphase arrest caused by *Dhc [S3372C]*, which would be consistent with the model proposed in the initial manuscript in which dynein has an essential function in anaphase progression in addition to RZZ stripping and SAC inactivation. However, we need to perform additional experiments to confirm that our early observations with the *Dhc [S3372C] Rod [Z3]* embryos are reproducible. We intend to include our results in the full resubmission.

****Referees cross-commenting****

The Mad2 mutant the authors use is a P-element insertion that was described by Buffin et al 2007 as a null mutant with regards to SAC signaling (it also does not produce any detectable protein by Western blot; Figure 1b). Nevertheless, since the analysis in Buffin et al was restricted to larval brains, I agree with reviewer #2 that it remains to be formally demonstrated that this Mad2 mutant fully abolishes the SAC in the early embryo. Unfortunately, as far as I am aware, reversine does not work well in *Drosophila*. An alternative would be to combine *Dhc(S3372C)* with the other Mad2 mutant used by Buffin et al, which (besides not producing detectable protein) lacks the Mad1 binding domain and can therefore be expected to be a definitive checkpoint null in all tissues.

We plan to check by immunoblotting if the *Mad2[P]* allele, which several groups have used as a null mutation in *Drosophila* (including in embryos (Défachelles et al. 2015)), fails to produce Mad2 protein in embryos. Our expectation is that this will be the case as the P-element integration should disrupt the protein coding capacity of the gene in all tissues. The alternative *Mad2* allele referred to by this reviewer (*Mad2[Δ]*; Buffin et al. 2007) also has a deletion of two other genes *S6k* and *kri* – encoding ribosomal protein S6 kinase and the krishah uracil phosphoribosyltransferase – which can confound phenotypic analysis. It would be very challenging to make a recombinant chromosome containing *Dhc [S3372C]*, *Mad2[Δ]* and a rescue construct for both *S6k* and *kri*. In any case, we believe that the most definitive way to address this point of the reviewer would be to show that Mad2 protein is absent in the *Mad2[P]* embryos. Interfering with the SAC via the *Rod[Z3]* allele offers another means to test the involvement of the checkpoint in the S3372C-mediated metaphase arrest (see response to previous point).

Revision Plan

Reviewer 2:

The paper is very well written, and the data are of high quality. Most of the claims - with one major exception (described below) - are well supported by the data. I have a few comments that might help to strengthen the conclusions.

Major comment:

1) In brief, I'm not convinced the mitotic arrest phenotype is not a consequence of impaired SAC silencing by the mutant dynein. The main tool the authors use to support their claim is a Mad2 mutant. They use this to determine if preventing SAC function (with the mutant Mad2) can override the ability of S3722C cells to progress into anaphase. The Mad2 mutant does in fact increase the proportion of cells exiting mitosis (from 0.8 to 14% of cells); however, the low number (14%) suggests that an inability to silence the SAC is not the reason the cells are not entering anaphase (i.e., it is SAC silencing-independent). My question is how penetrant the Mad2 mutant is? For example, how many cells with this Mad2 mutant would exit mitosis if the authors perturbed mitosis some other way (e.g., treatment with high concentrations of nocodazole)? If the number is still low (~14%), then this might be why the mutant can't rescue the mitotic exit phenotype for S3722C cells, and would challenge the following statements: "...it suggests that this can make, at best, a minor contribution to the mitotic arrest phenotype"; and "Remarkably, the MTBD mutation does not appear to block anaphase progression in embryos by preventing the well-characterized role of kinetochore-associated dynein in silencing the SAC, as the defect persists when the checkpoint is inactivated by mutation of Mad2. Collectively, these observations indicate that kinetochore dynein has a novel role in licensing the transition from metaphase to anaphase." Although previous studies might have assessed the penetrance of this mutant in other cell types, given the cell specificity of the mitotic arrest phenotype for S3722C (in embryonic cells, but not L3 neuroblasts), it will be important to provide such additional evidence in embryonic cells (e.g., nocodazole treatment of embryonic cells) to support these statements, especially in light of the bold conclusions and hypotheses they are making (e.g., "For example, the apparent variability in tension between sister kinetochores in S3372C embryos, which could reflect abnormal force generation by the mutant motor complex, might prevent APC/C activation through the complex series of signaling events that respond to chromosome biorientation."). Although it would be fascinating if the authors are correct that dynein provides another role in licensing anaphase onset, the well-established role for dynein in checkpoint silencing currently seems like the most parsimonious explanation.

The Karess lab have previously reported that the SAC is non-functional in *Mad2[P]* mutant embryos (Défachelles et al., 2015). This is based on their observation that, unlike wild-type controls, *Mad2[P]* syncytial embryos do not undergo mitotic arrest when the spindle assembly process is perturbed using low concentrations of colchicine (which, like nocodazole, targets microtubules). Défachelles et al. used a scoring system in which at least 50% of the spindles had to be non-mitotic to be scored, showing that the penetrance of the *Mad2[P]* phenotype in embryos

Revision Plan

is high. In hindsight, we should have made it clear in the original manuscript that Défachelles et al. reported that the SAC is inactive in *Mad2[P]* embryos (not just larval brains), and we now do so in the preliminary revision (lines 389-390). As described in our response to Reviewer 1, we will also test the involvement of the SAC in the S3372C-mediated metaphase arrest using the maternal effect *Rod [Z3]* mutation, which Défachelles et al. found also inactivates the SAC in embryos using the colchicine-based assay.

Minor comments:

1) The S3722C mutant appears to accumulate to higher-than-normal levels at KT's and to some extent along the spindle MTs. In addition to the representative images, it would be helpful to see a quantitation of this phenomenon for WT, S3722C, and S3722C/+ along with statistics.

We will perform quantification and statistical analysis on our existing images as requested and include this in the full revision.

4) The authors state: "Immunostaining revealed that whereas α 1-tubulin was present throughout the spindle apparatus, α 4-tubulin was enriched at the spindle poles (Figure S7A)."

Although I agree the α 4-tubulin appears somewhat enriched at the poles with respect to α 1-tubulin, a quantitation (with statistics) would be needed to support this claim. That being said, I agree the isotype is unlikely to account for the S3722C phenotype.

We will provide quantification of the α 1- and α 4-tubulin localization pattern in the full revision.

5) Trapping data show reduced stall force, yet increased stall time at low resistive forces for the mutant. This finding could potentially account for the reduced velocity of GFP-Rod noted in cells; however, I wonder if the authors noted altered velocity for dynein-driven bead movement under load in their trapping assays? This information would be useful to include in their manuscript.

Whilst the trapping experiments we performed were designed to extract stall forces, it is possible to generate force-velocity relationships from the data that allow us to report on velocities. These analyses indicate that S3386C dynein does indeed have reduced velocity under load compared to wild-type dynein (Figure R1). As we previously showed that S3386C does not perturb dynein velocity in the absence of load (Figure 8D), the new analysis reinforces the notion of a specific effect of the mutation on dynein performance when there is an opposing force. It also shows that reduced velocity of S-to-C mutant dynein under load could potentially account for the reduced velocity of GFP-Rod in mutant embryos, as the reviewer suggests. These observations increase our confidence that our *in vitro* observations with purified S-to-C mutant dynein are relevant for the *in vivo* phenotype. Whilst the force-velocity analysis was completed too recently to be included in the manuscript before the preliminary revision deadline, we will incorporate it in the full revision. We thank the reviewer for leading us to perform this informative analysis.

Revision Plan

Figure R1. Force-velocity plots for purified wild-type (WT) and S3386C dynein extracted from optical trapping data with dynein-dynactin-BICD2N complexes. All traces were recorded at 5 kHz. For both WT and S3386C dynein, traces containing stalls were concatenated. Custom MATLAB software was used to apply a median filter with a window size of 200 points. Subsequently, all events contained within 1 pN wide bins centered on 2, 3, 4, and 5 pN were identified. By dividing the change in force in each event by the typical spring constant used to measure the stall force (0.06 pN/nm), we were able to estimate the average velocity. To prevent the inclusion of detachment events, where dynein is not engaged with the axoneme, only non-negative average velocities were recorded. Circles are mean values and errors are S.E.M. Analysis is based on 350 and 466 trajectories for WT and S3386C dynein, respectively.

6) Is there a defocused spindle pole phenotype in the mutant cells? The cells in Video 1 and Fig. S6c appear to show as much, although other cells do not.

A fraction of spindles in the mutant embryos do indeed appear to have defocused spindle poles, which is perhaps not surprising given the defects in centrosome number and positioning that we reported. As we mentioned in the first submission, other defects in the spindles could conceivably be an indirect consequence of the metaphase arrest caused by altered dynein function at the kinetochore. We intend to quantify the penetrance of the defocused spindle pole phenotype and include the information in the full submission.

****Referees cross-commenting****

I wanted to reiterate my skepticism regarding the possibility that their data strongly support a SAC-independent role for dynein in the metaphase-to-anaphase transition (it seems Reviewer #3 might agree with me). I don't think it's impossible, but I'm not convinced they've made a very strong case for this model, which is noted in the title. The fact that proteins are accumulating at aligned kinetochores in the mutant cells (e.g., Rod and DLIC) in fact are consistent with a SAC silencing defect. Along these lines, I think reviewer #1's point regarding RZZ is a good one (that the mutant dynein is incapable of evicting RZZ specifically from aligned KTs), and should be tested prior to publication. My primary concern is that their conclusion is based entirely on the fact that the Mad2

Revision Plan

mutant does not fully restore mitotic exit to the dynein mutant cells. Given the cell-specificity of their dynein phenotype (in embryonic cells, but not L3 neuroblasts), I think testing the penetrance of their *Mad2* mutant in the embryonic cells would need to be assessed. In my review I suggested nocadazole, when I realized I meant to say reversine (oops!).

We appreciate that a SAC-independent role of dynein in anaphase progression is a surprising concept and therefore warrants further investigation. As described in our response to the point of Reviewer 1 about RZZ stripping, we will perform the suggested phenotypic analysis of *Dhc [S3372C]* in the presence of the *Rod[Z3]* allele (which has been shown by the Karess group to prevent association of RZZ components with the kinetochore in embryos and inactivate the SAC). We will also assess the effect of the *Mad2[P]* allele on *Mad2* expression in embryos to determine if this is a null allele, as described above. In response to a request by Reviewer 3, we will additionally examine localisation of the RZZ complex at kinetochores of *Dhc [S3372C] Mad2[P]* double mutant embryos. Like Reviewer 1, it is our understanding that the SAC inhibitor reversine does not work in *Drosophila* and therefore we do not plan to go down this route. Nonetheless, we expect the other experiments suggested by the reviewers will provide valuable information about the likelihood of SAC involvement in the metaphase arrest phenotype in S3372C embryos. We will, if necessary, revise or tone down our model about a SAC-independent dynein function based on the results, although our preliminary results with *Dhc [S3372C] Rod[Z3]* double mutants lend support to our initial model (see above).

Reviewer 3:

The manuscript is generally well written and conveys the major conclusions clearly and concisely. The data is generally of high quality, and largely supports the main conclusions, although there is one set of relatively straight-forward experiments that I think would be an important addition (see major comment #1, below).

Major Comments:

1. The observation that the S3372C mutation causes a mitotic arrest that is not SAC dependent (i.e. it still largely occurs even in a *Mad2* mutant background) is very surprising, and is the basis of the authors claim of a new, DHC-dependent, mechanism that allows embryo spindles to progress into anaphase.

As described above, we will further scrutinise the idea that the metaphase arrest in S3372C mutants is SAC-independent by examining the phenotype of *Dhc [S3372C] Rod[Z3]* double mutant embryos. We will also assess the effect of the *Mad2[P]* mutation on *Mad2* protein expression in embryos to determine if this is a null allele in this system, as previously concluded (Défachelles et al., 2015).

I think it would be important to assess whether the SAC components are still localising to the kinetochores in these S3372C, *Mad2* double mutants (e.g. is *Rod* still recruited to high-levels in

Revision Plan

the double mutant?). If the SAC components are still being recruited to the spindle (suggesting that they are still detecting that the spindle is not ready to go into anaphase), is it worth considering that Mad2 may not be essential for SAC function in these embryos?

We do not think that retention of RZZ components at the kinetochores of *Dhc [S3372C] Mad2[P]* double mutant embryos necessarily implies that the SAC is still active (an idea that would be heretical if indeed this is a null condition for Mad2). An alternative explanation is that S3372C mutation perturbs both dynein's function in stripping RZZ components from the kinetochore and another role of the motor at the kinetochore that is also required for anaphase progression (a possibility we discussed in the previous submission). Nonetheless, we intend to perform the suggested experiment as it may help us narrow down the mode of action of the S3372C mutation. As we have found that currently available antibodies to RZZ components do not work for immunostaining, this will require a series of genetic crosses to introduce the GFP-Rod transgene into the *Dhc [S3372C] Mad2[P]* background.

I say this because I find it hard to imagine how any, presumably mechanical, failure at the kinetochore that leads to the improper metaphase/anaphase transition in the S3372C mutants, would signal to the rest of the spindle to not transition to anaphase if the SAC is truly inactivated. Do the authors think these embryos have a completely unrelated surveillance system that detects the S3372C-dependent error (whatever that is) and arrests the spindles specifically in embryos? Or is the error itself sufficient to cause a spindle-wide arrest, which seems improbable?

If necessary, we will adjust our model based on the outcome of the planned experiments. If, on the other hand, our data are still supportive of a SAC-independent role of dynein, we can offer further speculation about potential underlying mechanisms.

2. I was surprised the authors made no attempt to quantify the level of over-accumulation of Dlic (Figure 6) or Rod (Figure 7) (and the lack of over-accumulation in other regions of the spindle). The images are convincing, so I don't doubt that this is the case, but I think some sort of quantification would be useful and I don't think it would be hard to come up with a way to do this (even just drawing a ROI around the approximate areas of interest). It would also be interesting to know whether other proteins like Spc25 (Figure 6) and Cdc20 (Figure S6) are recruited to normal levels at kinetochores.

We thank the reviewer for this important point. Whilst the phenotypes are indeed very strong, we agree that providing quantification would improve this aspect of the study. We will therefore perform the suggested analysis.

Revision Plan

3. Description of the revisions that have already been incorporated in the transferred manuscript

Please insert a point-by-point reply describing the revisions that were already carried out and included in the transferred manuscript. If no revisions have been carried out yet, please leave this section empty.

Reviewer 2:

Minor comments:

2) Although the mean inter-KT distance was unchanged between WT and S3722C cells, the authors note that the deviation from the mean was higher. Could this simply be a consequence of more highly dynamic oscillations of KT pairs (similar to that seen with Hec1-S69D in DeLuca et al., JCB 2018)? More dynamic oscillations could potentially lead to more variable distances between KT pairs.

Unless we are missing something, more dynamic oscillation of kinetochore pairs is not expected to affect variability in interkinetochore distance. Nonetheless, we are glad the reviewer brought this point up as it made us realise that our language when discussing these data in the discussion was too strong. We have now adjusted this section (lines 609-612) so as not to imply that changes in interkinetochore distance can only result from changes in tension:

“For example, the increased variability in interkinetochore distance in S3372C embryos may reflect abnormal tension between these structures in the absence of optimal force generation by dynein, which might prevent APC/C activation through the complex series of signaling events that respond to chromosome biorientation...”

3) It is interesting that the S3722C/Mad2-mutant cells are enriched in telophase (Fig. 7G). Does this not suggest another arrest point for these cells?

This is another insightful comment, which we have addressed with the following change to the Discussion of the preliminary revision (lines 618-621):

“As the small subset of *Mad2* S3372C spindles that were not in metaphase tended to be in telophase, the possibility that a high load function of dynein is additionally required for the telophase-interphase transition should also be investigated.”

7) The authors state: "This may reflect the relatively short length of *Drosophila* neurons making them less sensitive to partially impaired cargo transport." Could the extent of the phenotypes also be related to the lifespan of the flies? Do any of the diseases caused by these mutations have late-onset in patients? I wonder if a subtle defect in dynein behavior might not manifest for numerous years due to only minor changes in motility?

Revision Plan

The mutations are associated with neurodevelopmental defects that unfortunately for the patients and their families typically present from a very young age. Nonetheless, the reviewer makes a very good point that the short lifespan of *Drosophila* might be a factor in the lack of an overt phenotype in heterozygous flies. We have now mentioned this possibility on line 497-499 of the preliminary revision and thank the reviewer for the suggestion.

“This may reflect the relatively short length of neurons, or short lifespan, of *Drosophila* making them less sensitive to partially impaired cargo transport.”

Reviewer 3

Minor comments:

1. In the Discussion the authors state: "Our discovery of a missense mutation that strongly affects nuclear divisions in the embryo without disrupting other dynein functions offers a unique tool to study the mitotic roles of the motor". This should be reworded, as it suggests that the mutation affects all mitotic functions of DHC, which is clearly not the case (and also applies only to the embryo).

We agree with this point and have modified the preliminary revision accordingly (lines 585-587):

“Our discovery of a missense mutation that strongly affects nuclear divisions in the embryo without disrupting other dynein functions offers a unique tool to study specific mitotic roles of the motor at this stage.”

2. I think it worth more explicitly stating that there is no evidence that the defects the S3372C mutation lead to in the *in vitro* reconstituted system are the cause of the *in vivo* defects observed in the embryo. The authors are careful not to directly claim this, and I agree with their assertion that this is the most "parsimonious explanation" for their data, but I'm sure they would agree that this is far from proven, and it might be worth emphasising this point a little more.

As the reviewer is aware, we had investigated the most plausible alternative scenarios – stage specific protein destabilisation or ectopic disulphide bonding, differential interaction with tubulin isoforms, and differential phosphorylation – and found no evidence in support of them. We therefore believe that the force-sensitivity of the S3372C mutation is a likely explanation for the specific *in vivo* phenotype. We felt that using the phrase “most parsimonious explanation” would, nonetheless, convey the message that we are not ruling out other possibilities for the mode of action of the mutation *in vivo*. However, to address the reviewer’s point we have made this explicit with the following addition to the preliminary revision (lines 555-557).

“Nonetheless, we cannot exclude the possibility that the MTBD mutation affects another aspect of dynein function that contributes to the *in vivo* phenotype.”

Revision Plan

Related to this point, we would like to point out that the full revision will include new analysis of the optical trapping data that shows the S-to-C mutation reduces the velocity of dynein under load (performed in response to a point of Reviewer 2; Figure R1). The fact that we see reduced velocity of both GFP-Rod in S3372C mutant embryos (Figure 7C-E) and mutant DDB complexes experiencing load *in vitro* increases our confidence that our observations with purified dynein are relevant for the phenotype in embryos. We will incorporate this point in the full submission.

4. Description of analyses that authors prefer not to carry out

Please include a point-by-point response explaining why some of the requested data or additional analyses might not be necessary or cannot be provided within the scope of a revision. This can be due to time or resource limitations or in case of disagreement about the necessity of such additional data given the scope of the study. Please leave empty if not applicable.

Reviewer 2 raised the idea of inhibiting the SAC in a different way to *Mad2* disruption to further probe the idea of a SAC-independent function of dynein. The reviewer suggested the use of the SAC inhibitor reversine. However, as pointed out by reviewer 1, this drug reportedly does not work in *Drosophila*. As described above, we will instead determine the effect of disrupting the SAC in *Dhc* [S3372C] embryos using the previously characterised *Rod*[Z3] allele.

October 24, 2023

Re: JCB manuscript #202310022T

Dr. Simon L Bullock
MRC Laboratory of Molecular Biology
Cell Biology
Francis Crick Avenue
Cambridge CB2 0QH
United Kingdom

Dear Dr. Bullock,

Thank you for submitting your manuscript entitled "A force-sensitive mutation reveals a SAC-independent role for dynein in anaphase progression". We concur with the reviewers that these observations on a novel role of dynein in the mitotic spindle provide an intriguing conceptual advance, however we share their concerns primarily over the claims that this role is independent of the SAC.

The proposed changes noted in your revision plan are all important to solidify these claims and strengthen the overall model set forth. In addition to these points, however, we feel that it would be important to confirm the effects of colchicine on wt and Mad2 mutant embryos as claimed in Defachelles et al. That work did not provide a detailed view of mitotic defects (see Fig 1D), which we feel would offer a clear assessment of SAC status in this setting. These data will provide needed support for your conclusions on SAC-independent effects of dynein.

Please note that papers are generally considered through only one revision cycle, so any revised manuscript will likely be either accepted or rejected. We will be looking for clear enthusiasm from the reviewers in order to proceed with this work after revision.

The typical timeframe for revisions is three to four months. While most universities and institutes have reopened labs and allowed researchers to begin working at nearly pre-pandemic levels, we at JCB realize that the lingering effects of the COVID-19 pandemic may still be impacting some aspects of your work, including the acquisition of equipment and reagents. Therefore, if you anticipate any difficulties in meeting this aforementioned revision time limit, please contact us and we can work with you to find an appropriate time frame for resubmission.

If you choose to revise and resubmit your manuscript, please also attend to the following editorial points. Please direct any editorial questions to the journal office.

GENERAL GUIDELINES:

Text limits: Character count is < 40,000, not including spaces. Count includes title page, abstract, introduction, results, discussion, and acknowledgments. Count does not include materials and methods, figure legends, references, tables, or supplemental legends.

Figures: Your manuscript may have up to 10 main text figures. To avoid delays in production, figures must be prepared according to the policies outlined in our Instructions to Authors, under Data Presentation, <https://jcb.rupress.org/site/misc/ifora.xhtml>. All figures in accepted manuscripts will be screened prior to publication.

Supplemental information: There are strict limits on the allowable amount of supplemental data. Your manuscript may have up to 5 supplemental figures. Up to 10 supplemental videos or flash animations are allowed. A summary of all supplemental material should appear at the end of the Materials and methods section.

Please note that JCB now requires authors to submit Source Data used to generate figures containing gels and Western blots with all revised manuscripts. This Source Data consists of fully uncropped and unprocessed images for each gel/blot displayed in the main and supplemental figures. Since your paper includes cropped gel and/or blot images, please be sure to provide one Source Data file for each figure that contains gels and/or blots along with your revised manuscript files. File names for Source Data figures should be alphanumeric without any spaces or special characters (i.e., SourceDataF#, where F# refers to the associated main figure number or SourceDataFS# for those associated with Supplementary figures). The lanes of the gels/blots should be labeled as they are in the associated figure, the place where cropping was applied should be marked (with a box), and molecular weight/size standards should be labeled wherever possible.

If you choose to resubmit, please include a cover letter addressing the reviewers' comments point by point. Please also highlight all changes in the text of the manuscript.

Regardless of how you choose to proceed, we hope that the comments below will prove constructive as your work progresses. We would be happy to discuss them further once you've had a chance to consider the points raised. You can contact the journal office with any questions at cellbio@rockefeller.edu.

Thank you for thinking of JCB as an appropriate place to publish your work.

Sincerely,

Hironori Funabiki
Monitoring Editor
Journal of Cell Biology

Tim Fessenden
Scientific Editor
Journal of Cell Biology

We are very grateful to the reviewers for their insightful and constructive feedback. We have now submitted a revised manuscript based on their comments and the revision plan agreed to by the editors at Journal of Cell Biology. We believe the manuscript has been improved significantly since the first submission, particularly through new experiments that strengthen the evidence for a role for dynein in anaphase progression that is in addition to its canonical function in SAC silencing. Please find below a point-by-point response to the reviewer comments, in which line numbers refer to the merged PDF generated by the manuscript submission system (rather than the Word document). To assist with evaluation of the changes, we have tracked significant changes to the text in blue.

Reviewer #1 (Evidence, reproducibility and clarity (Required)):

The authors start out by examining the cellular and organismal effects of 6 human disease-linked mutations, as well as the mouse legs-at-odd-angles (Loa) mutation, after introducing the mutations into the *D. melanogaster* dynein heavy chain (Dhc) by genome editing. This reveals an overall correlation between the severity of the effect on dynein motor activity in vitro (determined in a previous study) and the penetrance of the corresponding mutant phenotype in the fly, with a couple of interesting exceptions that illustrate the value of performing structure-function analysis of dynein in animal models. The authors then focus on an additional missense mutation in the Dhc microtubule binding domain, fortuitously generated by imprecise editing, that results in a striking phenotype. The S3372C mutation supports normal development, including normal axonal transport of mitochondria and asymmetric mitosis of larval neuroblasts, but female flies are infertile. Through elegant genetics, ectopic disulfide bond formation with a nearby residue is ruled out as the cause of the maternal effect. S3372C results in metaphase arrest of early embryonic divisions, characterized by over-accumulation of dynein light intermediate chain (Dlic) and the dynein recruitment factor Rod at kinetochores, as well as by reduced poleward streaming of Rod along spindle microtubules. Surprisingly, the S3372C-induced metaphase arrest cannot be bypassed by inhibiting the spindle assembly checkpoint, implying that dynein promotes the metaphase-to-anaphase transition not solely through its known function in spindle assembly checkpoint silencing. In vitro motility and optical trapping experiments show that the mutant motor performs normally in a load-free regime but exhibits reduced peak force production and excessive pausing under load. Furthermore, molecular dynamics simulations reveal how the mutation affects dynein's interaction with microtubules, including a change in the positioning of the stalk. The authors conclude that the S3372C mutation specifically perturbs high-load functions of dynein, explaining the selective phenotype observed in vivo.

The experiments are technically on a very high level, the results are presented in a clear manner, and the conclusions are fully supported by the data.

We are naturally very pleased that the reviewer is enthusiastic about the study.

Minor suggestions (optional):

- In the first part of the paper, where Dhc mutations associated with neurological disease are examined, the H3808P and F579/Loa mutations are shown to cause mis-accumulation of synaptic vesicles in axons. The authors may want to perform this assay for the K129I, R1557Q, and K3226T mutations, as this would strengthen the comparative analysis of in vitro versus in vivo effects, summarized in Figure 1C. For example, K129I has a more severe effect in vitro than the Loa mutation, but the Loa mutation has a more pronounced phenotype on the organismal level. Would this also be the case in a cell-based assay?

We have now investigated the effects of the K129I, R1557Q and K3226T mutations on synaptic vesicle distribution in motor neuron axons of L3 larvae, as suggested. We did not observe any axonal accumulations of synaptic vesicles in these mutants, consistent with the

lack of organismal defects. The results are now incorporated in Fig. 1, C and E and described on line 130–132 of the manuscript. We thank the reviewer for leading us to make the analysis of the disease-associated mutations more comprehensive.

- The observation that the metaphase arrest of S3372C mutant embryos cannot be alleviated by the checkpoint-defective Mad2 mutant is very intriguing, as is the observation that Dlic and the RZZ subunit Rod over-accumulate at/near kinetochores. As discussed by the authors, one possibility is that the arrest is a consequence of dynein's failure to disassemble the corona by stripping, but, surprisingly, in a manner unrelated to dynein's role in SAC silencing. In this regard, it is interesting to note that fly RZZ mutants do not undergo metaphase arrest in the early embryo (Williams and Goldberg, 1994; Défachelles et al., 2015), whereas knockdown of Spindly, which functions in dynein recruitment downstream of RZZ, does lead to arrest (see Figure 2 in Clemente et al., 2018; PMID 29615558). Taken together, this raises the possibility that it is the failure to remove RZZ (and other associated corona components) from kinetochores that inhibits anaphase onset in S3372C embryos. It would therefore be interesting to test whether the metaphase arrest in S3372C embryos is alleviated in RZZ mutants.

This is a very good suggestion. To address this point, we have used recombination to generate a chromosome containing both *Dhc* [S3372C] and a maternal effect Rod allele (*Z3*), which has been shown to prevent association of the RZZ complex with the kinetochore and inactivate the SAC in embryos (Défachelles et al., 2015). Analysis of the *rod* [*Z3*] *Dhc* [S3372C] double mutant embryos revealed that the Rod mutation does not rescue the metaphase arrest associated with the dynein mutation (Fig. S3, G and H). This result supports our initial proposal that dynein has an essential function in anaphase progression in addition to RZZ stripping and SAC silencing. This result has been added to line 362–367 of the manuscript and incorporated in a more balanced discussion of the evidence for a non-canonical dynein role on line 537–547.

****Referees cross-commenting****

The Mad2 mutant the authors use is a P-element insertion that was described by Buffin et al 2007 as a null mutant with regards to SAC signaling (it also does not produce any detectable protein by Western blot; Figure 1b). Nevertheless, since the analysis in Buffin et al was restricted to larval brains, I agree with reviewer #2 that it remains to be formally demonstrated that this Mad2 mutant fully abolishes the SAC in the early embryo. Unfortunately, as far as I am aware, reversine does not work well in *Drosophila*. An alternative would be to combine *Dhc*(S3372C) with the other Mad2 mutant used by Buffin et al, which (besides not producing detectable protein) lacks the Mad1 binding domain and can therefore be expected to be a definitive checkpoint null in all tissues.

The alternative *mad2* allele referred to by this reviewer (*mad2* [Δ]; Buffin et al., 2007) also has a deletion of two other genes, *S6k* and *kri*, which encode ribosomal protein S6 kinase and a uracil phosphoribosyltransferase, respectively. Because these mutations would confound phenotypic analysis, Buffin et al. introduced a rescue construct for both *S6k* and *kri* onto the *mad2* [Δ] chromosome. It would be challenging to make a recombinant chromosome containing *Dhc* [S3372C], *mad2* [Δ] and a *S6k/kri* rescue construct. We therefore elected to first analyse Mad2 protein levels in embryos homozygous for the P-element allele via immunoblotting. We were unable to detect any Mad2 in the mutant embryos, whilst a strong signal was detected in the wild-type control (Fig. 8 A). This observation indicates that the P-element mutation is a protein null in embryos. A null status of the *mad2* [*P*] allele is in keeping with the observation that it phenocopies the aforementioned *mad2* deletion allele in larval stages (in the presence of the *S6k/kri* rescue construct) (Buffin et al., 2007).

Consistent with the absence of Mad2 protein, two other groups have concluded that the SAC is defective in *mad2 [P]* embryos (Défachelles et al., 2015; Yuan and O'Farrell, 2015). This was based on the results of treatments with the microtubule targeting drug colchicine. Regrettably, we did not make this clear in the first submission but now do so (line 347–351). Défachelles et al. delivered colchicine to *mad2 [P]* embryos via permeabilization of the vitelline membrane with heptane, whereas Yuan and O'Farrell use injection as the colchicine delivery method. Both studies reported that *mad2 [P]* allows spindles to overcome the metaphase arrest caused by microtubule depolymerization. Yuan and O'Farrell's images and time-lapse movies indicated that the escape of colchicine-injected *mad2 [P]* embryos was highly penetrant, with all mitotic figures able to decondense their chromosomes and exit mitosis in a few minutes.

In response to a comment from Reviewer 2 and a specific request of the editors, we have now repeated colchicine treatments in control and *mad2 [P]* embryos in order to quantify the frequency of mitotic progression. We found that whereas 96% of nuclei in wild-type embryos injected with colchicine were in metaphase, 95.7% of those in injected *mad2[P]* embryos exited mitosis (Fig. 8 C). The high frequency of mitotic progression observed in the *mad2[P]* embryos corroborates the conclusion of the previous studies that the mutation results in an inactive SAC.

We also performed an additional experiment, which was not requested by the reviewers, in which we injected colchicine into *mad2 [P] Dhc [S3372C]* embryos (Fig. 8 E). Like *S3372C* mutants, the double mutant embryos exhibited a highly penetrant metaphase arrest following colchicine injection. Thus, even in a situation in which there is a strong requirement for the SAC in preventing mitotic progression, Mad2 disruption is not sufficient to relieve the *S3372C*-induced mitotic block.

Together with the results from the *rod [Z3] Dhc [S3372C]* embryos described above, we believe these analyses significantly strengthen the case that dynein has a function in anaphase progression that is independent from its role in SAC silencing. Nonetheless, prompted by the reviewers' comments in the original submission, we now propose (rather than conclude) that there is a SAC-independent dynein function and discuss an alternative (albeit unlikely) explanation for our observations in the *mad2 [P] Dhc [S3372C]* embryos (that the dynein mutation renders Mad2 dispensable for the SAC; line 541–543). We have also adjusted the language in the title and abstract accordingly.

In summary, we have both strengthened the evidence for a non-canonical function for dynein in anaphase progression and provided a more balanced discussion of our findings. We believe these changes have improved the study significantly.

Reviewer #1 (Significance (Required)):

The cytoplasmic dynein 1 motor complex participates in a multitude of cellular processes that require microtubule minus-end-directed motility. Whereas in vitro reconstitution efforts have led to a detailed understanding of the motor's motile properties, the essential requirement of dynein for development has made it challenging to dissect how the motor contributes to specific aspects of intracellular organization and cell division in vivo. The need for a mechanistic understanding of how dynein motility is used for diverse functions in vivo is underscored by missense mutations in the motor subunit that cause human neurological disease. In this interesting and insightful study, Salvador-Garcia and colleagues characterize several missense mutations in dynein heavy chain (Dhc) using biochemical assays and genetic approaches in the fly, which reveals the distinct effects of disease-causing mutations in vivo and uncovers an unanticipated novel function of dynein in regulating mitotic progression.

This beautifully executed study has important implications for dynein's role in mitosis, in particular its role at the kinetochore, and is of broad interest to cell biologists studying the cytoskeleton, as it demonstrates that examining motor mutants with altered mechanical properties *in vivo* can reveal specific motor functions.

We thank the reviewer for the positive evaluation of the study's significance.

Reviewer #2 (Evidence, reproducibility and clarity (Required)):

In the manuscript by Salvador-Garcia et al., the authors assess the physiological consequences of dynein mutations in flies and *in vitro*. In addition to characterizing the manner by which disease causing mutations affect fly development and some aspects of their cell physiology, the authors focus on sporadic mutations that arose during the course of generating their mutant fly lines. Of particular interest was a mutation in the dynein MTBD: S3722C. This mutation caused very interesting phenotypes in flies (e.g., infertility in females likely due to mitotic arrest) as well as in reconstituted motility assays (e.g., reduced stall force). The authors posit that the mitotic arrest phenotype is a consequence of a dynein's role in initiating anaphase onset, and that this role is distinct from its well established role in silencing the spindle assembly checkpoint.

The paper is very well written, and the data are of high quality. Most of the claims - with one major exception (described below) - are well supported by the data. I have a few comments that might help to strengthen the conclusions.

We thank the reviewer for their positive comments and valuable suggestions for improving the manuscript.

Major comment:

1) In brief, I'm not convinced the mitotic arrest phenotype is not a consequence of impaired SAC silencing by the mutant dynein. The main tool the authors use to support their claim is a Mad2 mutant. They use this to determine if preventing SAC function (with the mutant Mad2) can override the ability of S3722C cells to progress into anaphase. The Mad2 mutant does in fact increase the proportion of cells exiting mitosis (from 0.8 to 14% of cells); however, the low number (14%) suggests that an inability to silence the SAC is not the reason the cells are not entering anaphase (i.e., it is SAC silencing-independent). My question is how penetrant the Mad2 mutant is? For example, how many cells with this Mad2 mutant would exit mitosis if the authors perturbed mitosis some other way (e.g., treatment with high concentrations of nocodazole)? If the number is still low (~14%), then this might be why the mutant can't rescue the mitotic exit phenotype for S3722C cells, and would challenge the following statements: "...it suggests that this can make, at best, a minor contribution to the mitotic arrest phenotype"; and "Remarkably, the MTBD mutation does not appear to block anaphase progression in embryos by preventing the well-characterized role of kinetochore-associated dynein in silencing the SAC, as the defect persists when the checkpoint is inactivated by mutation of Mad2. Collectively, these observations indicate that kinetochore dynein has a novel role in licensing the transition from metaphase to anaphase." Although previous studies might have assessed the penetrance of this mutant in other cell types, given the cell specificity of the mitotic arrest phenotype for S3722C (in embryonic cells, but not L3 neuroblasts), it will be important to provide such additional evidence in embryonic cells (e.g., nocodazole treatment of embryonic cells) to support these statements, especially in light of the bold conclusions and hypotheses they are making (e.g., "For example, the apparent variability in tension between sister kinetochores in S3372C embryos, which could reflect abnormal force generation by the mutant motor complex, might prevent APC/C activation through the complex series of signaling events that respond to chromosome

bioorientation."). Although it would be fascinating if the authors are correct that dynein provides another role in licensing anaphase onset, the well-established role for dynein in checkpoint silencing currently seems like the most parsimonious explanation.

We fully appreciate the reviewer's point that a SAC-independent role for dynein would be surprising and fascinating and therefore more evidence supporting this claim needs to be presented. Regrettably, we did not make it explicit in the first submission that two other groups have analyzed the effect of the *mad2* allele we used (*mad2* [P]) on the status of the SAC in early embryos. Both studies concluded that the SAC was defective in *mad2* [P] embryos (now stated on line 347–351). Défachelles et al. (2015) delivered the microtubule-targeting agent colchicine (which has an analogous effect to nocodazole) to *mad2* [P] embryos via permeabilization of the vitelline membrane with heptane, whereas Yuan and O'Farrell (2015) used injection as the colchicine delivery method. Both studies reported that *mad2* [P] allows embryos to overcome the metaphase arrest caused by microtubule depolymerization. Yuan and O'Farrell's images and time-lapse movies indicated that the escape of colchicine-injected *mad2* [P] embryos was highly penetrant, with all mitotic figures able to decondense their chromosomes and exit mitosis in a few minutes.

In response to the point of this reviewer and a specific request of the editors, we have now repeated colchicine treatments in control and *mad2* [P] embryos in order to quantify the penetrance of mitotic progression. We found that whereas 96% of nuclei in wild-type embryos injected with colchicine were in metaphase, 95.7% of those in injected *mad2* [P] embryos exited mitosis (Fig. 8 C). The high frequency of mitotic progression observed in the *mad2* [P] embryos corroborates the conclusion of the previous studies that the mutation results in an inactive SAC.

Following a comment from Reviewer 1, we used immunoblotting to show that Mad2 protein is undetectable in *mad2* [P] embryos (Fig. 8 A). This finding is consistent with the highly penetrant effect of *mad2* [P] following colchicine treatment. We also performed an additional experiment, which was not requested by the reviewers, in which we injected colchicine into *mad2* [P] *Dhc* [S3372C] double mutant embryos (Fig. 8 E). Like S3372C embryos, the double mutant embryos exhibited a highly penetrant metaphase arrest following colchicine injection. Thus, even in a situation in which there is a clear requirement for the SAC in preventing mitotic progression, Mad2 disruption is not sufficient to relieve the S3372C-induced metaphase arrest.

Furthermore, in response to a comment of Reviewer 1, we have asked if a maternal effect Rod mutation that prevents RZZ's kinetochore association and inactivates the SAC (Défachelles et al., 2015) is sufficient to prevent mitotic arrest caused by *Dhc* [S3372C]. We find that it is not (Fig. S3, G and H), strengthening the case for a role for dynein in promoting anaphase that is in addition to its canonical role in RZZ stripping and SAC silencing.

Minor comments:

1) The S3722C mutant appears to accumulate to higher-than-normal levels at KT's and to some extent along the spindle MTs. In addition to the representative images, it would be helpful to see a quantitation of this phenomenon for WT, S3722C, and S3722C/+ along with statistics.

This is another good point. We have added quantification and statistics for this experiment (Fig. 6 B).

2) Although the mean inter-KT distance was unchanged between WT and S3722C cells, the authors note that the deviation from the mean was higher. Could this simply be a consequence of more highly dynamic oscillations of KT pairs (similar to that seen with Hec1-

S69D in DeLuca et al., JCB 2018)? More dynamic oscillations could potentially lead to more variable distances between KT pairs.

Unless we are missing something, more dynamic oscillation of kinetochore pairs is not expected to affect variability in interkinetochore distance. Nonetheless, we are glad the reviewer raised this point as it made us realize that our language when discussing these data was too strong. We have now adjusted this section (line 552–554) so as not to imply that changes in interkinetochore distance can only result from changes in tension:

“For example, the increased variability in interkinetochore distance in S3372C embryos may reflect abnormal tension between these structures as a result of sub-optimal force generation by dynein...”

3) It is interesting that the S3722C/Mad2-mutant cells are enriched in telophase (Fig. 7G). Does this not suggest another arrest point for these cells?

Although the frequency of the metaphase arrest is very similar (and consistently very high) between experiments, the breakdown of mitotic stages for the remaining nuclei seems somewhat variable. For example, in our new experiments, we did not see an enrichment of telophase in the non-metaphase stages for controls in which 1% DMSO (a vehicle that we expect to be neutral) was injected into *mad2 [P] Dhc [S3372C]* embryos (Fig. 8 E). Furthermore, there was a higher proportion of telophase stages in a new cohort of untreated single *S3372C* mutants (Fig. S3 H) than there was in the equivalent experiment in the first submission (now Fig. 8 D). Therefore, we do not feel it is appropriate to speculate about a later arrest point.

4) The authors state: "Immunostaining revealed that whereas $\alpha 1$ -tubulin was present throughout the spindle apparatus, $\alpha 4$ -tubulin was enriched at the spindle poles (Figure S7A)."

Although I agree the $\alpha 4$ -tubulin appears somewhat enriched at the poles with respect to $\alpha 1$ -tubulin, a quantitation (with statistics) would be needed to support this claim. That being said, I agree the isotype is unlikely to account for the S3722C phenotype.

We have now provided quantification for the localization of the two tubulin isotypes (Fig. S4 A).

5) Trapping data show reduced stall force, yet increased stall time at low resistive forces for the mutant. This finding could potentially account for the reduced velocity of GFP-Rod noted in cells; however, I wonder if the authors noted altered velocity for dynein-driven bead movement under load in their trapping assays? This information would be useful to include in their manuscript.

Whilst the trapping experiments we performed were designed to extract stall forces, it is possible to generate force-velocity relationships from the data. We include the new analysis in Fig. 10 D (results summarized on line 407). These data reveal that S3386C dynein does indeed have reduced velocity under load compared to wild-type dynein. As we previously showed that S3386C does not perturb dynein velocity in the absence of load (now Fig. 9 D and Fig. S5 C), the new analysis reinforces the notion of a specific effect of the mutation on dynein performance when there is an opposing force. It also shows that reduced velocity of S-to-C mutant dynein under load could potentially account for the reduced velocity of GFP-Rod on mutant spindles, as the reviewer suggests (now mentioned on line 500–502). We believe these observations strengthen the case that our *in vitro* observations with purified S-to-C mutant dynein are relevant for the *in vivo* phenotype. We thank the reviewer for leading us to perform this informative analysis.

6) Is there a defocused spindle pole phenotype in the mutant cells? The cells in Video 1 and Fig. S6c appear to show as much, although other cells do not.

Whilst the mutant spindles have broader spindle poles than controls, leading to a rounder overall spindle shape (now quantified in Fig. S2 F), the poles appear to be focused. These features of spindle morphology have been seen previously when centrosome number is impaired (Wakefield et al., 2000). We now mention these observations on line 259–263.

7) The authors state: "This may reflect the relatively short length of *Drosophila* neurons making them less sensitive to partially impaired cargo transport." Could the extent of the phenotypes also be related to the lifespan of the flies? Do any of the diseases caused by these mutations have late-onset in patients? I wonder if a subtle defect in dynein behavior might not manifest for numerous years due to only minor changes in motility?

The mutations are associated with neurodevelopmental defects that, unfortunately for the patients and their families, typically present from a very young age. Nonetheless, the reviewer makes a very interesting point that the short lifespan of *Drosophila* might be a factor in the lack of an overt phenotype in heterozygous flies. We have now mentioned this possibility on line 450–452 of the revised manuscript.

"This may reflect Drosophila's relatively short neurons, or its short lifespan, reducing sensitivity to partially impaired cargo transport."

****Referees cross-commenting****

I wanted to reiterate my skepticism regarding the possibility that their data strongly support a SAC-independent role for dynein in the metaphase-to-anaphase transition (it seems Reviewer #3 might agree with me). I don't think it's impossible, but I'm not convinced they've made a very strong case for this model, which is noted in the title. The fact that proteins are accumulating at aligned kinetochores in the mutant cells (e.g., Rod and DLIC) in fact are consistent with a SAC silencing defect. Along these lines, I think reviewer #1's point regarding RZZ is a good one (that the mutant dynein is incapable of evicting RZZ specifically from aligned KTs), and should be tested prior to publication. My primary concern is that their conclusion is based entirely on the fact that the Mad2 mutant does not fully restore mitotic exit to the dynein mutant cells. Given the cell-specificity of their dynein phenotype (in embryonic cells, but not L3 neuroblasts), I think testing the penetrance of their Mad2 mutant in the embryonic cells would need to be assessed. In my review I suggested nocadazole, when I realized I meant to say reversine (oops!).

We thank the reviewer for encouraging us to address these issues. As Reviewer 1 points out, reversine does not appear to be active in *Drosophila*. Nonetheless, we believe the changes to the manuscript described above (including demonstrating the penetrance of the *mad2* mutation in embryos) strengthen significantly the evidence for a role for dynein in anaphase progression that is independent from its canonical role in SAC silencing. However, in light of the reviewers' previous comments, we have chosen to soften the language associated with this part of the manuscript, describing the SAC-independent function as a proposal (rather than a conclusion) (e.g. line 541–547) and adjusting the language in the title and abstract accordingly.

In summary, we have both strengthened the evidence for a non-canonical dynein function and provided a more balanced discussion of our findings. We believe these changes have improved the study significantly.

Reviewer #2 (Significance (Required)):

The manuscript by Salvador-Garcia et al. is a very interesting study dissecting the physiological consequences of dynein mutations in flies and in vitro. This study will be of high interest to those in the dynein/molecular motor field, as well as those that study mitosis and kinetochore function. One of the most interesting findings in the study is the identification of a mutation in dynein that specifically impacts its motility in conditions of high load. This mutant provides a novel tool to dissect load-dependent transport for dynein in other systems. Moreover, the study suggests a novel role for dynein in promoting anaphase onset; if the authors can provide additional support for this claim, the impact of this study would be greater.

Field of expertise: molecular motors, kinetochore function

We thank the reviewer for the positive evaluation.

Reviewer #3 (Evidence, reproducibility and clarity (Required)):

Summary

In this manuscript Salvador-Garcia et al. examine how several mutations in the Dynein heavy chain (DHC) influence Dynein function in an in vitro reconstituted system and in *Drosophila* embryos. Most importantly, they identify a novel substitution mutation (S3372C, generated as a by-product of a targeted CRISPR mutagenesis screen) that leads to a novel phenotype specifically in early *Drosophila* embryos (metaphase arrest) and that only impairs DHC function under load in vitro. Most surprisingly, the metaphase arrest in embryos does not appear to be due to a failure to inactivate the spindle-assembly-checkpoint (SAC), a known DHC function. This suggests that DHC has a hitherto unappreciated function in allowing spindles in the *Drosophila* embryo to progress from metaphase-to-anaphase.

The manuscript is generally well written and conveys the major conclusions clearly and concisely. The data is generally of high quality, and largely supports the main conclusions, although there is one set of relatively straight-forward experiments that I think would be an important addition (see major comment #1, below).

We thank the reviewer for their positive assessment and helpful suggestions for improving the manuscript.

Major Comments

1. The observation that the S3372C mutation causes a mitotic arrest that is not SAC dependent (i.e. it still largely occurs even in a Mad2 mutant background) is very surprising, and is the basis of the authors claim of a new, DHC-dependent, mechanism that allows embryo spindles to progress into anaphase. I think it would be important to assess whether the SAC components are still localising to the kinetochores in these S3372C, Mad2 double mutants (e.g. is Rod still recruited to high-levels in the double mutant?). If the SAC components are still being recruited to the spindle (suggesting that they are still detecting that the spindle is not ready to go into anaphase), is it worth considering that Mad2 may not be essential for SAC function in these embryos?

Following the comments of this reviewer and the other two reviewers, we have now performed new experiments to check if Mad2 is indeed required for SAC function in embryos. Firstly, we have shown that there is no detectable Mad2 protein in the *mad2 [P]* mutant embryos (Fig. 8 A). This finding is consistent with previous evidence from larval

brains that this is null mutation (Buffin et al., 2007; Gallaud et al., 2022). Secondly, we have repeated the colchicine treatments performed independently by the groups of Roger Karess and Pat O'Farrell in embryos (Défachelles et al., 2015; Yuan and O'Farrell, 2015) and corroborated their conclusion that the SAC is inactive in *mad2*[*P*] embryos. As now shown in Fig. 8 C, whereas 96% of nuclei in wild-type embryos injected with colchicine were in metaphase, 95.7% of those in injected *mad2* [*P*] embryos exited mitosis (Fig. 8 C). We also performed an additional experiment, which was not requested by the reviewers, in which we injected colchicine into *mad2* [*P*] *Dhc* [*S3372C*] embryos (Fig. 8, E). Like *S3372C* embryos, the double mutant embryos had a highly penetrant metaphase arrest following colchicine injection. Thus, even in a situation in which there is a clear requirement for the SAC in preventing mitotic progression, Mad2 disruption is not sufficient to relieve the *S3372C*-induced metaphase arrest. We cannot rule out that Mad2 is not required for the SAC specifically in *S3372C* embryos but we have difficulty conceiving how this could be the case. Nonetheless, we now mention this possibility on line 541–542 as part of our efforts to provide more balance to our interpretations about a SAC-independent effect of *S3372C* (see below).

We have been unable to assess levels of endogenous SAC components at the kinetochore in the *mad2* [*P*] *S3372C* embryos due to the available antibodies not working sufficiently well for immunofluorescence. We therefore crossed the GFP-Rod transgene into the *mad2* [*P*] *Dhc* [*S3372C*] background. Unfortunately, Rod overexpression in this genotype arrests development before mitotic divisions so we were unable to reach a conclusion about localization of Rod at the kinetochore. However, the result of an experiment suggested by Reviewer 1 addresses the involvement of Rod in the *S3372C* mutant phenotype by another means. We now show that the metaphase arrest caused by *Dhc* [*S3372C*] is not overcome by a maternal effect allele of Rod that prevents recruitment of RZZ to the kinetochore and impairs the SAC in embryos (Défachelles et al., 2015) (Fig. S3, G and H). This finding provides further evidence that dynein has a role in anaphase progression that is independent from the motor's canonical role in RZZ stripping and SAC silencing.

Nonetheless, we also chosen to soften the language associated with this part of the manuscript, describing the SAC-independent function as a proposal (rather than a conclusion) and adjusting the language in the title and abstract accordingly.

In summary, we have both strengthened the evidence that dynein does not only contribute to anaphase progression through its canonical role in RZZ stripping and SAC silencing and provided a more balanced discussion of our findings. We believe these changes have improved the study significantly.

I say this because I find it hard to imagine how any, presumably mechanical, failure at the kinetochore that leads to the improper metaphase/anaphase transition in the *S3372C* mutants, would signal to the rest of the spindle to not transition to anaphase if the SAC is truly inactivated. Do the authors think these embryos have a completely unrelated surveillance system that detects the *S3372C*-dependent error (whatever that is) and arrests the spindles specifically in embryos? Or is the error itself sufficient to cause a spindle-wide arrest, which seems improbable?

We agree it seems unlikely that a purely mechanical arrest in the *S3372C* mutants would be sufficient to prevent exit from metaphase when the SAC is inactive. The results of the new experiment in which we injected colchicine into *mad2* [*P*] *Dhc* [*S3372C*] double mutant embryos also argue against such a scenario. Instead, we favour the possibility that the MTBD mutation impairs other events that lead to APC/C activation or events downstream of APC/C activation (line 552–558). However, because resolving this issue will require a separate long-term study (and in light of a comment by Reviewer 2 that our hypotheses are very strong), we would prefer not to go further with our speculation at this stage.

2. I was surprised the authors made no attempt to quantify the level of over-accumulation of Dlic (Figure 6) or Rod (Figure 7) (and the lack of over-accumulation in other regions of the spindle). The images are convincing, so I don't doubt that this is the case, but I think some sort of quantification would be useful and I don't think it would be hard to come up with a way to do this (even just drawing a ROI around the approximate areas of interest). It would also be interesting to know whether other proteins like Spc25 (Figure 6) and Cdc20 (Figure S6) are recruited to normal levels at kinetochores.

We have added these quantifications (Fig. 6 B; and Fig. S3 C), adding a brief discussion of the new findings to the manuscript where necessary (e.g. line 309–312).

Minor Comments

1. In the Discussion the authors state: "Our discovery of a missense mutation that strongly affects nuclear divisions in the embryo without disrupting other dynein functions offers a unique tool to study the mitotic roles of the motor". This should be reworded, as it suggests that the mutation affects all mitotic functions of DHC, which is clearly not the case (and also applies only to the embryo).

We agree with this point and have modified the preliminary revision accordingly (line 521–523):

"Our discovery of a missense mutation that causes a highly penetrant metaphase arrest in the embryo without disrupting other dynein functions offers a unique tool to study the role of the motor in anaphase progression."

2. I think it worth more explicitly stating that there is no evidence that the defects the S3372C mutation lead to in the in vitro reconstituted system are the cause of the in vivo defects observed in the embryo. The authors are careful not to directly claim this, and I agree with their assertion that this is the most "parsimonious explanation" for their data, but I'm sure they would agree that this is far from proven, and it might be worth emphasising this point a little more.

As the reviewer is aware, we had investigated the most plausible alternative scenarios—stage specific protein destabilization or ectopic disulphide bonding, differential interaction with tubulin isoforms, and differential phosphorylation—and found no evidence in support of them. We therefore believe that the force sensitivity of the S3372C mutation is a likely explanation for the specific in vivo phenotype. We felt that using the phrase "most parsimonious explanation" would convey the message that we are not ruling out other possibilities for the mode of action of the mutation in vivo. However, to address the reviewer's point we have made this explicit with the following addition to the preliminary revision (line 497–500).

"Thus, whilst we cannot rule out the mutation affecting another aspect of dynein function that is important for anaphase progression, the most parsimonious explanation for the restricted phenotype is that the metaphase to anaphase transition in the embryo needs dynein to work in a specific load regime that is problematic for the mutant motor complex."

Related to this point, the revision now includes new analysis of the optical trapping data showing that S-to-C mutation reduces the velocity of dynein under load (Fig. 10 D; performed in response to a comment of Reviewer 2). The fact that we see reduced velocity of both GFP-Rod in S3372C mutant embryos (Fig. 7, D and E) and mutant DDB complexes experiencing load in vitro increases our confidence that our observations with purified dynein

are relevant for the phenotype in embryos. To make this point, the following sentence has been added to the one quoted above (line 500–502):

“Consistent with this notion, we found that the slower movement of purified dynein under load was mirrored by slower transport of GFP-Rod away from kinetochores in vivo.”

Reviewer #3 (Significance (Required)):

This is a well conducted study that significantly extends the author's previous work on how mutations in DHC (initially indentified in human patients) effect DHC function (Hoang et al., PNAS, 2017). The paper reports the striking central finding that the S3372C mutation produces a very unusual mitotic arrest phenotype specifically in *Drosophila* embryos, and the authors link this to the also striking finding that this mutation only disrupts DHC function in vitro when DHC is working under load. As mentioned above, this link is not proven here, but this is a solid working hypothesis that is potentially of significant interest to those working on molecular motors and their role in fundamental cell biology and human disease.

I am a cell biologist with expertise in the cytoskeleton, particularly during early *Drosophila* embryogenesis.

We thank the reviewer for the positive evaluation.

References:

- Buffin, E., D. Emre, and R.E. Karess. 2007. Flies without a spindle checkpoint. *Nat Cell Biol.* 9:565-72. <https://doi.org/10.1038/ncb1570>.
- Défachelles, L., S.G. Hainline, A. Menant, L.A. Lee, and R.E. Karess. 2015. A maternal effect rough deal mutation suggests that multiple pathways regulate *Drosophila* RZZ kinetochore recruitment. *J Cell Sci.* 128:1204-16. <https://doi.org/10.1242/jcs.165712>.
- Gallaud, E., L. Richard-Parpaillon, L. Bataille, A. Pascal, M. Metivier, V. Archambault, and R. Giet. 2022. The spindle assembly checkpoint and the spatial activation of Polo kinase determine the duration of cell division and prevent tumor formation. *PLoS Genet.* 18:e1010145. <https://doi.org/10.1371/journal.pgen.1010145>.
- Wakefield, J.G., J.Y. Huang, and J.W. Raff. 2000. Centrosomes have a role in regulating the destruction of cyclin B in early *Drosophila* embryos. *Curr Biol.* 10:1367-70. [https://doi.org/10.1016/s0960-9822\(00\)00776-4](https://doi.org/10.1016/s0960-9822(00)00776-4).
- Yuan, K., and P.H. O'Farrell. 2015. Cyclin B3 is a mitotic cyclin that promotes the metaphase-anaphase transition. *Curr Biol.* 25:811-816. <https://doi.org/10.1016/j.cub.2015.01.053>.

June 5, 2024

RE: JCB Manuscript #202310022R

Dr. Simon L Bullock
MRC Laboratory of Molecular Biology
Cell Biology
Francis Crick Avenue
Cambridge CB2 0QH
United Kingdom

Dear Dr. Bullock:

Thank you very much for your manuscript, "A force-sensitive mutation reveals a non-canonical role for dynein in anaphase progression". The manuscript has been assessed by the three original reviewers, and I am pleased to inform you that all the reviewers now support publication of the manuscript. As the reviewers #1 and #2 raise minor points, we would like to invite you to submit the final version after responding to these minor points, which we hope relatively straightforward to deal with.

To avoid unnecessary delays in the acceptance and publication of your paper, please carefully read the following information on formatting requirements.

A. MANUSCRIPT ORGANIZATION AND FORMATTING:

Full guidelines are available on our Instructions for Authors page, <http://jcb.rupress.org/submission-guidelines#revised>. Submission of a paper that does not conform to JCB guidelines will delay the acceptance of your manuscript.

- 1) Text limits: Character count for Articles is < 40,000, not including spaces. Count includes abstract, introduction, results, discussion, and acknowledgments. Count does not include title page, figure legends, materials and methods, references, tables, or supplemental legends.
- 2) Figures limits: Articles may have up to 10 main figures and 5 supplemental figures/tables.
- 3) Figure formatting: Scale bars must be present on all microscopy images, including inset magnifications. Molecular weight or nucleic acid size markers must be included on all gel electrophoresis. Please avoid pairing red and green for images and graphs to ensure legibility for color-blind readers. If red and green are paired for images, please ensure that the particular red and green hues used in micrographs are distinctive with any of the colorblind types. If not, please modify colors accordingly or provide separate images of the individual channels.
- 4) Statistical analysis: Error bars on graphic representations of numerical data must be clearly described in the figure legend. The number of independent data points (n) represented in a graph must be indicated in the legend. Statistical methods should be explained in full in the materials and methods. For figures presenting pooled data the statistical measure should be defined in the figure legends. Please also be sure to indicate the statistical tests used in each of your experiments (either in the figure legend itself or in a separate methods section) as well as the parameters of the test (for example, if you ran a t-test, please indicate if it was one- or two-sided, etc.). Also, if you used parametric tests, please indicate if the data distribution was tested for normality (and if so, how). If not, you must state something to the effect that "Data distribution was assumed to be normal but this was not formally tested."
- 5) Abstract and title: The abstract should be no longer than 160 words and should communicate the significance of the paper for a general audience. The title should be less than 100 characters including spaces. Make the title concise but accessible to a general readership.
- 6) Materials and methods: Should be comprehensive and not simply reference a previous publication for details on how an experiment was performed. Please provide full descriptions in the text for readers who may not have access to referenced manuscripts. We also provide a report from SciScore and an associate score, which we encourage you to use as a means of evaluating and improving the methods section.
- 7) Please be sure to provide the sequences for all of your primers/oligos and RNAi constructs in the materials and methods. You must also indicate in the methods the source, species, and catalog numbers (where appropriate) for all of your antibodies. Please also indicate the acquisition and quantification methods for immunoblotting/western blots.
- 8) Microscope image acquisition: The following information must be provided about the acquisition and processing of images:

- a. Make and model of microscope
- b. Type, magnification, and numerical aperture of the objective lenses
- c. Temperature
- d. Imaging medium
- e. Fluorochromes
- f. Camera make and model
- g. Acquisition software
- h. Any software used for image processing subsequent to data acquisition. Please include details and types of operations involved (e.g., type of deconvolution, 3D reconstitutions, surface or volume rendering, gamma adjustments, etc.).

10) Supplemental materials: There are strict limits on the allowable amount of supplemental data. Articles may have up to 5 supplemental figures. Please also note that tables, like figures, should be provided as individual, editable files. A summary of all supplemental material should appear at the end of the Materials and methods section.

13) ORCID IDs: ORCID IDs are unique identifiers allowing researchers to create a record of their various scholarly contributions in a single place. At resubmission of your final files, please provide an ORCID ID for all authors.

15) A data availability statement is required for all research article submissions. The statement should address all data underlying the research presented in the manuscript. Please visit the JCB instructions for authors for guidelines and examples of statements at (<https://rupress.org/jcb/pages/editorial-policies#data-availability-statement>).

Please note that JCB requires authors to submit Source Data used to generate figures containing gels and Western blots with all revised manuscripts. This Source Data consists of fully uncropped and unprocessed images for each gel/blot displayed in the main and supplemental figures. Since your paper includes cropped gel and/or blot images, please be sure to provide one Source Data file for each figure that contains gels and/or blots along with your revised manuscript files. File names for Source Data figures should be alphanumeric without any spaces or special characters (i.e., SourceDataF#, where F# refers to the associated main figure number or SourceDataFS# for those associated with Supplementary figures). The lanes of the gels/blots should be labeled as they are in the associated figure, the place where cropping was applied should be marked (with a box), and molecular weight/size standards should be labeled wherever possible. Source Data files will be directly linked to specific figures in the published article.

Journal of Cell Biology now requires a data availability statement for all research article submissions. These statements will be published in the article directly above the Acknowledgments. The statement should address all data underlying the research presented in the manuscript. Please visit the JCB instructions for authors for guidelines and examples of statements at (<https://rupress.org/jcb/pages/editorial-policies#data-availability-statement>).

B. FINAL FILES:

-- High-resolution figure and MP4 video files: See our detailed guidelines for preparing your production-ready images,

<https://jcb.rupress.org/fig-vid-guidelines>.

Thank you for your attention to these final processing requirements. Please revise and format the manuscript and upload materials within 7 days. If you need an extension for whatever reason, please let us know and we can work with you to determine a suitable revision period.

Thank you for this interesting contribution, we look forward to publishing your paper in Journal of Cell Biology.

Sincerely,

Hironori Funabiki
Monitoring Editor
Journal of Cell Biology

Tim Fessenden
Scientific Editor
Journal of Cell Biology

Reviewer #1 (Comments to the Authors (Required)):

The authors made a substantial effort to address all concerns raised in the first round of review. Importantly, the new experiments provide additional support for the conclusion that the dynein S3372C mutant prevents mitotic progression in the *Drosophila* embryo in a manner that is unrelated to dynein's known role in spindle assembly checkpoint silencing.

Minor comment:

In the abstract, I would remove "protein Mad2" from the sentence ending in "...silencing the spindle assembly checkpoint protein Mad2".

Reviewer #2 (Comments to the Authors (Required)):

I am mostly satisfied with the new data that have been added to the manuscript, as well as the revised manuscript. I have only a couple remaining minor concerns, described below. In particular, the authors should discuss, albeit briefly, the likely causal relationship between the S3372C and altered spindle morphology described in Figure 5. See point 2 below.

1) "The metaphase arrest and defects in centrosome arrangement were confirmed by time-lapse analysis of mutant embryos that had components of the spindle apparatus labeled fluorescently." I think this should be quantitated so that the extent of the arrest is quantitatively described (e.g., duration in mitosis).

2) "We also observed that, whilst spindle poles were still focused in S3372C embryos, they tended to be broader than in controls (e.g. Fig. 5, C and D; and Videos 1 and 2), leading to a rounder spindle shape (Fig. S2 F). These features of spindle morphology have been observed previously when centrosome number and arrangement are impaired (Wakefield et al., 2000)." Centrosome detachment, and defocused spindle poles have both been previously observed in cells depleted for dynein (e.g. Goshima, Nedelec and Vale, JCB 2005). This is consistent with the S3772C mutant impairing dynein function beyond the metaphase-to-anaphase transition, and should be discussed in the manuscript at the appropriate points. Note this also challenges the statement: "Thus, the S3372C mutation appears to affect a subset of dynein functions in the early embryo that include-and are perhaps limited to-the transition of mitotic spindles from metaphase to anaphase."

Reviewer #3 (Comments to the Authors (Required)):

In their revised manuscript Salvador-Garcia et al. have done a good job of addressing all of the concerns I raised in my initial review. In particular, the new data showing that Mad2 is normally required for the SAC in embryos, and that the embryos still arrest in mitosis in the absence of Mad2 is convincing. I am therefore very supportive of publication of the revised manuscript in The JCB.

Reviewer #1 (Comments to the Authors (Required)):

The authors made a substantial effort to address all concerns raised in the first round of review. Importantly, the new experiments provide additional support for the conclusion that the dynein S3372C mutant prevents mitotic progression in the *Drosophila* embryo in a manner that is unrelated to dynein's known role in spindle assembly checkpoint silencing.

Minor comment:

In the abstract, I would remove "protein Mad2" from the sentence ending in "...silencing the spindle assembly checkpoint protein Mad2".

This is a good suggestion. The abstract has been updated.

Reviewer #2 (Comments to the Authors (Required)):

I am mostly satisfied with the new data that have been added to the manuscript, as well as the revised manuscript. I have only a couple remaining minor concerns, described below. In particular, the authors should discuss, albeit briefly, the likely causal relationship between the S3372C and altered spindle morphology described in Figure 5. See point 2 below.

1) "The metaphase arrest and defects in centrosome arrangement were confirmed by time-lapse analysis of mutant embryos that had components of the spindle apparatus labeled fluorescently." I think this should be quantitated so that the extent of the arrest is quantitatively described (e.g., duration in mitosis).

We have now added a quantitative statement to the Video 1 legend about the penetrance and duration of the metaphase arrest in the time-lapse movies.

"In 57 out of 60 movies (of 15-min duration each) of S3372C embryos that expressed fluorescent markers of the mitotic apparatus, no progression of metaphase arrested spindles was observed. In the remaining movies, progression of highly abnormal mitotic spindles that were formed by fusion of two spindles was seen."

2) "We also observed that, whilst spindle poles were still focused in S3372C embryos, they tended to be broader than in controls (e.g. Fig. 5, C and D; and Videos 1 and 2), leading to a rounder spindle shape (Fig. S2 F). These features of spindle morphology have been observed previously when centrosome number and arrangement are impaired (Wakefield et al., 2000)." Centrosome detachment, and defocused spindle poles have both been previously observed in cells depleted for dynein (e.g. Goshima, Nedelec and Vale, JCB 2005). This is consistent with the S3372C mutant impairing dynein function beyond the metaphase-to-anaphase transition, and should be discussed in the manuscript at the appropriate points. Note this also challenges the statement: "Thus, the S3372C mutation appears to affect a subset of dynein functions in the early embryo that include-and are perhaps limited to-the transition of mitotic spindles from metaphase to anaphase."

We don't follow this argument from the reviewer as spindles are still focused in the mutant embryos, as stated. Moreover, the statement they refer to about a subset of dynein functions being affected by S3372C is clearly concerned with the observation that dynein-dependent maintenance of polar bodies in M-phase is intact in the mutant embryos. Moreover, in the Discussion, we comment that a direct effect of S3372C on dynein function at spindle poles or centrosomes (in addition to a role at kinetochores) cannot be ruled out.

“Although we cannot exclude an effect of the S3372C mutation on dynein’s functions in other parts of the spindle apparatus, such as the poles and centrosomes, the build-up of dynein in the vicinity of the kinetochore, as well as slower transport of Rod away from this site, suggest that impaired kinetochore functions of dynein make an important contribution to the mitotic arrest.”

We therefore feel that further discussion of the point raised by the reviewer is unnecessary.

Reviewer #3 (Comments to the Authors (Required)):

In their revised manuscript Salvador-Garcia et al. have done a good job of addressing all of the concerns I raised in my initial review. In particular, the new data showing that Mad2 is normally required for the SAC in embryos, and that the embryos still arrest in mitosis in the absence of Mad2 is convincing. I am therefore very supportive of publication of the revised manuscript in The JCB.

This reviewer did not raise any additional issues.